# VARIANCE-REDUCING COUPLINGS FOR RANDOM FEATURES

**Isaac Reid**[1], **Stratis Markou**[1], **Krzysztof Choromanski**[*2,3], **Richard E. Turner**[1], **Adrian Weller**[1,4]

[1]University of Cambridge, [2]Google DeepMind, [3]Columbia, [4]Alan Turing Institute

## ABSTRACT

Random features (RFs) are a popular technique to scale up kernel methods in machine learning, replacing exact kernel evaluations with stochastic Monte Carlo estimates. They underpin models as diverse as efficient transformers (by approximating attention) to sparse spectrum Gaussian processes (by approximating the covariance function). Efficiency can be further improved by speeding up the convergence of these estimates: a *variance reduction* problem. We tackle this through the unifying lens of *optimal transport*, finding couplings to improve RFs defined on both Euclidean and discrete input spaces. They enjoy theoretical guarantees and sometimes provide strong downstream gains, including for scalable inference on graphs. We reach surprising conclusions about the benefits and limitations of variance reduction as a paradigm, showing that other properties of the coupling should be optimised for attention estimation in efficient transformers.

## 1 INTRODUCTION

Kernel methods are ubiquitous in machine learning (Canu and Smola, 2006; Kontorovich et al., 2008; Campbell, 2002). Through the kernel trick, they provide a mathematically principled and elegant way to perform nonlinear inference using linear learning algorithms. The eponymous *kernel function* $k : \mathcal{X} \times \mathcal{X} \to \mathbb{R}$ measures the 'similarity' between two datapoints. The input domain $\mathcal{X}$ may be continuous, e.g. the set of vectors in $\mathbb{R}^d$, or discrete, e.g. the set of graph nodes or entire graphs.

**Random features for kernel approximation.** Though very effective on small datasets, kernel methods suffer from poor scalability. The need to materialise and invert the *Gram matrix* $\mathbf{K} := [k(\boldsymbol{x}_i, \boldsymbol{x}_j)]_{i=1}^N$ leads to a time complexity cubic in the size of the dataset $N$. Substantial research has been dedicated to improving scalability by approximating this matrix, a prominent example being *random features* (RFs) (Rahimi and Recht, 2007; 2008; Avron et al., 2017b; Liu et al., 2022). These randomised mappings $\phi : \mathbb{R}^d \to \mathbb{R}^s$ construct low-dimensional or sparse feature vectors that satisfy

$$k(\boldsymbol{x}, \boldsymbol{y}) = \mathbb{E}\left(\phi(\boldsymbol{x}; \{\boldsymbol{\omega}_i\}_{i=1}^m)^\top \phi(\boldsymbol{y}; \{\boldsymbol{\omega}_i\}_{i=1}^m)\right). \tag{1}$$

The expectation $\mathbb{E}$ is taken over an ensemble of *random frequencies* $\{\boldsymbol{\omega}_i\}_{i=1}^m$ drawn from a distribution $\eta$. The space in which $\{\boldsymbol{\omega}\}_{i=1}^m$ live and manner in which they are combined to construct $\phi(\boldsymbol{x}; \{\boldsymbol{\omega}_i\}_{i=1}^m)$ depends on the particular input space $\mathcal{X}$ and kernel function $k$ being approximated. This paper will consider several examples. Hereafter, the dependence on $\{\boldsymbol{\omega}_i\}_{i=1}^m$ will be suppressed to reduce notational clutter. The set of RFs $\{\phi(\boldsymbol{x}_i)\}_{i=1}^N$ can be used to construct a low-rank or sparse approximation of the Gram matrix, providing substantial space and time complexity savings. RFs exist for a variety of kernels, including for continuous and discrete input spaces (Dasgupta et al., 2010; Johnson, 1984; Choromanski et al., 2020; Rahimi and Recht, 2007; Tripp et al., 2024).

**Variance reduction for RFs.** Replacing $\mathbb{E}$ by the mean over random samples of $\{\boldsymbol{\omega}_i\}_{i=1}^m$, Eq. 1 can be understood as a *Monte Carlo* (MC) estimate of $k$. In applications, it is often found that this estimate converges slowly. This can be addressed by taking many samples $m$, but this undermines the efficiency gains of RFs. Therefore, substantial effort has been dedicated to *reducing the variance* of the kernel estimates. Variance reduction methods include quasi-Monte Carlo (QMC; Dick et al., 2013; Yang et al., 2014a), common random numbers (CRNs; Glasserman and Yao, 1992), antithetic variates (Hammersley and Morton, 1956) and structured Monte Carlo (SMC; Yu et al., 2016). These techniques work by replacing i.i.d. frequencies $\{\boldsymbol{\omega}_i\}_{i=1}^m$ by a *dependent ensemble*, with the sample dependencies designed to improve RF convergence.

---

[*]Senior lead.

**Limitations of previous techniques.** The best choice of dependencies between $\{\boldsymbol{\omega}_i\}_{i=1}^m$ is an active research area. Though straightforward to apply, standard QMC techniques are suboptimal. They are based on hard-coded 'low-discrepancy sequences' so typically do not incorporate information about the particular kernel $k$ being approximated. Empirical performance may be poor and theoretical guarantees lacking in the low-sample, high-dimensional regime (Rowland et al., 2018; Morokoff and Caflisch, 1995), which is precisely where RFs are most important. On the other hand, hand-crafted SMC dependencies, which impose strict geometrical conditions like orthogonality between frequencies, tend to fare better (Yu et al., 2016). But they are difficult to design, theoretical guarantees are hard-won and optimality is not guaranteed. RFs for estimating kernels defined on discrete spaces like the nodes of a graph have only recently been developed (Choromanski, 2023; Tripp et al., 2024), so here very few effective variance reduction techniques have even been proposed. This paper asks: *can we devise a principled framework for coupling RFs, providing variance reduction across basis functions and input domains, including with very few samples?*

**Optimal transport.** To answer this, we propose to frame variance reduction as *optimal transport* (OT): an active research area of applied mathematics that studies how to move (probability) mass between distributions as efficiently as possible (Villani et al., 2009). This novel perspective equips us with proof techniques and numerical tools to identify the *best possible dependencies* between samples, giving lower kernel estimator variance compared to previous approaches. OT allows us to improve couplings for RFs in both Euclidean and discrete spaces, including with different basis functions. To our knowledge, this has never before been achieved in the same paper.

**Our contributions.** This work presents unifying strategies to reduce the variance of random features.

1. We frame the problem of variance reduction of RFs as *optimal transport* (OT) (**Sec. 2**), and use this perspective to improve the convergence of *three* popular classes of RFs: random Fourier features, random Laplace features and graph random features.
2. For random Fourier features (RFFs) and random Laplace features (RLFs), we exactly solve the OT problem for the *norms* of $m = 2$ orthogonal frequencies (**Sec. 3**). We introduce *pairwise norm-coupling*, which guarantees lower variance for arbitrary $m$.
3. For graph random features (GRFs), we couple the *lengths* of random walks by finding a bipartite matching between the quantiles of the marginal distributions (**Sec. 4**). This is the first time a coupling between random walks has been optimised using data, beating hard-coded algorithms.
4. We test our algorithms on UCI datasets and real-world graphs, verifying that OT couplings substantially reduce kernel estimator variance (**Secs 3 and 4**). We show that this sometimes translates to much better performance in downstream tasks, including for inference with scalable graph-based Gaussian processes. However, we also reach surprising conclusions about the limitations of variance reduction for RFs, including for efficient transformers.

All proofs are saved for the Appendices, but are also sketched in the main body where space allows.

## 2 PRELIMINARIES

**From kernel estimation to optimal transport (OT).** Define the *kernel estimator* $\widehat{k}(\boldsymbol{x}_i, \boldsymbol{x}_j) \coloneqq \phi(\boldsymbol{x}_i)^\top \phi(\boldsymbol{x}_j)$. Recall that $\phi(\cdot)$ is computed using random frequencies $\{\boldsymbol{\omega}_i\}_{i=1}^m$, with the space in which they live, distribution from which they are drawn, and manner in which they are combined dependent on the particular kernel being approximated. The estimator is unbiased provided each frequency $\boldsymbol{\omega}_i$ obeys some marginal distribution $\eta$. Importantly, independence of $\{\boldsymbol{\omega}_i\}_{i=1}^m$ is *not* required: any joint distribution with marginals $\eta$ gives an unbiased estimator. We refer to the set of such joint distributions as *couplings*.

The coupling between the frequencies determines the estimator variance. We want to solve:

$$\text{minimise } \mathcal{I}(\mu) = \mathbb{E}_{\boldsymbol{\omega}_{1:m} \sim \mu} c(\boldsymbol{\omega}_{1:m}) \quad \text{for} \quad \mu \in \Lambda_m(\eta), \tag{2}$$

where we defined the *cost function* $c(\boldsymbol{\omega}_{1:m}) \coloneqq \left(\phi(\boldsymbol{x})^\top \phi(\boldsymbol{y})\right)^2$ and $\Lambda_m(\eta)$ denotes the set of couplings of $m$ random variables with marginal measures $\eta$. This is precisely the *Kantorovich formulation* of a multi-marginal OT problem (see Eq. 4 of the seminal OT text of Villani (2021)). We will generally consider cost functions where the minimiser exists and we want to find efficient new MC couplings, so the task is to find the *optimal coupling* $\mu^* = \arg\min_{\mu \in \Lambda_m(\chi_d)} \left[\mathbb{E}_{\omega_{1:m} \sim \mu} c(\omega_{1:m})\right]$ with the smallest estimator variance. The relationship between variance reduction and OT was also noted by Rowland et al. (2018) in a different context.

**(Approximately) solving the OT problem.** The formulation of Eq. 2 depends on the particular RF mechanism and kernel being approximated. We will show that one can solve it exactly for RFFs and RLFs (**Sec. 3**) and approximately for GRFs (**Sec. 4**), which have input domains $\mathcal{X} = \mathbb{R}^d$ ($d$-dimensional Euclidean space) and $\mathcal{X} = \mathcal{N}$ (the set of graph nodes) respectively. This gives new couplings with lower RF variance than previous algorithms.

## 3 RANDOM FOURIER FEATURES AND RANDOM LAPLACE FEATURES

**RFFs and RLFs.** To begin, we consider the task of approximating the popular *Gaussian kernel* $k(\boldsymbol{x}_i, \boldsymbol{x}_j) := \exp(-\|\boldsymbol{x}_i - \boldsymbol{x}_j\|^2/2)$ with data $\{\boldsymbol{x}_i\}_{i=1}^N \subset \mathbb{R}^d$. This can be achieved using Rahimi and Recht's celebrated *random Fourier features* (RFFs) (Rahimi and Recht, 2007),

$$\phi_{\text{RFF}}(\boldsymbol{x}) = \sqrt{\frac{1}{m}} \left( \odot_{i=1}^m \left[ \sin(\boldsymbol{\omega}_i^\top \boldsymbol{x}), \cos(\boldsymbol{\omega}_i^\top \boldsymbol{x}) \right] \right), \tag{3}$$

where $\odot$ denotes concatenation. These provide an unbiased estimate if the frequencies $\{\boldsymbol{\omega}_i\}_{i=1}^m$ are marginally Gaussian, $\boldsymbol{\omega}_i \sim \mathcal{N}(0, \mathbf{I}_d)$. RFFs are widely used for scaling kernel methods such as Gaussian processes (GPs; Williams and Rasmussen, 2006) and support vector machines (SVMs; Scholkopf and Smola, 2018). The time complexity of computing the exact posterior of a GP is $\mathcal{O}(N^3)$, where $N$ is the number of datapoints. Using RFFs, one can approximate the posterior with $m \ll N$ features, reducing this cost to $\mathcal{O}(Nm^2)$. Changing basis functions, $k(\boldsymbol{x}_i, \boldsymbol{x}_j)$ can also be approximated using *random Laplace features* (RLFs) (Yang et al., 2014b),

$$\phi_{\text{RLF}}(\boldsymbol{x}) = \sqrt{\frac{1}{m}} \exp(-\|\boldsymbol{x}\|^2)(\odot_{i=1}^m \exp(\boldsymbol{\omega}_i^\top \boldsymbol{x})), \tag{4}$$

where again $\boldsymbol{\omega}_i \sim \mathcal{N}(0, \mathbf{I}_d)$. Unlike RFFs, RLFs guarantee positive kernel estimates. This makes them better suited to approximating attention in efficient transformers (Choromanski et al., 2020), where negative estimates cause training instabilities. Using $m$ RLFs to get a low-rank decomposition of attention with $N$ $d$-dimensional tokens, one can reduce the time complexity of transformers from $\mathcal{O}(N^2 + Nd)$ to $\mathcal{O}(Nmd)$ with low performance loss.

**Orthogonal random features.** A common variance reduction technique for both RFFs and RLFs is the *orthogonality trick* (Yu et al., 2016; Rowland et al., 2018; Reid et al., 2023; Choromanski et al., 2018). Exploiting the isotropy of $\mathcal{N}(0, \mathbf{I}_d)$, one can constrain the frequency vectors $\{\boldsymbol{\omega}_i\}_{i=1}^m$ to be exactly orthogonal whilst preserving their marginal distributions. This is found to reduce the kernel estimator variance and improve performance in downstream tasks. Whilst this technique couples the *directions* of the random frequencies $\{\widehat{\boldsymbol{\omega}}_i\}_{i=1}^m$, their *norms* $\{\omega_i\}_{i=1}^m$ (with $\omega_i := \|\boldsymbol{\omega}_i\|_2$) are left independent so the coupling is suboptimal. By solving an OT problem, we will show how coupling the norms can further reduce estimator variance.

### 3.1 SOLVING THE OT PROBLEM FOR MAXIMAL VARIANCE REDUCTION

Consider an ensemble of $m$ orthogonal random frequency directions $\{\widehat{\boldsymbol{\omega}}_i\}_{i=1}^m$, jointly randomly rotated so they are marginally isotropic. Our task is to couple their norms $\{\omega_i\}_{i=1}^m$ to suppress the RFF and RLF kernel estimator variance. The marginal distribution of each $\omega_i$ must be $\chi_d$ (a Chi distribution with $d$ degrees of freedom) to ensure that each $\boldsymbol{\omega}_i$ is marginally Gaussian. We can extend recent results by Reid et al. (2023) to compute the OT cost functions.

**Lemma 3.1** (OT formulation for RFFs and RLFs). *When estimating $k(\boldsymbol{x}, \boldsymbol{y})$ with $m$ orthogonal RFFs and RLFs, the OT formulation of the variance reduction problem is:*

$$\mu^* = \arg\min_{\mu \in \Lambda_m(\chi_d)} \left[ \mathbb{E}_{\omega_{1:m} \sim \mu} c(\omega_{1:m}) \right], \quad where \tag{5}$$

$$c_{RFF}(\omega_{1:m}) = \sum_{i,j \neq i}^m \sum_{k=0}^\infty \frac{(-1)^k z^{2k} \left( \omega_i^2 + \omega_j^2 \right)^k}{2^{2k} k! \Gamma(k + \frac{d}{2})}, \quad c_{RLF}(\omega_{1:m}) = \sum_{i,j \neq i}^m \sum_{k=0}^\infty \frac{v^{2k} (\omega_i^2 + \omega_j^2)^k}{2^{2k} k! \Gamma(k + \frac{d}{2})}, \tag{6}$$

*with $z := \|\boldsymbol{x} - \boldsymbol{y}\|_2$ and $v := \|\boldsymbol{x} + \boldsymbol{y}\|_2$. $\Gamma$ is the gamma function.*

This is a tough multi-marginal OT problem. However, remarkably, we can solve it *exactly*, under mild asymptotic assumptions for RFFs, when $m = 2$. The following result is novel.

**Theorem 3.2** (Solution to OT problem when $m = 2$). *Denote by $F_{\chi_d}(\cdot)$ the cumulative distribution function (CDF) of $\chi_d$. Consider $m = 2$ orthogonal frequencies with norms $(\omega_1, \omega_2)$. For RLFs, the OT problem in Eq. 5 is solved by the* negative monotone *coupling*

$$F_{\chi_d}(\omega_1) + F_{\chi_d}(\omega_2) = 1. \tag{7}$$

*For RFFs, Eq. 7 ensures lower cost than any other coupling, provided $z$ is sufficiently small.*

*Proof sketch.* We defer a full proof of this important result to App. A.2; here is a brief sketch. OT plans satisfy a property called 'c-monotonicity', which specifies how the support of the optimal coupling depends on the cost function. For RLFs, $c_{\text{RLF}}$ immediately implies negative monotonicity (Eq. 7). For RFFs, this is only true for the first nontrivial term in $z$. By bounding the contribution from the remaining terms, one can show that Eq. 7 still guarantees lower variance than any other coupling if $z$ is small enough. Specifically, letting $\mu_{\text{NM}}$ denote the negative monotone coupling, for any other coupling $\mu' \in \Lambda_2(\eta) \setminus \{\mu_{\text{NM}}\}$ there exists some constant $\delta(\mu') > 0$ such that $\mathcal{I}(\mu_{\text{NM}}) < \mathcal{I}(\mu')$ for all $z < \delta$ (Lemma A.6). $\qquad\square$

Given $m = d$ orthogonal frequencies, one can partition the ensemble into $\lfloor \frac{d}{2} \rfloor$ orthogonal pairs, with one remaining frequency if $d$ is odd. For every pair, one can impose negative monotone coupling (Eq. 7). We refer to such ensembles as *pairwise norm-coupled* (PNC).

> **Definition 3.3** (Pairwise norm-coupled RFs). *RFs are* pairwise norm-coupled *(PNC) if $d$ orthogonal frequencies $\{\omega_i\}_{i=1}^d$ are arranged in $\lfloor \frac{d}{2} \rfloor$ pairs, each of which is negative montone-coupled so that $F_{\chi_d}(\omega_1) + F_{\chi_d}(\omega_2) = 1$. Different pairs are independent.*

PNC is no more expensive than i.i.d. norms. To reduce the variance further, one can take multiple independent PNC ensembles. An important corollary of Thm. 3.2 is as follows.

**Corollary 3.4** (Superiority of pairwise norm-coupled RFs). *For **any** $m$, the variance of pairwise norm-coupled RFs is **guaranteed to be lower** than orthogonal RFs with independent norms, in full generality for RLFs and provided $z$ is small enough for RFFs.*

Negative monotone coupling differs from OT plans usually seen in machine learning; it is a *space-filling* coupling that seeks long transport plans that give diverse samples. However, it is a popular heuristic technique for variance reduction via common random numbers (CRNs) in computational statistics (Glasserman and Yao, 1992). To our knowledge, this is the first result applying it to improving the convergence of orthogonal RFs, and the first corresponding guarantees for variance reduction. We make one further theoretical contribution for RLFs.

**Theorem 3.5** (Recovering antithetic sampling with RLFs). *For RLFs with $m = 2$ frequencies whose respective orientations $(\widehat{\omega}_1, \widehat{\omega}_2)$ are unconstrained, variance is minimised by conditioning that $\omega_1 = -\omega_2$ almost surely (that is, opposite directions and equal norms).*

This coupling is known as *antithetic sampling* (Hammersley and Morton, 1956). Thm. 3.5 shows that, given a PNC ensemble $\{\omega_i\}_{i=1}^d$, we can obtain further variance reduction by augmenting it to $\{\pm\omega_i\}_{i=1}^d$. Antithetic sampling is also a common (though often heuristically motivated) variance reduction strategy used e.g. when estimating attention in Performers (Choromanski et al., 2020). We can reinterpret its effectiveness as an OT coupling.

## 3.2 PUSHING FURTHER WITH NUMERICAL OT SOLVERS

**Multi-marginal OT.** In Sec. 3.1 we proposed PNC RFs: a computationally efficient coupling that is guaranteed to reduce variance for any $m$. We obtained it by solving the variance reduction OT problem exactly in $m = 2$, then combining $\lfloor \frac{d}{2} \rfloor$ independent copies to get the ensemble. Can we do better by inducing dependencies between the *all* the $m$ frequencies' norms? Solving this multi-marginal OT problem analytically is a tough open problem.

**Copulas as numerical OT solvers.** Whilst an analytic solution to the multi-marginal OT variance reduction problem is (for now) out of reach, we can make progress using a numerical OT solver. Our strategy is to restrict $\Lambda_m(\chi_d)$, the full set of joint distributions over $m$ random variables with $\chi_d$ marginals, to a tractable subset amongst which we can efficiently optimise and sample. One such subset is provided by *Gaussian copulas* (Nelsen, 2006; Haugh, 2016): joint distributions obtained by

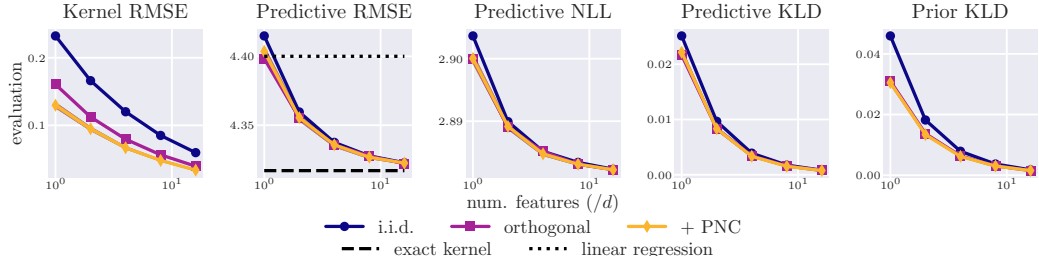

Figure 1: **Downstream performance on a split of the POWER dataset.** Kernel estimator RMSE, test predictive RMSE, test negative log likelihoods, KL divergence to the true predictive posterior, and KL divergence to the true prior, estimated with RFFs. We have successfully reduced the variance of the kernel approximation, but this may not help all downstream metrics. We also include test predictive RMSEs with the exact kernel (to which we converge) and linear regression for comparison.

| FOURIER FEATURES | CONCRETE | ABALONE | CPU | POWER | AIRFOIL | BOSTON |
|---|---|---|---|---|---|---|
| I.I.D. | $1.000 _{\pm 0.028}$ | $1.000 _{\pm 0.042}$ | $1.000 _{\pm 0.082}$ | $1.000 _{\pm 0.037}$ | $1.000 _{\pm 0.023}$ | $1.000 _{\pm 0.018}$ |
| HALTON | $1.028 _{\pm 0.029}$ | $0.991 _{\pm 0.042}$ | $0.995 _{\pm 0.082}$ | $0.913 _{\pm 0.033}$ | $0.927 _{\pm 0.021}$ | $1.176 _{\pm 0.022}$ |
| ORTHOGONAL | $0.627 _{\pm 0.019}$ | $0.535 _{\pm 0.023}$ | $0.617 _{\pm 0.070}$ | $0.669 _{\pm 0.024}$ | $0.586 _{\pm 0.015}$ | $0.639 _{\pm 0.016}$ |
| + PNC | $\mathbf{0.563} _{\pm 0.019}$ | $\mathbf{0.433} _{\pm 0.019}$ | $\mathbf{0.544} _{\pm 0.071}$ | $\mathbf{0.547} _{\pm 0.020}$ | $\mathbf{0.481} _{\pm 0.011}$ | $\mathbf{0.606} _{\pm 0.018}$ |
| LAPLACE FEATURES | CONCRETE | ABALONE | CPU | POWER | AIRFOIL | BOSTON |
| I.I.D. | $1.000 _{\pm 0.092}$ | $1.000 _{\pm 0.036}$ | $1.000 _{\pm 0.086}$ | $1.000 _{\pm 0.018}$ | $1.000 _{\pm 0.026}$ | $1.000 _{\pm 0.029}$ |
| HALTON | $0.721 _{\pm 0.067}$ | $0.777 _{\pm 0.031}$ | $0.779 _{\pm 0.084}$ | $0.728 _{\pm 0.015}$ | $0.721 _{\pm 0.021}$ | $0.893 _{\pm 0.028}$ |
| ORTHOGONAL | $0.418 _{\pm 0.041}$ | $0.546 _{\pm 0.026}$ | $\mathbf{0.614} _{\pm 0.098}$ | $0.527 _{\pm 0.013}$ | $0.489 _{\pm 0.016}$ | $0.360 _{\pm 0.019}$ |
| + PNC + ANTITHETIC | $\mathbf{0.367} _{\pm 0.043}$ | $\mathbf{0.486} _{\pm 0.027}$ | $0.618 _{\pm 0.119}$ | $\mathbf{0.438} _{\pm 0.013}$ | $\mathbf{0.418} _{\pm 0.016}$ | $\mathbf{0.324} _{\pm 0.019}$ |

Table 1: Performance of RFFs and RLFs on kernel estimation with UCI datasets with different coupling schemes, taking $kd$ random frequencies for RFFs and $2kd$ for RLFs where $k \in \mathbb{N}$ (see main text for details). We show RMSEs to the ground truth kernel values, normalised such that the RMSE of the I.I.D. estimator is equal to one. Lower is better. Error bars are standard errors on RMSEs.

taking a multivariate Gaussian and pushing each of its coordinates forward first with the Gaussian CDF $F_{\mathcal{N}}$, and then the $\chi_d$ inverse CDF $F_{\chi_d}^{-1}$. If the diagonal terms of the underlying Gaussian covariance matrix $\mathbf{\Sigma} \in \mathbb{R}^{m \times m}$ are equal to $1$ (i.e. it is a correlation matrix), this has the prescribed marginals so unbiasedness is baked in. Meanwhile, correlations *between* the random variables are controlled by the off-diagonal entries of $\mathbf{\Sigma}$. This parameterises a broad set of couplings, including PNC (Def. 3.3). In App. A.5 we demonstrate that it is possible to use gradient descent with the reparameterisation trick to *learn* the optimal copula covariance matrix $\mathbf{\Sigma}$, approximately solving the multi-marginal OT problem. We do this by minimising the kernel approximation error on training data, exploiting the fact that all operations to construct the features are differentiable. In doing so, we optimise the RF coupling. Remarkably, this data-dependent optimisation does *not* to find couplings much better than PNC: see the training curves in Fig. 6. This suggests that our scheme may already be close optimal for $m \neq 2$. Intuitively, one cannot simultaneously anticorrelate too many random variables, so strong pairwise couplings already perform very well. Whilst copulas have previously been used as numerical OT solvers (Chi et al., 2019), this is (to our knowledge) their first application to learning a Monte Carlo coupling.

## 3.3 EXPERIMENTS FOR NORM-COUPLED RFS

To test PNC RFs (Def. 3.3), we now compute kernel estimates with RFFs and RLFs for UCI datasets. We choose the kernel lengthscale parameters based on a training set, by training a GP (RFFs) or selecting reasonable values for Performers (RLFs) (Choromanski et al., 2020). We then compute the kernel approximation RMSE (Frobenius norm error of $\widehat{\mathbf{K}}$) on a test set. Full details are in App. B.1.

**Results for variance reduction.** Table 1 shows the results. For RFFs, we take $m = kd$ orthogonal frequencies, with $k \in \mathbb{N}$. For RLFs, we also include their antiparallel directions, giving $m = 2kd$ frequencies. For each dataset, the RMSEs are normalised by the result with i.i.d. features, which means the results are the same for all $k$. As a baseline, we include RFs constructed using *Halton sequences* (Dick et al., 2013; Yang et al., 2014a), a fixed, off-the-shelf QMC scheme that can provide small gains but is clearly suboptimal. The third row shows orthogonal frequencies with independent

norms (Yu et al., 2016). When we *also* couple the frequencies' norms using our PNC scheme (plus antithetic sampling for RLFs, due to Thm 3.5), we access even lower estimator variance at no extra computational cost. Note that the small $z$ condition for RFFs is found to be nonrestrictive in practice.

**Downstream tasks.** We have achieved our objective of variance reduction, a popular and intensely studied goal in the literature (Yu et al., 2016; Rowland et al., 2018; Reid et al., 2023; Likhosherstov et al., 2022; Yang et al., 2014a; Le et al., 2013; Bojarski et al., 2017; Choromanski et al., 2017; Lyu, 2017; Shen et al., 2017; Dao et al., 2017; Munkhoeva et al., 2018). It is conventionally understood that PNC RFs should therefore improve downstream performance in applications. Surprisingly, when we run exhaustive Gaussian process experiments in App. B.2, we do *not* observe such a gain. Fig. 1 demonstrates this on a train-test split of the POWER dataset for different values of $m$ (we use a single split to highlight the small spread between different coupling methods on fixed data). We find that lower kernel estimator RMSE may not improve downstream performance. This is because, when optimising a coupling, we minimise the variance of *pointwise* kernel estimates $\{k(\boldsymbol{x}_i, \boldsymbol{x}_j)\}_{i,j=1}^N$. However, functions like the predictive mean and KL divergence are highly nonlinear in these estimates. For example, they may involve inverting a Gram matrix. Downstream quantities therefore depend on the *joint* distribution of the kernel estimates, which are modified nontrivially by the coupling. Variance reduction alone cannot guarantee an improvement.

**Performers.** As a concrete example, consider estimating *attention*, $\widehat{a}_{ij} := \widehat{k}(\boldsymbol{x}_i, \boldsymbol{x}_j) / \sum_{l=1}^N \widehat{k}(\boldsymbol{x}_i, \boldsymbol{x}_l)$ using random Laplace features (Choromanski et al., 2020). This normalises the kernel evaluation between the $i$ and $j$ tokens by the sum with all the other tokens. Taylor expanding, if the kernel estimators have equal means $\mu$, the average mean square error $\mathrm{MSE}(\widehat{a}_i) := \frac{1}{N} \sum_{j=1}^N \mathrm{MSE}(\widehat{a}_{ij})$ obeys

$$\mathrm{MSE}(\widehat{a}_i) = \frac{1}{N^2 \mu^2} \left( \frac{1}{N} \sum_{j=1}^N \mathrm{Var}(\widehat{k}(\boldsymbol{x}_i, \boldsymbol{x}_j)) - \frac{1}{N^2} \sum_{j_1, j_2 = 1}^N \mathrm{Cov}(\widehat{k}(\boldsymbol{x}_i, \boldsymbol{x}_{j_1}), \widehat{k}(\boldsymbol{x}_i, \boldsymbol{x}_{j_2})) \right) + \mathcal{O}(\frac{1}{N^3}). \quad (8)$$

By coupling the frequency norms, PNC reduces $\mathrm{Var}(\widehat{k}(\boldsymbol{x}_i, \boldsymbol{x}_j))$ as intended. However, it also reduces the covariance $\mathrm{Cov}(\widehat{k}(\boldsymbol{x}_i, \boldsymbol{x}_{j_1}), \widehat{k}(\boldsymbol{x}_i, \boldsymbol{x}_{j_2}))$, so $\mathrm{MSE}(\widehat{a}_i)$ does not actually substantially improve overall. In stark contrast, if we instead take the *positive* monotone (PM) coupling where $\{\omega_i\}_{i=1}^m$ are all equal almost surely, then $\mathrm{Var}(\widehat{k}(\boldsymbol{x}_i, \boldsymbol{x}_j))$ is *maximised* (see App. B.3). But these strong, positive correlations also increase $\mathrm{Cov}(\widehat{k}(\boldsymbol{x}_i, \boldsymbol{x}_{j_1}), \widehat{k}(\boldsymbol{x}_i, \boldsymbol{x}_{j_2}))$ by an even greater amount. Hence, we find that $\mathrm{MSE}(\widehat{a}_i)$ *falls* (see Fig. 7). This is surprising: maximising the pointwise kernel estimator variance by solving the OT problem with the 'wrong' sign on the cost function reduces the MSE of the attention scores after normalisation. In fact, the improvement is so big that

Table 2: Performer test accuracies on ImageNet with different coupling schemes. Counterintuitively, *maximising* the pointwise kernel estimator variance by positively correlating feature norms boosts performance – a different OT problem to the naive, obvious choice.

| LAPLACE FEATURES | TEST ACC. |
|---|---|
| ORTHOGONAL | $0.625_{\pm 0.003}$ |
| ORTHOGONAL + PNC | $0.620_{\pm 0.003}$ |
| ORTHOGONAL + PM | $\mathbf{0.633}_{\pm 0.003}$ |

it increases the average test accuracy of Performers trained on ImageNet (Deng et al., 2009) by **+0.8%**, whereas PNC makes no statistically significant difference. See Table 2. This demonstrates the limitations of simple variance reduction and invites a more careful treatment, considering the downstream quantities of interest. App. B.3 gives further discussion and transformer training details.

## 4 GRAPH RANDOM FEATURES

We now shift our attention from $\mathbb{R}^d$ to the discrete domain. Consider an undirected graph $\mathcal{G}(\mathcal{N}, \mathcal{E})$ where $\mathcal{N} := \{v_1, ..., v_N\}$ is the set of nodes and $\mathcal{E}$ is the set of edges, with $(v_i, v_j) \in \mathcal{E}$ if and only if there exists an edge between $v_i$ and $v_j$ in $\mathcal{G}$. *Graph node kernels* $k : \mathcal{N} \times \mathcal{N} \to \mathbb{R}$ are positive definite, symmetric functions defined on pairs of nodes of $\mathcal{G}$, reflecting some notion of their 'closeness' via the graph edges and weights. $k$ captures the structure of $\mathcal{G}$, letting practitioners repurpose popular kernelised learning algorithms to the discrete domain (Smola and Kondor, 2003b;a). Examples include the diffusion, regularised Laplacian, cosine and random walk kernels, all of which are typically considered as functions of the graph Laplacian matrix (Kondor and Lafferty, 2002). We give a short introduction in App. C.1.

**Graph random features.** As in the Euclidean setting, graph kernel methods scale poorly due to the $\mathcal{O}(N^3)$ time complexity of inverting the Gram matrix $\mathbf{K} := [k(v_i, v_j)]_{i,j=1}^N$. In fact, even *computing* $\mathbf{K}$ often incurs a cubic cost since it involves multiplying large adjacency matrices. Research has been dedicated to improving efficiency by approximating $\mathbf{K}$, including *graph random features* (GRFs)

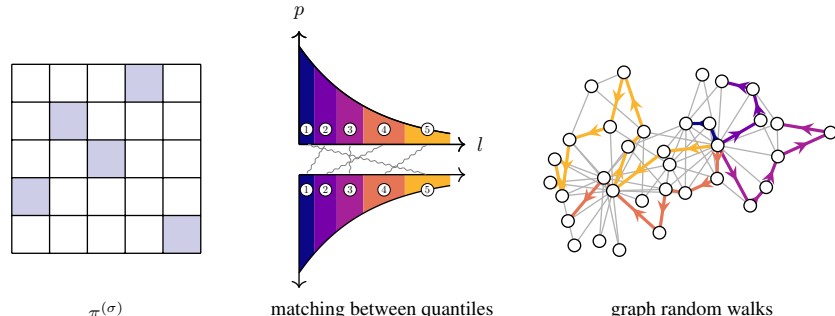

$\pi^{(\sigma)}$         matching between quantiles         graph random walks

Figure 2: Schematic overview of $\sigma$-coupled GRFs with $\sigma = 24351$. *Left*: permutation density with uniform marginals. *Centre*: bipartite matching between quantiles of geometric distributions over walk lengths. *Right*: ensemble of graph random walks with lengths to be coupled.

(Choromanski, 2023; Reid et al., 2024b). These are sparse vectors $\{\phi_{\text{GRF}}(v_i)\}_{i=1}^N \subset \mathbb{R}^N$ that satisfy $\mathbf{K}_{ij} = \mathbb{E}\left[\phi_{\text{GRF}}(v_i)^\top \phi_{\text{GRF}}(v_j)\right]$, so their dot product is equal to the true graph kernel in expectation. GRFs are constructed in subquadratic time.

**Coupled random walks.** For GRFs, the 'frequencies' $\{\boldsymbol{\omega}_i\}_{i=1}^m$ are *simple random walks*: sequences of graph nodes $(v_i)_{i=1}^l$ with $(v_i, v_{i+1}) \in \mathcal{E}$. At every timestep, the walker chooses one of its neighbours uniformly and at random. The length of the walk $l$ is also a random variable, drawn from a geometric distribution $l \sim \text{G}(p), p \in (0, 1)$. In other words, the walker terminates with probability $p$ at every timestep. Random walks are usually taken to be independent, but this can lead to slow mixing times, poor efficiency and high kernel estimator variance (Alon et al., 2007; Zhou et al., 2015). A natural question is: *can we couple graph random walks to improve the convergence of GRFs?*

Sec. 3 couples Gaussian vector norms. A simple, analogous approach to couple random walks is via their *lengths*. For GRFs, the constraint for unbiasedness is that the marginal distribution of each variable $l$ must remain geometric. Reid et al. (2024c) proposed a simple, ad-hoc algorithm to achieve this called *antithetic termination*, which directly anticorrelates the walkers' termination events at every timestep (see App. C.2). They provide asymptotic theoretical guarantees, but only for particular choices of graph kernel. In the next section, we will see that our novel method of *optimising* a coupling with data performs much better.

### 4.1 APPROXIMATING AND SOLVING THE OT PROBLEM FOR GRFS

Here, we present our novel approach for formulating and approximately solving the variance reduction OT problem with GRFs. It works by mapping to a corresponding *bipartite matching problem* which we can solve efficiently with linear programming techniques. Fig. 2 gives a visual overview.

**Constructing GRFs from random walks.** To obtain $\phi_{\text{GRF}}(v_i) \in \mathbb{R}^N$, one samples $m$ random walks $\{\boldsymbol{\omega}_k^{(i)}\}_{k=1}^m$ out of node $v_i \in \mathcal{N}$ and averages their 'projections',

$$\phi_{\text{GRF}}(v_i) = \frac{1}{m} \sum_{k=1}^m \psi(\boldsymbol{\omega}_k^{(i)}). \tag{9}$$

The projection function $\psi(\cdot) : \Omega \to \mathbb{R}^N$ maps from the set of graph random walks $\Omega := \left\{ (v_i)_{i=1}^l \mid v_i \in \mathcal{N}, (v_i, v_{i+1}) \in \mathcal{E}, l \in \mathbb{N} \right\}$ to a sparse $N$-dimensional feature vector satisfying $k(v_i, v_j) = \mathbb{E}_{\boldsymbol{\omega}^{(i)}, \boldsymbol{\omega}^{(j)}}[\psi(\boldsymbol{\omega}^{(i)})^\top \psi(\boldsymbol{\omega}^{(j)})]$. $\psi(\cdot)$ depends on the particular kernel being approximated. We direct the reader to the work of Reid et al. (2024b) for an introduction to GRFs, and also provide background in App. C.1. $\psi(\cdot)$ is a complicated function; it is difficult to reason about analytically but straightforward to compute for a particular walk. Moreover, its input walks are discrete random variables so $\psi(\cdot)$ is *not* differentiable with respect to the lengths $l_k^{(i)}$ (where $k = 1, ..., m$ and $v_i \in \mathcal{N}$ is the walker's start node). This precludes straightforward gradient-based optimisation (Sec. 3.2).

**A pair of walkers.** Initially consider just $m = 2$ walkers. The kernel estimator $\widehat{k}(v_i, v_j)$ is

$$\widehat{k}(v_i, v_j) = \phi_{\text{GRF}}(v_i)^\top \phi_{\text{GRF}}(v_j) = \frac{1}{4} \left( \psi(\boldsymbol{\omega}_1^{(i)}) + \psi(\boldsymbol{\omega}_2^{(i)}) \right)^\top \left( \psi(\boldsymbol{\omega}_1^{(j)}) + \psi(\boldsymbol{\omega}_2^{(j)}) \right), \tag{10}$$

which is unbiased provided (i) the marginal distribution of each $\boldsymbol{\omega}_{1,2}^{(i,j)}$ is a simple random walk with geometrically distributed length (hereafter denoted $\eta_{\mathrm{G}}$) and (ii) walks from node $v_i$ are independent from walks from node $v_j$. The variance of the estimator depends on

$$\mathbb{E}\left(\widehat{k}(v_i, v_j)^2\right) = \frac{1}{16}\mathbb{E}_{\boldsymbol{\omega}_{1,2}^{(i,j)}}\left(\left[\left(\psi(\boldsymbol{\omega}_1^{(i)}) + \psi(\boldsymbol{\omega}_2^{(i)})\right)^\top \left(\psi(\boldsymbol{\omega}_1^{(j)}) + \psi(\boldsymbol{\omega}_2^{(j)})\right)\right]^2\right), \quad (11)$$

where the expectation is taken over both the *directions* and *lengths* of the random walks. Suppose the directions remain independent, but the lengths $l_1^{(i)}$ and $l_2^{(i)}$ (and likewise $l_1^{(j)}$ and $l_2^{(j)}$) are to be coupled for variance reduction, analogously to the vector norms in Sec. 3. Let $\mathbb{E}_{\mathrm{dirs}}$ denote the expectation over the walkers' directions. We want to minimise:

$$\mathbb{E}_{(l_1^{(i)}, l_2^{(i)}) \sim \mu, (l_1^{(j)}, l_2^{(j)}) \sim \mu} \mathbb{E}_{\mathrm{dirs}}\left(\left[\left(\psi(\boldsymbol{\omega}_1^{(i)}) + \psi(\boldsymbol{\omega}_2^{(i)})\right)^\top \left(\psi(\boldsymbol{\omega}_1^{(j)}) + \psi(\boldsymbol{\omega}_2^{(j)})\right)\right]^2\right) \text{ for } \mu \in \Lambda_2(\eta_{\mathrm{G}}). \quad (12)$$

**OT, permutation densities and bipartite matchings.** The OT problem in Eq. 12 is analytically intractable. To make progress, we must make approximations. We report **full details** in App. C.3, limiting the main text to a high-level discussion of the main points.

First, to make the objective amenable to Monte Carlo approximation, we move $\mathbb{E}_{\mathrm{dirs}}$ inside the square. This is because, unlike the expression in Eq. 12, $\mathbb{E}_{\mathrm{dirs}}(\psi(\boldsymbol{\omega}_{1,2}^{(i,j)}))$ can be efficiently estimated by simulating random walks. Second, we must optimise amongst the class of couplings $\Lambda_2(\eta_{\mathrm{G}})$, joint distributions of two discrete random variables with geometrically distributed marginals. As in Sec. 3.2, a sensible numerical approach is to limit oneself to a tractable subclass of $\Lambda_2(\eta_{\mathrm{G}})$. Taking inspiration from numerical OT, consider the family of measures $\pi^{(\sigma)}$ on $[0,1]^2$ described by the *permutation densities* $p_\sigma(x, y) := n\mathbb{1}_{\sigma(\lceil nx \rceil) = \lceil ny \rceil}$, with $\sigma$ a permutation of order $n$ (that is, a bijection $\sigma : [\![n]\!] \to [\![n]\!]$). The unit square is split into a $n \times n$ grid where each row and column has a single 'tile' of probability density $n$ and is 0 otherwise (Fig. 2 left). Both marginal distributions of $\pi^{(\sigma)}$ are uniform on $[0,1]$ and are transformed to a probability distribution $\eta$ by pushing forward with the inverse CDF, $F_\eta^{-1}(\cdot) := \inf\{x \in \mathbb{R} : F_\eta(x) \geq \cdot\}$. Transforming both coordinates in this way yields a joint measure $\mu^{(\sigma)} \in \Lambda_2(\eta)$ which will give an unbiased estimator. $n!$ such couplings exist for a given permutation order $n$; we aim to efficiently find the one with the lowest variance.

The permutations $\sigma \in S_n$ can be interpreted as *matchings* between the $n$ quantiles of the geometric distributions over the lengths of a pair of walkers (Fig. 2 centre). With the correct choice of $\sigma$, they can ensure that e.g. if one of the walk lengths is short then the other tends to be long, diversifying the ensemble (Fig. 2 right). Optimising $\sigma$, the approximate OT problem can be written

$$\sigma^* = \arg\min_{\sigma \in S_n} \sum_{q_1 \in [\![n]\!]} \sum_{q_2 \in [\![n]\!]} \left[\left(\widehat{\psi}(q_1^{(i)}) + \widehat{\psi}(\sigma(q_1)^{(i)})\right)^\top \left(\widehat{\psi}(q_2^{(j)}) + \widehat{\psi}(\sigma(q_2)^{(j)})\right)\right]^2 \quad (13)$$

where $\widehat{\psi}(q^{(i)}) := \mathbb{E}_{u \sim \mathcal{U}((\frac{q-1}{n}, \frac{q}{n}])}\left(\mathbb{E}_{\mathrm{dirs}}\psi(F_{\eta_{\mathrm{G}}}^{-1}(u)^{(i)})\right)$. $\mathcal{U}((a, b])$ is the uniform distribution on the interval $(a, b]$. Eq. 13 is a *quadratic assignment problem* (Finke et al., 1987; Burkard et al., 1998). This family is generally NP hard, but in our case the cost function has some convenient extra symmetric structure. In fact, the following is true.

**Theorem 4.1** (Solving Eq. 13). *Given a set of vectors $\{\widehat{\psi}(q^{(i)}), \widehat{\psi}(q^{(j)})\}_{q=1}^n \subset \mathbb{R}^N$, Eq. 13 can be solved with high probability in time complexity independent of $N$. Moreover, in the special case where $i = j$, it can be solved in polynomial time in $n$ (under mild technical assumptions on the set).*

*Proof sketch.* Details and definitions are in App. C.3.1. The time complexity can be made independent of the number of nodes $N$ by performing dimensionality reduction using the celebrated Johnson-Lindenstrauss transformation (Dasgupta et al., 2010), which preserves pairwise dot products with high probability. In the special case $i = j$, Eq. 13 can be rewritten as finding a permutation $\sigma$ that minimises the $L_2$ norm of some particular $N^2$-dimensional vector. In App. C.3.1 we provide a novel algorithm to achieve this efficiently by projecting onto a sequence of random Gaussian vectors, requiring only a mild geometrical condition called $\epsilon$-separation. See Lemma C.2 for full details. $\square$

More pragmatically, one can set $q_1 = q_2$ to simplify to a related *linear assignment problem*, which can be solved efficiently in time $\mathcal{O}(n^3)$ using e.g. the Hungarian algorithm (Kuhn, 1955). We empirically investigate how this final approximation modifies the objective in App. C.3. Taking the optimal permutation and corresponding coupling $\mu^{(\sigma)}$, we define $\sigma$-*coupled GRFs* as follows.

> **Definition 4.2** ($\sigma$-coupled GRFs). *GRFs are $\sigma$-coupled if they are constructed using pairs of random walks with lengths drawn from the coupling $\mu^{(\sigma)}$, with the optimal permutation $\sigma$ obtained by solving a matching problem between the quantiles of the distributions over walk lengths (specifically, Eq. 56 in App. C.3).*

Besides being easy to optimise and sample from, there are also more rigorous OT motivations for the choice of $\sigma$-couplings $\mu^{(\sigma)}$. They relate to the asymptotic behaviour of $\mu^{(\sigma)}$ as the permutation order $n \to \infty$ and the *stability of OT plans* (Villani, 2021). We defer this technical point to App. D. As in Sec. 3.2, another interesting question is whether one could couple the lengths of $m > 2$ walkers. This is challenging and has received little attention in the literature. One possibility would be to combine $m - 1$ permutations, finding a minimum-weights $m$-partite matching with all subgraphs constrained to be complete $K_m$. Another approach would be to approximately solve this multi-marginal OT problem using the Sinkhorn-Knopp algorithm (Sinkhorn and Knopp, 1967; Cuturi, 2013).

**Broader applicability.** As a final remark, the utility of our algorithm extends to graph-based estimators beyond GRFs. For example, it can be used to improve estimates of the *PageRank vector*, a popular measure of the importance of the nodes of a graph proposed by Page et al. (1998) to rank websites in search engine results. $\sigma$-couplings consistently match or beat the previous best algorithm for coupling walk lengths (Reid et al., 2024c). See Fig. 9 in App. E for the full results.

## 4.2 EXPERIMENTS WITH $\sigma$-COUPLED GRFs

We now empirically evaluate $\sigma$-coupled GRFs for variance reduction of graph node kernel estimates. For real-world graphs, we show that lower variance unlocks better approximate inference with scalable graph-based Gaussian processes (GPs), a novel application of GRFs.

**Gram matrix approximation.** GRFs take the *termination probability $p_{\text{halt}}$* as a hyperparameter, setting the rate of decay of the geometric distribution over walk length. A smaller value of $p_{\text{halt}}$ samples longer walks and gives more accurate kernel estimates, but takes longer to run. The optimal coupling depends on $p_{\text{halt}}$. We consider the values $p_{\text{halt}} \in \{0.1, 0.2, 0.3, 0.4, 0.5\}$, finding the optimal permutation $\sigma$ in each case. To find the optimal $\sigma$, we solve the matching problem (App. C.3) for a random Erdős-Rényi graph with $N = 100$ nodes, taking a permutation order $n = 30$ and choosing the 2-regularised Laplacian kernel as our target. We use the Hungarian algorithm, averaging the cost matrix over every possible node pair $(v_i, v_j) \in \mathcal{N}^2$. Having computed couplings $\mu^{(\sigma)}$ for each $p_{\text{halt}}$, we then test the corresponding $\sigma$-coupled GRFs on a variety of real-world graphs. Fig. 3 shows the results for `cora` ($N = 2708$), with the rest left to App. F.1. We plot the relative Frobenius norm error of the Gram matrix approximation $\|\mathbf{K} - \widehat{\mathbf{K}}\|_{\text{F}}/\|\mathbf{K}\|_{\text{F}}$ with walkers that are i.i.d., antithetic (Reid et al., 2024c) or $\sigma$-coupled. For each $p_{\text{halt}}$, $\sigma$-coupled GRFs give equally good or smaller kernel estimator errors. Our OT approach outperforms antithetic termination: a data-independent, hard-coded algorithm designed specifically to improve GRFs (Reid et al., 2024c). We include visualisations of the optimal permutations for

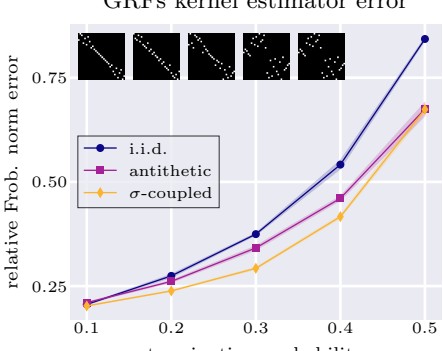

Figure 3: Kernel estimator error vs termination probability. Insets show permutations for $p_{\text{halt}} \in \{0.1, 0.2, 0.3, 0.4, 0.5\}$.

different values of $p_{\text{halt}}$ in the inset, verifying that the $\sigma$-coupling adapts to different hyperparamaters.

**Novel application: $\sigma$-coupled GRFs for scalable graph-based GPs.** We now apply $\sigma$-coupled GRFs to scalable graph-based Gaussian processes[1] (GPs), where improved estimation of the covariance function permits better approximate inference. Scalable GPs are a novel application of GRFs that may be of independent interest (Borovitskiy et al., 2021; Mostowsky et al., 2024).

Consider the task of *probabilistic graph interpolation*. This aims to predict unknown graph function values, along with principled uncertainty estimates, from an observed set (Pfaff et al., 2020). Take

---

[1]This may be better described by Gaussian *distributions* than Gaussian *processes* since for fixed $\mathcal{G}$ we have a finite number of random variables, but we use the latter for consistency with recent literature.

mesh graphs $\mathcal{G}$ where every node $v_i \in \mathcal{N}$ has a normal vector $\boldsymbol{n}_i \in \mathbb{R}^3$ (Dawson-Haggerty, 2023). Our task is to predict the $z$-components of a masked set, $\{(\boldsymbol{n}_i)_z\}_{i=1}^{N_{\text{test}}}$ with $N_{\text{test}} = \lfloor 0.05N \rfloor$. To achieve this, we use a graph-based GP with a heat kernel covariance function. We compute a sparse, unbiased approximation of this kernel using GRFs with $\{16, 32, 64\}$ walkers that are i.i.d., antithetic (Reid et al., 2024c) or $\sigma$-coupled. Details of GP hyperparameter optimisation are given in App. F.2. Fig. 4 shows the results. For mesh graphs of different sizes (the largest as big as 8700 nodes), we plot the relative Frobenius norm error of the Gram matrix approximation, the test root mean square error (RMSE), and the KL divergence to the true posterior. Our variance reduction method unlocks more accurate predictions and better uncertainty quantification, sometimes by a factor of $> 2$.

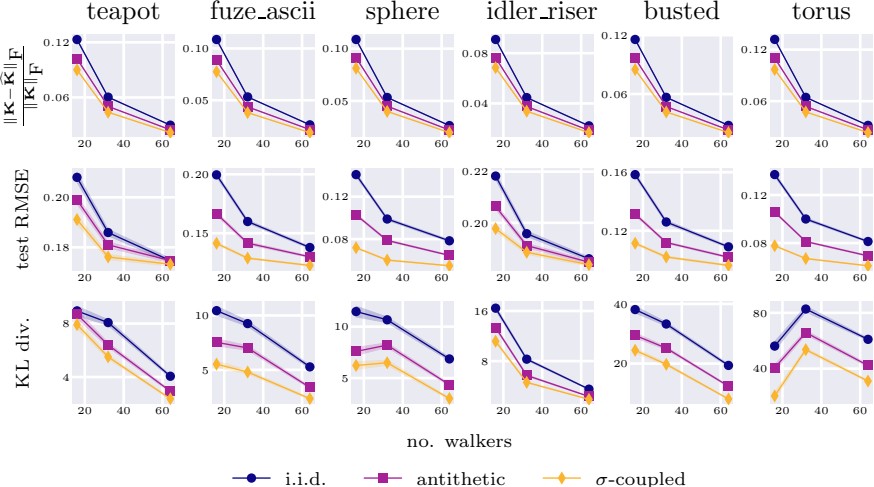

Figure 4: Graph GP regression. Rows show the kernel estimation error, test set RMSE, and KL-divergence between the true and approximate posterior. Lower is better. $\sigma$-coupled GRFs give better predictions and uncertainty estimates, sometimes by a factor of $> 2$. Standard errors are shaded.

**Probabilistic interpolation of traffic data.** We also train a scalable graph-based GP on a traffic flow dataset of the highways of San Jose, California, curated by Borovitskiy et al. (2021) using data from Chen et al. (2001) and OpenStreetMap. The graph has 1016 nodes, with speed only known at 325. We use 250 randomly-chosen nodes as training data and the remainder as test data. With this sparse, noisy dataset, $\sigma$-coupled GRFs again give substantially better predictions and uncertainty estimates. Fig. 11 in App. F.3 shows the full results.

## 5 DISCUSSION AND OUTLOOK

OT provides a powerful, unifying paradigm for variance reduction with random features. It offers perspectives, proof techniques and numerical algorithms for finding novel RF couplings on continuous and discrete input domains, **substantially beating previous algorithms in both settings and with disparate basis functions**.

**Variance reduction is *not* all you need.** Whilst the presence of variance reduction is unambiguous, downstream benefits tell a more nuanced story. With GRFs for scalable GPs, variance reduction permits much better approximate inference (Sec. 4.2). With RFFs and RLFs, this is not the case. For instance, when approximating attention in Performers (Choromanski et al., 2020), *maximising* the pointwise kernel estimator variance – the 'wrong' OT problem – turns out to improve predictive performance after row normalisation. This shows that, though popular, naive variance reduction is not always the right goal.

**Right framing, wrong cost function.** Therefore, we posit that OT provides the *right framing* for the problem of coupling RFs, but sometimes pointwise kernel variance is the *wrong cost function*. This choice may not fully capture how the *joint* distribution over kernel estimates determines downstream performance. Coupling to optimise e.g. the spectral properties of $\widehat{\mathbf{K}}$ (Choromanski et al., 2018; Avron et al., 2017a) or the variance of row-normalised attention scores may prove better. These objectives are rarely considered in the literature. Fortunately, OT provides a suite of theoretical and numerical tools achieve this; one simply modifies the cost function in Eq. 2, optimising a different characteristic of the coupling. We hope this research will spur future work in this exciting direction.

# 6 CONTRIBUTIONS AND ACKNOWLEDGEMENTS

**Relative contributions.** IR conceptualised the project, proposed the coupling mechanisms in Defs 3.3 and 4.2, proved the major theoretical contributions, ran the GRF experiments and wrote the manuscript. SM designed and ran the RFF and RLF GP experiments (Sec. 3.3), helped shape the project's direction and made core contributions to the text. KC acted as the senior lead, providing technical guidance and developing the algorithms for the matching problem in App. C.3.1. RET met frequently throughout the project, giving important advice and support. AW provided helpful discussion, supervision and feedback on the manuscript.

IR acknowledges support from a Trinity College External Studentship and a Google PhD Fellowship. Part of the work was completed as a student researcher at Google. SM acknowledges funding from the Vice Chancellor's and the George and Marie Vergottis scholarship of the Cambridge Trust, and the Qualcomm Innovation Fellowship. RET is supported by the EPSRC Probabilistic AI Hub (EP/Y028783/1). AW acknowledges support from a Turing AI fellowship under grant EP/V025279/1 and the Leverhulme Trust via CFI.

We thank Bruno Mlodozeniec for his suggestion to use copulas in Sec. 3.2 and Mark Rowland for insightful discussions about multi-marginal optimal transport and the limitations of pointwise variance reduction. Viacheslav Borovitskiy helped guide our discussion of scalable graph-based GPs and, together with Iskander Azangulov, kindly provided updated code for loading the traffic data graph in Sec. 4.2 and App. F.3. We thank Matt Ashman and Arijit Sehanobish for their thoughtful feedback on the text, and Jihao Andreas Lin for interesting suggestions about possible GP applications.

# 7 ETHICS AND REPRODUCIBILITY

**Ethics statement**: Our work is foundational with no immediate ethical concerns apparent to us. However, increases in scalability provided by improvements to MC algorithms could exacerbate existing and incipient risks of machine learning, from bad actors or as unintended consequences.

**Reproducibility statement**: Every effort has been made to ensure the work's reproducibility. The core algorithms are presented in Defs 3.3 and 4.2, with exhaustive details and discussion in the Appendices. Theoretical results are proved with full assumptions in Apps A and C.3.1, with proof sketches included in the main text for clarity. All datasets are available online. We give links to suitable repositories in every instance. Where possible, results are reported with uncertainties to facilitate comparison. Code is available at: `https://github.com/cambridge-mlg/learnable-qmc`.

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

# A   SOLVING THE OT PROBLEM FOR RFFS AND RLFS

In this appendix, we provide proofs for the theoretical results in Sec. 3.1 and supplement the discussion of copula-based numerical OT solvers in Sec. 3.2.

## A.1   PROOF OF LEMMA 3.1

We begin by proving Lemma 3.1, which formulates variance reduction for RFFs and RLFs as an optimal transport problem. We will first reason about the simpler case of RLFs, then consider RFFs.

*Proof of Lemma 3.1.* Consider the following recently-derived result by Reid et al. (2023).

**Lemma A.1** (Kernel estimator MSE for RLFs (Reid et al., 2023))**.** *When estimating the Gaussian kernel* $k(\boldsymbol{x}, \boldsymbol{y}) := \exp(-\|\boldsymbol{x} - \boldsymbol{y}\|_2^2/2)$ *for datapoints* $\{\boldsymbol{x}, \boldsymbol{y}\} \subset \mathbb{R}^d$ *using random Laplace features (synonymously, positive random features), the mean square error of the kernel estimate* $\widehat{k}(\boldsymbol{x}, \boldsymbol{y})$ *is given by:*

$$MSE(\widehat{k}) = \frac{e^{-2x^2 - 2y^2}}{m} \left( (e^{2v^2} - e^{v^2}) + (m-1)(\rho(\boldsymbol{x}, \boldsymbol{y}) - e^{v^2}) \right) \tag{14}$$

*where* $m$ *is the number of sampled random frequencies,* $v := \|\boldsymbol{x}_i + \boldsymbol{x}_j\|_2$ *is a data-dependent scalar and* $\rho(\boldsymbol{x}_i, \boldsymbol{x}_j)$ *is the* RF-conformity,

$$\rho(\boldsymbol{x}, \boldsymbol{y}) := \frac{\Gamma(\frac{d}{2})}{m(m-1)} \sum_{i,j \neq i} \mathbb{E}_{\omega_{ij}} \left( \sum_{k=0}^{\infty} \frac{v^{2k} \omega_{ij}^{2k}}{2^{2k} k! \Gamma(k + \frac{d}{2})} \right). \tag{15}$$

*Here,* $\omega_{ij} := \|\boldsymbol{\omega}_i + \boldsymbol{\omega}_j\|_2$ *is the norm of the resultant of a pair of distinct frequencies and* $\Gamma$ *is the Gamma function.*

*Proof.* Reid et al. (2023). □

The simple derivation, reported in full by Reid et al. (2023), is based on rewriting the angular integral as a Hankel transform, yielding a Bessel function of the first kind with a known Taylor expansion.

Supposing that $\boldsymbol{\omega}_i \perp \boldsymbol{\omega}_j$, we have that $\omega_{ij} = \sqrt{\omega_i^2 + \omega_j^2}$, where $\omega_i := \|\boldsymbol{\omega}_i\|_2$ denotes the $L_2$-norm of the frequency $\boldsymbol{\omega}_i$. Note that, for $\boldsymbol{\omega}_i$ to be marginally Gaussian, we require that the marginal distribution over its *norm* is $\chi_d$, a Chi distribution with $d$ degrees of freedom. Substituting this into Eq. 14 and dropping terms unmodified by the coupling and irrelevant multiplicative factors, it is straightforward to arrive at the expression for the cost function $c_{\text{RLF}}$ in Eq. 6.

Finding the cost function for RFFs is only slightly more difficult. We begin by citing another recent result by Reid et al. (2023).

**Lemma A.2** (Kernel estimator MSE for RFFs (Reid et al., 2023))**.** *When estimating the Gaussian kernel* $k(\boldsymbol{x}, \boldsymbol{y}) := \exp(-\|\boldsymbol{x} - \boldsymbol{y}\|_2^2/2)$ *for datapoints* $\{\boldsymbol{x}, \boldsymbol{y}\} \subset \mathbb{R}^d$ *using random Fourier features, the mean square error of the kernel estimate* $\widehat{k}(\boldsymbol{x}, \boldsymbol{y})$ *is given by:*

$$MSE(\widehat{k}) = \frac{1}{m} \left( \frac{(1 - e^{-z^2})^2}{2} + (m-1)(\zeta(\boldsymbol{x}, \boldsymbol{y}) - e^{-z^2}) \right) \tag{16}$$

*where* $m$ *is the number of samples,* $\boldsymbol{z} := \boldsymbol{x} - \boldsymbol{y}$ *and* $\zeta(\boldsymbol{x}, \boldsymbol{y})$ *is defined by*

$$\zeta(\boldsymbol{x}, \boldsymbol{y}) := \frac{1}{m(m-1)} \sum_{i,j \neq i} \mathbb{E}_{\boldsymbol{\omega}_i, \boldsymbol{\omega}_j} \left[ \cos(\boldsymbol{\omega}_i^\top \boldsymbol{z}) \cos(\boldsymbol{\omega}_j^\top \boldsymbol{z}) \right]. \tag{17}$$

*Proof.* Reid et al. (2023). □

Note the close resemblance to Eq. 15. The only term that depends on couplings between the random frequencies $\{\boldsymbol{\omega}_i\}_{i=1}^m$ is $\zeta(\boldsymbol{x}, \boldsymbol{y})$, which we seek to suppress with carefully engineered correlations. From elementary trigonometry, $\cos \boldsymbol{\omega}_i^\top \boldsymbol{z} \cos \boldsymbol{\omega}_j^\top \boldsymbol{z} = \frac{1}{2}(\cos((\boldsymbol{\omega}_i + \boldsymbol{\omega}_j)^\top \boldsymbol{z}) + \cos((\boldsymbol{\omega}_i - \boldsymbol{\omega}_j)^\top \boldsymbol{z}))$, and for any coupling scheme $\boldsymbol{\omega}_i \pm \boldsymbol{\omega}_j$ is isotropic. Defining the random variables $\omega^{(+)} := \|\boldsymbol{\omega}_i + \boldsymbol{\omega}_j\|_2$ and $\omega^{(-)} := \|\boldsymbol{\omega}_i - \boldsymbol{\omega}_j\|_2$ and integrating out the angular part,

$$\zeta(\boldsymbol{x}_i, \boldsymbol{x}_j) = \frac{1}{m(m-1)} \sum_{i,j \neq i} \Gamma(d/2) 2^{\frac{d}{2}-2} \mathbb{E}_{\omega^{(\pm)}} \left[ (\omega^{(+)} z)^{1-\frac{d}{2}} J_{\frac{d}{2}-1}(\omega^{(+)} z) + (\omega^{(+)} z)^{1-\frac{d}{2}} J_{\frac{d}{2}-1}(\omega^{(-)} z) \right]$$

$$\tag{18}$$

where $J_\alpha(x)$ is a Bessel function of the first kind, order $\alpha$. If $\boldsymbol{\omega}_i \perp \boldsymbol{\omega}_j$, $\omega^{(+)} = \omega^{(-)}$ so their distributions are identical. It follows that we can write

$$\zeta(\boldsymbol{x}, \boldsymbol{y}) = \frac{\Gamma(\frac{d}{2})}{m(m-1)} \sum_{i, j \neq i} \mathbb{E}_{\omega_i, \omega_j} \left[ \sum_{k=0}^{\infty} \frac{(-1)^k z^{2k} \left(\omega_i^2 + \omega_j^2\right)^k}{2^{2k} k! \Gamma(k + \frac{d}{2})} \right]. \tag{19}$$

Again dropping multiplicative factors and terms unmodified by the coupling, we arrive at the RFF OT cost function $c_{\text{RFF}}$ specified in Eq. 6. This completes the derivation. $\qquad\square$

### A.2 PROOF OF THM. 3.2

We now solve the OT problem formulated in Lemma 3.1 *exactly* in the special case that $m = 2$ (with mild asymptotic assumptions for RFFs). For the reader's convenience, we copy it from the main text below.

**Theorem A.3** (Solution to OT problem when $m = 2$). *Denote by $F_{\chi_d}(\cdot)$ the cumulative distribution function (CDF) of $\chi_d$. Consider $m = 2$ orthogonal frequencies with norms $(\omega_1, \omega_2)$. For RLFs, the OT problem in Eq. 5 is solved by the* negative monotone *coupling*

$$F_{\chi_d}(\omega_1) + F_{\chi_d}(\omega_2) = 1. \tag{20}$$

*For RFFs, Eq. 7 ensures lower cost than any other coupling, provided $z$ is sufficiently small.*

The proof of Thm. 3.2 uses ideas from optimal transport theory. In particular, it modifies arguments made for a related problem by (among others) Thorpe (2019), to which we direct the interested reader for further context and discussion. Before giving the proof, we establish some basic definitions and results.

**Definition A.4** (*c*-monotone sets). *Given a cost function $c : \mathbb{R}^d \times \mathbb{R}^d \to \mathbb{R}$, we refer to a set $\Gamma \in \mathbb{R}^2$ as* $c$-monotone *if for all pairs $(x_1, y_1), (x_2, y_2) \in \Gamma$ we have that*

$$c(x_1, y_1) + c(x_2, y_2) \leq c(x_1, y_2) + c(x_2, y_1). \tag{21}$$

It is intuitive that $c$-monotonicity should be a property of the support of Kantorovich optimal transport plan: if we could have accessed lower cost by sending $x_1 \to y_2$ and $x_2 \to y_1$ instead of $x_1 \to y_1$ and $x_2 \to y_2$, the plan would have done this instead. This is formalised as follows.

**Lemma A.5** (Support of optimal transport plan is $c$-monotone (Thorpe, 2019)). *Consider $\eta \in \mathcal{P}(\mathbb{R})$, and assume that $\mu^* \in \Lambda(\eta)$ is the Kantorovich optimal transport plan for a continuous cost function $c(x, y)$. Then for all $(x_1, y_1), (x_2, y_2) \in \text{supp}(\mu^*)$ we have that*

$$c(x_1, y_1) + c(x_2, y_2) \leq c(x_1, y_2) + c(x_2, y_1). \tag{22}$$

*Proof.* Thorpe (2019). $\qquad\square$

We are now ready to provide our proof of Thm. 3.2.

*Proof of Thm. 3.2.* Inspecting the cost functions in Eq. 6, it is clear that in the special case $m = 2$ we have that

$$c_{\text{RFF}}(\omega_1, \omega_2) = \sum_{k=0}^{\infty} \frac{(-1)^k z^{2k} \left(\omega_1^2 + \omega_2^2\right)^k}{2^{2k} k! \Gamma(k + \frac{d}{2})}, \quad c_{\text{RLF}}(\omega_1, \omega_2) = \sum_{k=0}^{\infty} \frac{v^{2k} (\omega_1^2 + \omega_2^2)^k}{2^{2k} k! \Gamma(k + \frac{d}{2})}, \tag{23}$$

with $z := \|\boldsymbol{x} - \boldsymbol{y}\|_2$ and $v := \|\boldsymbol{x} + \boldsymbol{y}\|_2$ as usual.

First consider $c_{\text{RLF}}$. We claim that for any $x_1, x_2, y_1, y_2 \geq 0$ satisfying Eq. 22 for this $c_{\text{RLF}}$, $x_1 < x_2$ implies $y_1 \geq y_2$. This is seen by observing that

$$c_{\text{RLF}}(x_1, y_1) + c_{\text{RLF}}(x_2, y_2) - c_{\text{RLF}}(x_1, y_2) - c_{\text{RLF}}(x_2, y_1)$$

$$= \sum_{k=0}^{\infty} \frac{v^{2k}}{2^{2k} k! \Gamma(k + \frac{d}{2})} \left[ (x_1^2 + y_1^2)^k + (x_2^2 + y_2^2)^k - (x_1^2 + y_2^2)^k - (x_2^2 + y_1^2)^k \right]$$

$$= \sum_{k=0}^{\infty} \frac{v^{2k}}{2^{2k} k! \Gamma(k + \frac{d}{2})} \sum_{i=0}^{k} \binom{k}{i} \left[ (y_2^2)^{k-i} - (y_1^2)^{k-i} \right] \left[ (x_2^2)^i - (x_1^2)^i \right]. \tag{24}$$

Supposing that $x_1 < x_2$, Eq. 22 is satisfied if and only if $y_1 \geq y_2$, verifying the statement.

Denote by $\Gamma_{\mathrm{RLF}} := \mathrm{supp}(\mu^*)$ the support of the optimal transport plan for the cost function $c_{\mathrm{RLF}}$, and consider some point $(x_0, y_0) \in \Gamma$. An immediate implication of the statement above is that

$$\Gamma_{\mathrm{RLF}} \subset \{(x,y) : x \leq x_0, y \geq y_0\} \cup \{(x,y) : x \geq x_0, y \leq y_0\}. \tag{25}$$

Let $A = [0, x_0] \times [y_0, \infty)$, $B = [0, x_0) \times [0, y_0)$, $C = [x_0, \infty) \times [0, y_0]$ and $D = (x_0, \infty] \times (y_0, \infty]$. Note that $A \cup B \cup C \cup D = \mathbb{R}^+ \times \mathbb{R}^+$ so $\mu^*(A \cup B \cup C \cup D) = 1$ since the measure is normalised. The subsets are also disjoint apart from $A \cap C = (x_0, y_0)$, a singleton of zero measure.[2] Eq. 25 implies that $\mu^*(B) = 0 = \mu^*(D)$, whereupon $1 = \mu^*(A \cup C) = \mu^*(A) + \mu^*(C) = \mu^*(A \cup B) + \mu^*(C \cup B)$. Now $A \cup B = ([0, x_0] \times \mathbb{R}^+) \setminus (\{x_0\} \times [0, y_0))$ and the set $\{x_0\} \times [0, y_0)$ is zero measure. Therefore, $\mu^*(A \cup B) = \mu^*([0, x_0] \times \mathbb{R}^+) = F_{\chi_d}(x_0)$. Likewise, $\mu^*(C \cup B) = \mu^*(\mathbb{R}^+ \times [0, y_0]) = F_{\chi_d}(y_0)$. Relabelling $(x_0, y_0) \in \Gamma$ by $(\omega_1, \omega_2)$, Eq. 20 immediately follows. This completes the proof that negative monotone coupling minimises the kernel estimator variance for orthogonal RLFs.

Let us now turn to the case of RFFs. The optimal transport plan $\mu^*$ will instead be $c_{\mathrm{RFF}}$-monotone. Unfortunately, Eq. 24 does *not* hold in general for $c_{\mathrm{RFF}}$, so we cannot immediately use the same arguments to conclude that the OT plan is negative monotone. However, it *does* hold if we just consider the first few terms of its Taylor expansion in $z$. The following is true.

**Lemma A.6** (Negative monotone coupling for RFFs). *Denote by $\mu_{NM}$ the negative monotone coupling. Consider a coupling $\mu' \in \Lambda_2(\eta) \setminus \{\mu_{NM}\}$, i.e. any other feasible transport map which is not negative monotone. There exists some constant $\delta(\mu') > 0$ such that $\mathcal{I}(\mu_{NM}) < \mathcal{I}(\mu')$ for all $z < \delta$ (where $\mathcal{I}(\mu)$ denotes the expectation of the cost function under $\mu$).*

*Proof.* Recall that, for RFFs, the cost function is of the form

$$c_{\mathrm{RFF}}(\omega_1, \omega_2) = \sum_{k=0}^{\infty} \frac{(-1)^k z^{2k} (\omega_1^2 + \omega_2^2)^k}{2^{2k} k! \Gamma(k + \frac{d}{2})}, \tag{26}$$

and that we would like to solve $\mu^* = \arg\min_{\mu \in \Lambda_2(\eta)} \left[ \mathbb{E}_{\omega_{1,2} \sim \mu} c_{\mathrm{RFF}}(\omega_1, \omega_2) \right]$. In general, $\mu^* = \mu^*(z, d)$. This is a very challenging OT problem; we do not believe that a simple closed-form solution exists. However, we can make progress expanding in $k$. The $k = 0$ and $k = 1$ terms are trivial since they do not contain $\omega_{1,2}$ cross terms. We have

$$\mu^* = \arg\min_{\mu \in \Lambda_2(\eta)} \left[ \frac{z^4}{32 \Gamma(2 + \frac{d}{2})} \mathbb{E}\left((\omega_1^2 + \omega_2^2)^2\right) + \sum_{k=3}^{\infty} \frac{(-1)^k z^{2k}}{2^{2k} k! \Gamma(k + \frac{d}{2})} \mathbb{E}\left((\omega_1^2 + \omega_2^2)^k\right) \right]. \tag{27}$$

We can solve the OT problem exactly for the first term, recovering negative monotone coupling. That is,

$$\arg\min_{\mu \in \Lambda_2(\eta)} \left[ \frac{z^4}{32 \Gamma(2 + \frac{d}{2})} \mathbb{E}\left((\omega_1^2 + \omega_2^2)^2\right) \right] = \mu_{\mathrm{NM}} \tag{28}$$

for any $z > 0$ and $d$ finite. Note also that

$$\max_{\mu \in \Lambda_2(\eta)} \mathbb{E}\left((\omega_1^2 + \omega_2^2)^k\right) = 2^k \mathbb{E}\left(\omega^{2k}\right) \tag{29}$$

because the coupling that *maximises* the expectation is positive monotone (this follows from our previous arguments almost automatically). This is evaluated as the $k$th moment of a $\chi_d^2$ distribution, which is nothing other than $2^k \Gamma(\frac{k}{2} + d) / \Gamma(\frac{k}{2})$. This means we can upper bound the magnitude of the $k$th term in the expansion by: $z^{2k} / \left(k! \Gamma(\frac{k}{2})\right)$. So for *any* coupling (feasible transport plan) we can upper bound the magnitude of the sum on the right by: $g(z) := \sum_{k=3}^{\infty} z^{2k} / \left(k! \Gamma(\frac{k}{2})\right)$. This goes to 0 as $z \to 0$ and increases monotonically.

Consider some coupling $\mu' \in \Lambda_2(\eta) \setminus \{\mu_{\mathrm{NM}}\}$. Since the minimiser is unique (if $\mu'$ is not negative monotone, it is not a minimiser), there is a positive constant $c$ such that

$$c := \mathbb{E}_{\mu'} \left(\omega_1^2 + \omega_2^2\right)^2 - \mathbb{E}_{\mu_{\mathrm{NM}}} \left(\omega_1^2 + \omega_2^2\right)^2 > 0. \tag{30}$$

---

[2] e.g. since $(x_0, y_0) \subset \{x_0\} \times \mathbb{R}^+$ and $\mu^+(\{x_0\} \times \mathbb{R}^+) = p_\chi(x = x_0) = 0$ since the marginal measure $\chi$ is nonatomic.

But we also have that

$$\left| \mathbb{E}_{\mu_{\mathrm{NM}}} \sum_{k=3}^{\infty} \frac{(-1)^k z^{2k} \left(\omega_1^2 + \omega_2^2\right)^k}{2^{2k} k! \Gamma(k + \frac{d}{2})} - \mathbb{E}_{\mu^*} \sum_{k=3}^{\infty} \frac{(-1)^k z^{2k} \left(\omega_1^2 + \omega_2^2\right)^k}{2^{2k} k! \Gamma(k + \frac{d}{2})} \right| < 2g(z). \tag{31}$$

Therefore, to guarantee that $\mathcal{I}(\mu_{\mathrm{NM}}) < \mathcal{I}(\mu')$, it is sufficient that

$$\frac{z^4}{32\Gamma(2 + \frac{d}{2})} c > 2g(z). \tag{32}$$

Since $g(z)/z^4$ is also monotonically increasing and evaluates to $0$ and $z = 0$ (by inspecting its Taylor expansion), it is always possible to solve this to find $z = \delta(\mu')$ such that, for $z < \delta(\mu')$, $\mathcal{I}(\mu_{\mathrm{NM}}) < \mathcal{I}(\mu')$ is guaranteed. This holds for any dimensionality $d$ and any measure $\mu'$ that is not negative monotone. $\square$

This shows that the negative monotone coupling gives lower expected cost with RFFs than any other coupling if $z$ is sufficiently small, concluding our proof of Thm. 3.2. $\square$

**Comments on small $z$ for RFFs.** We have seen that for RFFs it is only possible to prove optimality of $\mu_{\mathrm{NM}}$ when $z$ is small enough. Here, we comment on why this result is nonetheless interesting and useful. Firstly, note that in practice pairwise norm coupling still substantially suppresses kernel estimator variance, even at data lengthscales chosen by training an exact GP independent of the RF construction (Table 1). Even if at bigger $z$ the method no longer provides the smallest possible estimator variance, it can still substantially reduce it compared to an i.i.d. coupling. Second, from a more theoretical perspective, the large $d$, small sample regime is exactly where standard QMC methods often fail. It is interesting that OT-driven methods can still provide theoretical guarantees in this low-sample, high-dimensionality setting. Last, we note that, for RFF variance reduction schemes, it is very common to only guarantee gains in the asymptotic limit. This is also the case e.g. for orthogonality (Reid et al., 2023; Yu et al., 2016): a well-established and widely-used algorithm.

### A.3 PROOF OF COROLLARY 3.4

We now prove that pairwise norm-coupled RFs (Def. 3.3) provide strictly lower kernel estimator variance than i.i.d. RFs.

*Proof of Corollary 3.4.* Supposing that we have $m = d$ frequencies, the sum $\sum_{i,j \neq i}^{d}$ has $d(d-1)$ terms in total. Of these, $2\lfloor \frac{d}{2} \rfloor$ correspond are negative monotone norm couplings, and the remainder are independent. The independent terms are the same in the pairwise norm-coupled and fully i.i.d. configurations, so can be ignored. By Thm. 3.2, we have seen that negative monotone coupling exactly solves the variance reduction OT problem for RLFs, so these variance contributions will be strictly smaller in the norm-coupled case. It immediately follows that pairwise norm-coupled RLFs give strictly lower kernel estimator variance than orthogonal independent-norm RLFs. For RFFs, Thm. A.6 shows that negative monotone coupling is better than i.i.d. if $z$ is small enough, so the result again follows. $\square$

### A.4 PROOF OF THM. 3.5

We now drop the restriction that $\widehat{\omega}_1 \perp \widehat{\omega}_2$ and consider the variance reduction problem for $m = 2$ frequencies whose respective direction is unconstrained. We will prove Thm. 3.5, which asserts that in this case the best possible coupling is antithetic, $\omega_1 = -\omega_2$.

*Proof of Thm. 3.5.* Recalling the expression for RLF variance in Lemma A.1, the more general OT problem under consideration is

$$\mu^* = \arg\min_{\mu \in \Lambda_2(\mathcal{N})} \left[ \mathbb{E}_{\omega_1, \omega_2 \sim \mu} \sum_{k=0}^{\infty} \frac{v^{2k} \|\omega_1 + \omega_2\|_2^k}{2^{2k} k! \Gamma(k + \frac{d}{2})} \right]. \tag{33}$$

The term in square parentheses is an expectation of an infinite sum, every term of which is greater than or equal to $0$. The sum is manifestly minimised if $\|\omega_1 + \omega_2\|_2 = 0$, which sets every term (apart from the first) to $0$. This is achieved if and only if $\omega_1 = -\omega_2$: a valid coupling called 'antithetic sampling'. Any other joint distribution assigns nonzero probability to the event $\|\omega_1 + \omega_2\|_2 > 0$, so this optimal coupling is unique. $\square$

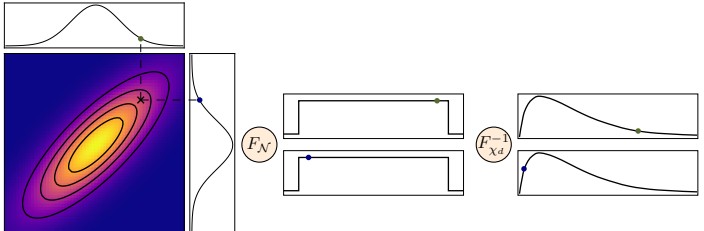

Figure 5: Copula schematic for $d = 2$. Random variables are drawn from a Gaussian distribution with correlation matrix $\boldsymbol{\Sigma}$. They are pushed forward using $F_{\mathcal{N}}$ then $F_{\chi_d}^{-1}$ to obtain coupled variables with marginal $\chi_d$ distributions. $\boldsymbol{\Sigma}$ is learned using gradient-based optimisation, approximately solving the multi-marginal OT problem in Eq. 5.

### A.5 COPULAS AS NUMERICAL OT SOLVERS

In the main text, we noted that finding an analytic solution to the multi-marginal OT problem for RFFs and RLFs (Eq. 5) is an open problem. In Sec. 3.2, we briefly presented an alternative numerical approach using *copulas*. Here, we discuss this in greater detail.

**Copulas.** A copula is a multivariate cumulative distribution function (CDF) whose marginals are uniformly distributed on $[0, 1]$. By Sklar's theorem (Sklar, 1959), its joint distribution can be arbitrary. Given a copula, we can easily enforce the constraint that its marginals are $\chi_d$ by pushing each coordinate forward with the inverse CDF $F_{\chi_d}^{-1}(\cdot)$, whilst retaining necessary flexibility in the joint to reduce estimator variance. Fig. 5 gives a schematic. Copulas can be used to model dependencies between random variables and are popular tools in quantitative finance (Haugh, 2016).

**Gaussian copulas.** The general family of copulas is still intractable to optimise and sample from so we constrain ourselves to *Gaussian copulas*. These are distributions with uniform marginals whose joint distributions are determined by multivariate Gaussians, defined below.

**Definition A.7** (Gaussian copula). *Let $\{g_i\}_{i=1}^m \sim \mathcal{N}(0, \boldsymbol{\Sigma})$ where $\boldsymbol{\Sigma} \in \mathbb{R}^{m \times m}$ is a correlation matrix, i.e. a positive definite matrix with unit diagonals, and let $F_{\mathcal{N}}$ be the CDF of the standard univariate Gaussian. We say $\{u_i\}_{i=1}^m$, where $u_i \coloneqq F_{\mathcal{N}}(g_i)$, is distributed according to a Gaussian copula with covariance $\boldsymbol{\Sigma}$. We use the notation $\{u_i\}_{i=1}^m \sim GC(\boldsymbol{\Sigma})$ to denote this.*

**Parameterising correlation matrices.** Gaussian copulas are easy to sample from since they involve sampling a multivariate Gaussian and applying the univariate Gaussian CDF. We are therefore left with the task of finding an appropriate correlation matrix $\boldsymbol{\Sigma}$, for which we turn to numerical optimisation. The family of $m \times m$ correlation matrices can be parameterised by a vector $\boldsymbol{\theta} \in \mathbb{R}^{m(m-1)/2}$. In fact, there exist tractable bijections between unconstrained vectors of real numbers $\boldsymbol{\theta} \in \mathbb{R}^{m(m-1)/2}$ and lower triangular Cholesky factors $\mathbf{L}_{\boldsymbol{\theta}}$ such that $\boldsymbol{\Sigma} = \mathbf{L}_{\boldsymbol{\theta}} \mathbf{L}_{\boldsymbol{\theta}}^\top$ is a valid correlation matrix (Bhat and Mondal, 2021). In particular, suppose that for each $i = 1, \ldots, N$, and $j = 1, \ldots, i$, we have $\theta_{ij} \in \mathbb{R}^+$, where $\theta_{ii} = 1$. Then the parameterisation we use is

$$L_{ij} = \begin{cases} \frac{\theta_{ij}}{s_i} & \text{for } i \leq j, \\ 0 & \text{otherwise,} \end{cases} \tag{34}$$

where $s_i = \sqrt{\sum_{j=1}^i \theta_{ij}^2}$. Note that, since we are directly parameterising the Cholesky factor, we can sample from the associated Gaussian copula with $\mathcal{O}(m^2)$ computational cost.

**Optimising correlation matrices.** In order to pick an appropriate correlation matrix $\boldsymbol{\Sigma}$, we optimise it directly to minimise the root mean squared error (RMSE) loss

$$\mathcal{L}(\boldsymbol{\theta}) = \mathbb{E}_{\{u_i\}} \left[ \sqrt{\frac{1}{N^2} \sum_{i,j=1}^N (\phi_{\text{RF}}(\boldsymbol{x}_i)^\top \phi_{\text{RF}}(\boldsymbol{x}_j) - k(\boldsymbol{x}_i, \boldsymbol{x}_j))^2} \right], \tag{35}$$

where $\{u_i\} \sim \text{GC}(\mathbf{L}_{\boldsymbol{\theta}} \mathbf{L}_{\boldsymbol{\theta}}^\top)$. Note that $\phi_{\text{RF}}$ here depends on $u_i$ by pushing forward with $F_{\chi_d}^{-1}$ and using the result as random frequency norms, though we have suppressed this dependence for notational

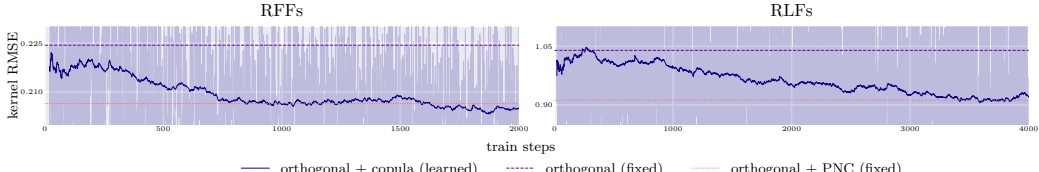

Figure 6: Training RMSE losses on a split of the Boston dataset. For the 'orthogonal + copula (learned)' configuration, we display the raw training loss (light blue) as well as an exponential moving average (dark blue). The performance of the coupling scheme returned by optimising the copula matches the performance of the 'orthogonal + PNC' scheme that we proved is optimal for the $m = 2$ case. The optimisation is very noisy but this can be mitigated by increasing the number of samples used in the reparameterisation trick.

simplicity. Assuming that $\phi_{\mathrm{RF}}$ is differentiable with respect to $\omega_i$, which is the case in RFFs and RLFs, we can optimise the copula parameters $\boldsymbol{\theta}$ by estimating the loss in Eq. 35, computing its gradients with respect to $\boldsymbol{\theta}$, and updating its values accordingly.

**Training curves.** Fig. 6 shows an example of training curves for RFFs and RLFs using a numerically optimised Gaussian copula to learn an appropriate norm-coupling, here on the Boston dataset. We observe that the numerically optimised copula recovers the performance of the pairwise norm coupling scheme we proposed. This suggests that the proposed scheme may in fact be (close to) optimal. Rigorous analytical investigation is an important avenue for future work.

## B RFF AND RLF EXPERIMENTAL DETAILS

In this appendix, we supplement the discussion in Sec. 3.3, providing more details of our experimental setup for Gram matrix estimation. We also apply norm-coupled RFs to sparse spectrum Gaussian processes (Lázaro-Gredilla et al., 2010), showing that in this case variance reduction does not help downstream performance. We provide experimental details for the Performer (Choromanski et al., 2020) results in Table 2.

### B.1 DETAILS FOR RFF AND RLF EXPERIMENTS

**Overview.** In both our RFF and RLF experiments, we compare different coupling schemes for approximating the Gaussian kernel. The Gaussian kernel, including a lengthscale parameter $\ell$, an output scale variable $\sigma_v$ and a noise scale parameter $\sigma_n$, takes the form

$$k(x_i, x_j) = \sigma_v^2 \exp\left(-\frac{1}{2\ell^2}||\boldsymbol{x}_i - \boldsymbol{x}_j||_2^2\right).$$

Our baselines include standard methods for sampling random frequency vectors for use within RFFs and RLFs: i.i.d. sampling and Halton sequences (Halton, 1960). In addition, for both settings, we consider ensembles of frequency vectors that are coupled to have orthogonal directions but i.i.d. lengths. For a dataset of dimension $d$, for RFFs we use ensembles of $d$ orthogonal vectors. For RLFs we use ensembles of $2d$ vectors, including $d$ orthogonal basis vectors and their $d$ antiparallel vectors.

**Selecting kernel hyperparameters.** We want to compare our coupling schemes using realistic kernel hyperparameter values, which we determine as follows. A realistic application setting for RFFs is within GPs for probabilistic regression. Therefore, we first fit a GP on a tractable subset of the data, specifically a maximum of 256 randomly chosen datapoints, to select appropriate parameters $\ell, \sigma_v$ and $\sigma_n$. We optimise the exact GP marginal likelihood with respect to these hyperparameters, and subsequently fix them. On the other hand, it is well-documented that RLFs suffer from poor estimator concentration when the data norm becomes large because of the exponential function in the feature map (Eq. 4); see e.g. Thm. 4 and App. F6 of Choromanski et al. (2020) or Thm. 4.3 of Likhosherstov et al. (2022), where the authors bound the $L_2$-norm of queries and keys. This is anecdotally responsible for the deterioration in performance of Performers when networks become

very deep. To reflect this fact and choose a data regime where vanilla RLFs can perform reasonably well (so we can assess any gains from our coupling), we set the lengthscale $\ell$ to two twice the average summed norm of the data, namely

$$\ell = \frac{2}{N_{\text{train}}^2} \sum_{i,j=1}^{N_{\text{train}}} ||\boldsymbol{x}_i + \boldsymbol{x}_j||_2, \tag{36}$$

over the training set. We train the rest of the kernel parameters ($\sigma_v$ and $\sigma_n$) to maximise the marginal likelihood of the data under the exact GP.

**Splitting procedure.** To obtain mean evaluation metrics and standard errors, we evaluate the methods on multiple random splits as follows. For each dataset, we conduct cross validation with 20 splits, splitting each dataset into a training and a test set. Because we train an exact GP to determine kernel hyperparameters and evaluate its predictive NLL, we need to limit the number of datapoints used in both the training and the test set. We set them to a maximum of 256 points each by sub-sampling at random without replacement. After training the GP, we evaluate the metrics on the test set, and repeat this procedure for all 20 splits.

**Optimisation details.** We train the exact GP using the Adam optimiser (Kingma and Ba, 2014), using a learning rate of $10^{-2}$. The exact GP optimisation stage converges around 1000 steps, and we run it up to 5000 steps.

## B.2 PNC RFFs FOR GAUSSIAN PROCESSES

**Kernel and posterior approximations.** Suppose that we have drawn an ensemble of frequency vectors from which we construct random features $\{\phi_{\text{RF}}(\boldsymbol{x}_i)_{i=1}^N\}$. Group these in a large design matrix $\boldsymbol{\Phi}$,

$$\boldsymbol{\Phi} := [\phi_{\text{RF}}(\boldsymbol{x}_i)]_{i=1}^N, \tag{37}$$

where $\phi_{\text{RF}}(\boldsymbol{x}_i)$ of course depends on the ensemble frequencies $\{\boldsymbol{\omega}_i\}_{i=1}^m$ (suppressed for notational compactness). For RFFs $\boldsymbol{\Phi} \in \mathbb{R}^{N \times 2m}$ whereas for RLFs $\boldsymbol{\Phi} \in \mathbb{R}^{N \times m}$. We estimate the Gram matrix by $\widehat{\mathbf{K}} := \boldsymbol{\Phi}\boldsymbol{\Phi}^\top$.

As noted by Lázaro-Gredilla et al. (2010), this RF kernel approximation is exactly equivalent to a linear model, namely

$$\boldsymbol{y} = \boldsymbol{w}\boldsymbol{\Phi} + \boldsymbol{\epsilon}, \tag{38}$$

where $\boldsymbol{w} \sim \mathcal{N}(\mathbf{0}, \mathbf{I})$ and $\boldsymbol{\epsilon} \sim \mathcal{N}(\mathbf{0}, \sigma_n^2 \mathbf{I})$. The prior covariance of this linear model is $\boldsymbol{\Phi}^\top \boldsymbol{\Phi} + \sigma_n^2 \mathbf{I}$, which is, by construction, equal in expectation to the exact covariance produced by the kernel, namely $\mathbf{K} + \sigma_n^2 \mathbf{I}$. The predictive means of the approximate linear model and corresponding exact model are

$$\boldsymbol{\mu}_{\text{approx}} = \boldsymbol{\Phi}_p \left( \frac{1}{\sigma_n^2} \boldsymbol{\Phi}_d \boldsymbol{\Phi}_d^\top + \boldsymbol{I} \right)^{-1} \boldsymbol{\Phi}_d \boldsymbol{y}, \tag{39}$$

$$\boldsymbol{\mu}_{\text{exact}} = \mathbf{K}_{pd}(\mathbf{K}_{dd} + \sigma_n^2 \mathbf{I})^{-1} \boldsymbol{y}, \tag{40}$$

whereas the predictive covariances are

$$\mathbf{C}_{\text{approx}} = \boldsymbol{\Phi}_p^\top \left( \frac{1}{\sigma_n^2} \boldsymbol{\Phi}_d \boldsymbol{\Phi}_d^\top + \boldsymbol{I} \right)^{-1} \boldsymbol{\Phi}_p + \sigma_n^2 \mathbf{I}, \tag{41}$$

$$\mathbf{C}_{\text{exact}} = \mathbf{K}_{pp} - \mathbf{K}_{pd}(\mathbf{K}_{dd} + \sigma_n^2 \mathbf{I})^{-1} \mathbf{K}_{dp}. \tag{42}$$

Here, $\boldsymbol{\Phi}_d$ and $\boldsymbol{\Phi}_p$ are the design matrices corresponding to the training inputs and prediction outputs respectively, $\mathbf{K}_{dd}$ is the covariance matrix corresponding the training inputs, $\mathbf{K}_{pp}$ is the covariance matrix corresponding to the prediction inputs and $\mathbf{K}_{pd}$ and $\mathbf{K}_{dp}$ are the cross-covariance matrices between the training and prediction datapoints. These models become exactly equivalent in the limit of an infinite number of features $m$ since the kernel approximation becomes exact.

We have seen that pairwise norm coupling improves the approximation of the Gram matrices $\mathbf{K}$. In particular, we are able to suppress the variance of each pointwise kernel estimate (Thm. 3.2 and Corr. 3.4), and therefore the relative Frobenius norm error between the true and approximate Gram matrices (Table 1). In light of the discussion above, it would be natural to assume that this would

result in more accurate approximations of the predictive mean and covariance. However, in the following section we will see that surprisingly this is *not* the case.

**Evaluating posterior approximation quality.** Table 3 takes the RFs from Sec. 3.3, where we found that our coupling can substantially improve the quality of kernel estimation. It then reports the KL divergence between the exact predictive posterior and the approximate predictive posteriors computed with RFFs and RLFs, respectively. To be clear, Eqs 39 and 41 can be rewritten

$$\boldsymbol{\mu}_{\text{approx}} = \widehat{\mathbf{K}}_{pd}(\widehat{\mathbf{K}}_{dd} + \sigma_n^2 \boldsymbol{I})^{-1}\boldsymbol{y}, \quad \mathbf{C}_{\text{approx}} = \widehat{\mathbf{K}}_{pp} - \widehat{\mathbf{K}}_{pd}(\widehat{\mathbf{K}}_{dd} + \sigma_n^2 \mathbf{I})^{-1}\widehat{\mathbf{K}}_{dp} \tag{43}$$

where $\widehat{\mathbf{K}}_{dd} = \boldsymbol{\Phi}_d^\top \boldsymbol{\Phi}_d$, $\widehat{\mathbf{K}}_{pd} = \boldsymbol{\Phi}_d^\top \boldsymbol{\Phi}_p$, $\widehat{\mathbf{K}}_{pd} = \boldsymbol{\Phi}_p^\top \boldsymbol{\Phi}_d$ and $\widehat{\mathbf{K}}_{pp} = \boldsymbol{\Phi}_p^\top \boldsymbol{\Phi}_p$. It is then straightforward to compute the KL divergence between Gaussian distributions with means $\boldsymbol{\mu}_{\text{approx}}$, $\boldsymbol{\mu}_{\text{exact}}$ and covariances $\mathbf{C}_{\text{approx}}$, $\mathbf{C}_{\text{exact}}$.

Surprisingly, we find that, even though our couplings improve the accuracy of kernel approximation, the approximate mean and covariance and hence the KL divergence to the true posterior do not necessarily improve. In Table 1 we routinely see variance reductions of 10-20%, but even with very many trials this is not reflected in the data splits used for Table 3. This is also the case for predictive RMSEs (normalised and unnormalised by the i.i.d. result) and the predictive negative log likelihoods on held out test sets, also reported in Table 3.

The reason for this experimental finding is that, as is clear in Eq. 43, the posterior is highly nonlinear in $\widehat{\mathbf{K}}$. This is on account of the presence of matrix multiplications and inversions. These nonlinear operations on the Gram matrix entries mean that, despite our *pointwise* estimates being unbiased, $\boldsymbol{\mu}_{\text{approx}}$ and $\mathbf{C}_{\text{approx}}$ are in fact biased. We have achieved our objective of variance reduction of $\widehat{\mathbf{K}}$ – with pairwise norm coupling, it is both theoretically mandated and empirically observed. But clearly this does not necessitate variance (or bias) reduction for $\boldsymbol{\mu}_{\text{approx}}$ and $\mathbf{C}_{\text{approx}}$. The relationship between the distribution of $\widehat{\mathbf{K}}$ and the distribution of the approximate posterior is more complex.

**Variance reduction does not always help predictive performance.** To sharpen this point, we now take a *single* data split of the POWER dataset and plot the the kernel approximation RMSE (i.e. kernel estimator variance), as well as various quantities of predictive interest, against the number of features $m$. We exclude the Halton coupling (which is consistently worse than 'orthogonal') for clarity of presentation. Fig. 1 (in the main text) shows the results.

The left hand panel confirms that we have achieved our stated objective of variance reduction. Moreover, for all couplings the quality of approximation improves as we introduce more ensembles of size $d$. Reading left to right, the other three panels show: (i) the KL divergence between the exact and approximate GP *predictive posteriors* (as in Table 3), (ii) the predictive RMSE on a held out test set, and (iii) the KL divergence between the exact and approximate GP *priors*. In every instance, orthogonality provides a substantial gain but, despite the encouraging results for kernel RMSE, there is no additional benefit from PNC.

However, it would be wrong to draw the simplistic conclusion that PNC does not give large enough variance savings to see downstream gains: comparable reductions from orthogonality yield substantial improvements. The problem is more fundamental, relating to how the joint distribution of kernel estimates – beyond just the second moment of its pointwise entries – interacts with nonlinear operations like matrix multiplication and inversion.

Table 4 gives companion results to Table 3 but for $m = 8d$ (instead of $m = d$). In this regime, all the kernel estimators are closer to the groundtruth kernel value so the difference between their performance on downstream tasks is smaller – even between i.i.d. and orthogonal frequencies, which are well-separated when $m = d$ in Table 3. Note that this is also behaviour is also clear from the upper range of number of features in Fig. 1, where again the performances saturate.

| **KL divergence (between approximate and exact predictive posteriors)** | | | | | | |
|---|---|---|---|---|---|---|
| FOURIER FEATURES | CONCRETE | ABALONE | CPU | POWER | AIRFOIL | BOSTON |
| I.I.D. | $1.491_{\pm0.106}$ | $0.013_{\pm0.001}$ | $1.570_{\pm0.417}$ | $0.150_{\pm0.007}$ | $0.357_{\pm0.025}$ | $2.128_{\pm0.197}$ |
| HALTON | $1.548_{\pm0.104}$ | $0.014_{\pm0.001}$ | $1.596_{\pm0.419}$ | $0.138_{\pm0.006}$ | $0.356_{\pm0.024}$ | $2.263_{\pm0.204}$ |
| ORTHOGONAL | $1.166_{\pm0.113}$ | $0.004_{\pm0.000}$ | $1.635_{\pm0.423}$ | $0.029_{\pm0.002}$ | $0.235_{\pm0.017}$ | $1.990_{\pm0.191}$ |
| + PNC | $1.168_{\pm0.113}$ | $0.004_{\pm0.000}$ | $1.589_{\pm0.416}$ | $0.029_{\pm0.002}$ | $0.235_{\pm0.017}$ | $1.985_{\pm0.190}$ |
| LAPLACE FEATURES | CONCRETE | ABALONE | CPU | POWER | AIRFOIL | BOSTON |
| I.I.D. | $0.552_{\pm0.064}$ | $0.028_{\pm0.003}$ | $5.925_{\pm1.961}$ | $0.199_{\pm0.011}$ | $0.230_{\pm0.024}$ | $0.759_{\pm0.068}$ |
| HALTON | $0.574_{\pm0.064}$ | $0.024_{\pm0.003}$ | $5.811_{\pm1.897}$ | $0.146_{\pm0.009}$ | $0.213_{\pm0.022}$ | $0.834_{\pm0.074}$ |
| ORTHOGONAL | $0.486_{\pm0.059}$ | $0.014_{\pm0.002}$ | $5.494_{\pm1.774}$ | $0.059_{\pm0.005}$ | $0.165_{\pm0.018}$ | $0.679_{\pm0.058}$ |
| + PNC + ANTITHETIC | $0.482_{\pm0.058}$ | $0.014_{\pm0.002}$ | $5.468_{\pm1.780}$ | $0.050_{\pm0.006}$ | $0.165_{\pm0.018}$ | $0.673_{\pm0.058}$ |
| **Predictive RMSE (normalised)** | | | | | | |
| FOURIER FEATURES | CONCRETE | ABALONE | CPU | POWER | AIRFOIL | BOSTON |
| I.I.D. | $1.000_{\pm0.027}$ | $1.000_{\pm0.043}$ | $1.000_{\pm0.179}$ | $1.000_{\pm0.016}$ | $1.000_{\pm0.025}$ | $1.000_{\pm0.067}$ |
| HALTON | $1.013_{\pm0.027}$ | $1.001_{\pm0.043}$ | $1.005_{\pm0.179}$ | $0.991_{\pm0.016}$ | $0.999_{\pm0.025}$ | $1.021_{\pm0.068}$ |
| ORTHOGONAL | $0.917_{\pm0.026}$ | $0.994_{\pm0.044}$ | $1.019_{\pm0.182}$ | $0.915_{\pm0.019}$ | $0.930_{\pm0.025}$ | $0.974_{\pm0.066}$ |
| + PNC | $0.917_{\pm0.026}$ | $0.994_{\pm0.044}$ | $1.033_{\pm0.180}$ | $0.915_{\pm0.019}$ | $0.930_{\pm0.025}$ | $0.975_{\pm0.066}$ |
| LAPLACE FEATURES | CONCRETE | ABALONE | CPU | POWER | AIRFOIL | BOSTON |
| I.I.D. | $1.000_{\pm0.028}$ | $1.000_{\pm0.043}$ | $1.000_{\pm0.192}$ | $1.000_{\pm0.017}$ | $1.000_{\pm0.029}$ | $1.000_{\pm0.073}$ |
| HALTON | $1.007_{\pm0.028}$ | $0.997_{\pm0.044}$ | $1.003_{\pm0.190}$ | $0.964_{\pm0.017}$ | $0.989_{\pm0.029}$ | $1.019_{\pm0.075}$ |
| ORTHOGONAL | $0.979_{\pm0.027}$ | $0.990_{\pm0.044}$ | $1.003_{\pm0.187}$ | $0.899_{\pm0.018}$ | $0.958_{\pm0.028}$ | $0.979_{\pm0.071}$ |
| + PNC + ANTITHETIC | $0.981_{\pm0.027}$ | $0.991_{\pm0.044}$ | $1.006_{\pm0.187}$ | $0.909_{\pm0.018}$ | $0.958_{\pm0.028}$ | $0.982_{\pm0.071}$ |
| **Predictive RMSE (unnormalised)** | | | | | | |
| FOURIER FEATURES | CONCRETE | ABALONE | CPU | POWER | AIRFOIL | BOSTON |
| I.I.D. | $11.333_{\pm0.304}$ | $2.587_{\pm0.112}$ | $64.259_{\pm11.506}$ | $4.840_{\pm0.076}$ | $3.468_{\pm0.086}$ | $4.976_{\pm0.332}$ |
| HALTON | $11.479_{\pm0.305}$ | $2.589_{\pm0.112}$ | $64.562_{\pm11.505}$ | $4.799_{\pm0.077}$ | $3.464_{\pm0.086}$ | $5.081_{\pm0.336}$ |
| ORTHOGONAL | $10.398_{\pm0.292}$ | $2.572_{\pm0.114}$ | $65.477_{\pm11.721}$ | $4.430_{\pm0.093}$ | $3.225_{\pm0.088}$ | $4.847_{\pm0.331}$ |
| + PNC | $10.398_{\pm0.291}$ | $2.572_{\pm0.115}$ | $66.404_{\pm11.594}$ | $4.429_{\pm0.093}$ | $3.225_{\pm0.088}$ | $4.852_{\pm0.330}$ |
| LAPLACE FEATURES | CONCRETE | ABALONE | CPU | POWER | AIRFOIL | BOSTON |
| I.I.D. | $10.098_{\pm0.284}$ | $2.587_{\pm0.112}$ | $85.772_{\pm16.446}$ | $5.025_{\pm0.083}$ | $3.243_{\pm0.094}$ | $4.224_{\pm0.310}$ |
| HALTON | $10.173_{\pm0.285}$ | $2.580_{\pm0.113}$ | $86.060_{\pm16.333}$ | $4.845_{\pm0.083}$ | $3.209_{\pm0.093}$ | $4.303_{\pm0.315}$ |
| ORTHOGONAL | $9.882_{\pm0.272}$ | $2.562_{\pm0.115}$ | $86.042_{\pm15.997}$ | $4.518_{\pm0.091}$ | $3.108_{\pm0.092}$ | $4.136_{\pm0.301}$ |
| + PNC + ANTITHETIC | $9.908_{\pm0.270}$ | $2.563_{\pm0.115}$ | $86.258_{\pm15.998}$ | $4.569_{\pm0.090}$ | $3.108_{\pm0.092}$ | $4.150_{\pm0.301}$ |
| **Predictive NLL** | | | | | | |
| FOURIER FEATURES | CONCRETE | ABALONE | CPU | POWER | AIRFOIL | BOSTON |
| I.I.D. | $4.609_{\pm0.106}$ | $2.388_{\pm0.042}$ | $7.077_{\pm0.926}$ | $3.037_{\pm0.020}$ | $2.764_{\pm0.043}$ | $4.808_{\pm0.437}$ |
| HALTON | $4.666_{\pm0.108}$ | $2.389_{\pm0.042}$ | $7.105_{\pm0.924}$ | $3.025_{\pm0.020}$ | $2.762_{\pm0.043}$ | $4.942_{\pm0.449}$ |
| ORTHOGONAL | $4.301_{\pm0.095}$ | $2.381_{\pm0.042}$ | $7.381_{\pm1.122}$ | $2.913_{\pm0.022}$ | $2.640_{\pm0.041}$ | $4.684_{\pm0.432}$ |
| + PNC | $4.304_{\pm0.095}$ | $2.381_{\pm0.042}$ | $7.543_{\pm1.086}$ | $2.913_{\pm0.022}$ | $2.639_{\pm0.041}$ | $4.693_{\pm0.433}$ |
| LAPLACE FEATURES | CONCRETE | ABALONE | CPU | POWER | AIRFOIL | BOSTON |
| I.I.D. | $3.930_{\pm0.059}$ | $2.389_{\pm0.042}$ | $12.981_{\pm3.953}$ | $3.076_{\pm0.023}$ | $2.644_{\pm0.041}$ | $3.384_{\pm0.235}$ |
| HALTON | $3.948_{\pm0.060}$ | $2.387_{\pm0.042}$ | $12.929_{\pm3.927}$ | $3.024_{\pm0.022}$ | $2.628_{\pm0.040}$ | $3.453_{\pm0.245}$ |
| ORTHOGONAL | $3.880_{\pm0.056}$ | $2.379_{\pm0.042}$ | $12.722_{\pm3.867}$ | $2.935_{\pm0.022}$ | $2.581_{\pm0.038}$ | $3.312_{\pm0.223}$ |
| + PNC + ANTITHETIC | $3.885_{\pm0.055}$ | $2.380_{\pm0.042}$ | $12.729_{\pm3.864}$ | $2.948_{\pm0.022}$ | $2.582_{\pm0.038}$ | $3.319_{\pm0.223}$ |

Table 3: **Downstream performance on held out test sets when $m = d$.** Includes KL divergence between exact and approximate GP predictive posteriors; predictive RMSEs (normalised to RMSE of I.I.D. and unnormalised); and predictive negative log likelihoods of the approximate GP predictive posteriors. Reported errors are equal to two standard errors, i.e. $98\%$ confidence intervals, computed by averaging across splits. Note that differences *between* data splits are responsible for large errors; *within* each split we take enough trials that the standard errors become small. Substantially lower kernel estimator variance does *not* in general guarantee better predictive performance, though performance tends to be no worse. While $m = d$ features is not enough for good predictive performance in absolute terms, it exaggerates the difference between couplings and makes for easier comparison (see e.g. Fig. 1). We average over 20 data splits.

**Kernel estimator RMSE (unnormalised)**

| FOURIER FEATURES | CONCRETE | ABALONE | CPU | POWER | AIRFOIL | BOSTON |
|---|---|---|---|---|---|---|
| I.I.D. | $0.527_{\pm 0.029}$ | $0.088_{\pm 0.012}$ | $0.264_{\pm 0.035}$ | $0.137_{\pm 0.014}$ | $0.301_{\pm 0.018}$ | $0.107_{\pm 0.009}$ |
| HALTON | $0.550_{\pm 0.031}$ | $0.089_{\pm 0.013}$ | $0.262_{\pm 0.036}$ | $0.123_{\pm 0.014}$ | $0.281_{\pm 0.019}$ | $0.126_{\pm 0.011}$ |
| ORTHOGONAL | $0.340_{\pm 0.014}$ | $0.061_{\pm 0.007}$ | $0.160_{\pm 0.033}$ | $0.094_{\pm 0.010}$ | $0.181_{\pm 0.012}$ | $0.074_{\pm 0.005}$ |
| +PNC | $0.304_{\pm 0.013}$ | $0.055_{\pm 0.006}$ | $0.141_{\pm 0.030}$ | $0.079_{\pm 0.007}$ | $0.154_{\pm 0.011}$ | $0.071_{\pm 0.005}$ |

| LAPLACE FEATURES | CONCRETE | ABALONE | CPU | POWER | AIRFOIL | BOSTON |
|---|---|---|---|---|---|---|
| I.I.D. | $1.312_{\pm 0.118}$ | $0.382_{\pm 0.159}$ | $0.645_{\pm 0.212}$ | $0.376_{\pm 0.044}$ | $0.629_{\pm 0.082}$ | $0.392_{\pm 0.057}$ |
| HALTON | $1.336_{\pm 0.380}$ | $0.297_{\pm 0.047}$ | $0.453_{\pm 0.075}$ | $0.249_{\pm 0.030}$ | $0.539_{\pm 0.084}$ | $0.357_{\pm 0.031}$ |
| ORTHOGONAL | $1.096_{\pm 0.103}$ | $0.285_{\pm 0.071}$ | $0.433_{\pm 0.121}$ | $0.159_{\pm 0.015}$ | $0.466_{\pm 0.062}$ | $0.433_{\pm 0.166}$ |
| +PNC + ANTITHETIC | $1.123_{\pm 0.148}$ | $0.227_{\pm 0.048}$ | $0.391_{\pm 0.092}$ | $0.113_{\pm 0.012}$ | $0.461_{\pm 0.087}$ | $0.338_{\pm 0.052}$ |

**KL divergence (between approximate and exact predictive posteriors)**

| FOURIER FEATURES | CONCRETE | ABALONE | CPU | POWER | AIRFOIL | BOSTON |
|---|---|---|---|---|---|---|
| I.I.D. | $0.122_{\pm 0.010}$ | $0.003_{\pm 0.001}$ | $0.046_{\pm 0.013}$ | $0.003_{\pm 0.001}$ | $0.017_{\pm 0.004}$ | $0.139_{\pm 0.030}$ |
| HALTON | $0.122_{\pm 0.010}$ | $0.503_{\pm 0.948}$ | $0.045_{\pm 0.012}$ | $0.003_{\pm 0.000}$ | $0.017_{\pm 0.004}$ | $0.150_{\pm 0.031}$ |
| ORTHOGONAL | $0.117_{\pm 0.009}$ | $0.002_{\pm 0.001}$ | $0.042_{\pm 0.011}$ | $0.003_{\pm 0.000}$ | $0.016_{\pm 0.004}$ | $0.133_{\pm 0.029}$ |
| +PNC | $0.115_{\pm 0.008}$ | $0.002_{\pm 0.001}$ | $0.041_{\pm 0.011}$ | $0.003_{\pm 0.000}$ | $0.015_{\pm 0.004}$ | $0.135_{\pm 0.029}$ |

| LAPLACE FEATURES | CONCRETE | ABALONE | CPU | POWER | AIRFOIL | BOSTON |
|---|---|---|---|---|---|---|
| I.I.D. | $0.234_{\pm 0.053}$ | $0.067_{\pm 0.050}$ | $1.827_{\pm 1.802}$ | $0.003_{\pm 0.001}$ | $0.051_{\pm 0.023}$ | $0.742_{\pm 0.319}$ |
| HALTON | $0.231_{\pm 0.048}$ | $0.066_{\pm 0.050}$ | $1.992_{\pm 2.062}$ | $0.003_{\pm 0.000}$ | $0.047_{\pm 0.023}$ | $0.754_{\pm 0.321}$ |
| ORTHOGONAL | $0.227_{\pm 0.049}$ | $0.066_{\pm 0.050}$ | $1.889_{\pm 1.838}$ | $0.003_{\pm 0.001}$ | $0.048_{\pm 0.023}$ | $0.713_{\pm 0.319}$ |
| +PNC + ANTITHETIC | $0.232_{\pm 0.053}$ | $0.067_{\pm 0.052}$ | $1.846_{\pm 1.776}$ | $0.003_{\pm 0.001}$ | $0.051_{\pm 0.024}$ | $0.713_{\pm 0.315}$ |

**Predictive RMSE (normalised)**

| FOURIER FEATURES | CONCRETE | ABALONE | CPU | POWER | AIRFOIL | BOSTON |
|---|---|---|---|---|---|---|
| I.I.D. | $1.000_{\pm 0.057}$ | $1.000_{\pm 0.073}$ | $1.000_{\pm 0.438}$ | $1.000_{\pm 0.051}$ | $1.000_{\pm 0.054}$ | $1.000_{\pm 0.174}$ |
| HALTON | $0.997_{\pm 0.054}$ | $1.001_{\pm 0.073}$ | $0.992_{\pm 0.429}$ | $1.001_{\pm 0.051}$ | $0.999_{\pm 0.056}$ | $0.996_{\pm 0.173}$ |
| ORTHOGONAL | $0.996_{\pm 0.056}$ | $1.001_{\pm 0.073}$ | $1.004_{\pm 0.443}$ | $1.000_{\pm 0.051}$ | $0.999_{\pm 0.056}$ | $0.999_{\pm 0.175}$ |
| +PNC | $1.001_{\pm 0.056}$ | $1.001_{\pm 0.073}$ | $1.008_{\pm 0.445}$ | $1.000_{\pm 0.051}$ | $0.997_{\pm 0.056}$ | $0.994_{\pm 0.171}$ |

| LAPLACE FEATURES | CONCRETE | ABALONE | CPU | POWER | AIRFOIL | BOSTON |
|---|---|---|---|---|---|---|
| I.I.D. | $1.000_{\pm 0.050}$ | $1.000_{\pm 0.080}$ | $1.000_{\pm 0.487}$ | $1.000_{\pm 0.051}$ | $1.000_{\pm 0.061}$ | $1.000_{\pm 0.199}$ |
| HALTON | $0.992_{\pm 0.050}$ | $1.001_{\pm 0.081}$ | $1.020_{\pm 0.484}$ | $1.000_{\pm 0.051}$ | $0.996_{\pm 0.060}$ | $1.004_{\pm 0.201}$ |
| ORTHOGONAL | $0.993_{\pm 0.050}$ | $1.000_{\pm 0.083}$ | $1.031_{\pm 0.507}$ | $1.000_{\pm 0.051}$ | $0.995_{\pm 0.060}$ | $0.995_{\pm 0.198}$ |
| +PNC + ANTITHETIC | $0.996_{\pm 0.052}$ | $1.002_{\pm 0.083}$ | $1.022_{\pm 0.492}$ | $1.000_{\pm 0.051}$ | $0.999_{\pm 0.061}$ | $0.992_{\pm 0.198}$ |

**Predictive RMSE (unnormalised)**

| FOURIER FEATURES | CONCRETE | ABALONE | CPU | POWER | AIRFOIL | BOSTON |
|---|---|---|---|---|---|---|
| I.I.D. | $8.016_{\pm 0.458}$ | $2.595_{\pm 0.189}$ | $56.066_{\pm 24.574}$ | $4.321_{\pm 0.221}$ | $2.689_{\pm 0.146}$ | $3.954_{\pm 0.689}$ |
| HALTON | $7.996_{\pm 0.429}$ | $2.597_{\pm 0.190}$ | $55.601_{\pm 24.064}$ | $4.326_{\pm 0.221}$ | $2.687_{\pm 0.150}$ | $3.938_{\pm 0.683}$ |
| ORTHOGONAL | $7.987_{\pm 0.449}$ | $2.597_{\pm 0.190}$ | $56.312_{\pm 24.840}$ | $4.321_{\pm 0.223}$ | $2.686_{\pm 0.150}$ | $3.948_{\pm 0.690}$ |
| +PNC | $8.026_{\pm 0.446}$ | $2.597_{\pm 0.189}$ | $56.506_{\pm 24.955}$ | $4.323_{\pm 0.221}$ | $2.682_{\pm 0.150}$ | $3.928_{\pm 0.677}$ |

| LAPLACE FEATURES | CONCRETE | ABALONE | CPU | POWER | AIRFOIL | BOSTON |
|---|---|---|---|---|---|---|
| I.I.D. | $8.244_{\pm 0.410}$ | $2.710_{\pm 0.218}$ | $81.045_{\pm 39.468}$ | $4.319_{\pm 0.221}$ | $2.773_{\pm 0.170}$ | $4.339_{\pm 0.862}$ |
| HALTON | $8.175_{\pm 0.410}$ | $2.713_{\pm 0.219}$ | $82.627_{\pm 39.211}$ | $4.320_{\pm 0.222}$ | $2.763_{\pm 0.168}$ | $4.356_{\pm 0.871}$ |
| ORTHOGONAL | $8.183_{\pm 0.411}$ | $2.710_{\pm 0.224}$ | $83.571_{\pm 41.127}$ | $4.320_{\pm 0.220}$ | $2.761_{\pm 0.166}$ | $4.319_{\pm 0.860}$ |
| +PNC + ANTITHETIC | $8.212_{\pm 0.428}$ | $2.715_{\pm 0.225}$ | $82.838_{\pm 39.877}$ | $4.320_{\pm 0.219}$ | $2.770_{\pm 0.169}$ | $4.305_{\pm 0.857}$ |

**Predictive NLL**

| FOURIER FEATURES | CONCRETE | ABALONE | CPU | POWER | AIRFOIL | BOSTON |
|---|---|---|---|---|---|---|
| I.I.D. | $3.467_{\pm 0.085}$ | $2.375_{\pm 0.073}$ | $5.021_{\pm 0.202}$ | $2.888_{\pm 0.057}$ | $2.407_{\pm 0.048}$ | $2.793_{\pm 0.258}$ |
| HALTON | $3.471_{\pm 0.084}$ | $2.376_{\pm 0.073}$ | $5.021_{\pm 0.205}$ | $2.888_{\pm 0.057}$ | $2.408_{\pm 0.048}$ | $2.797_{\pm 0.262}$ |
| ORTHOGONAL | $3.459_{\pm 0.088}$ | $2.375_{\pm 0.073}$ | $5.023_{\pm 0.204}$ | $2.888_{\pm 0.057}$ | $2.406_{\pm 0.048}$ | $2.787_{\pm 0.258}$ |
| +PNC | $3.462_{\pm 0.088}$ | $2.375_{\pm 0.073}$ | $5.018_{\pm 0.202}$ | $2.888_{\pm 0.057}$ | $2.407_{\pm 0.048}$ | $2.784_{\pm 0.262}$ |

| LAPLACE FEATURES | CONCRETE | ABALONE | CPU | POWER | AIRFOIL | BOSTON |
|---|---|---|---|---|---|---|
| I.I.D. | $3.546_{\pm 0.113}$ | $2.438_{\pm 0.104}$ | $9.694_{\pm 7.105}$ | $2.888_{\pm 0.057}$ | $2.442_{\pm 0.060}$ | $3.693_{\pm 0.978}$ |
| HALTON | $3.542_{\pm 0.111}$ | $2.437_{\pm 0.104}$ | $9.722_{\pm 7.220}$ | $2.888_{\pm 0.057}$ | $2.438_{\pm 0.058}$ | $3.708_{\pm 0.989}$ |
| ORTHOGONAL | $3.544_{\pm 0.112}$ | $2.436_{\pm 0.102}$ | $9.836_{\pm 7.410}$ | $2.887_{\pm 0.057}$ | $2.440_{\pm 0.059}$ | $3.637_{\pm 0.947}$ |
| +PNC + ANTITHETIC | $3.543_{\pm 0.114}$ | $2.435_{\pm 0.102}$ | $9.686_{\pm 7.279}$ | $2.888_{\pm 0.057}$ | $2.437_{\pm 0.058}$ | $3.648_{\pm 0.947}$ |

Table 4: **Equivalent results for $m = 8d$.** Companion results to Table 3 taking $m = 8d$ features, so the quality of kernel approximation and performance on downstream tasks are both much better. In all cases, the approximate kernel converges to the true kernel so it becomes more difficult to distinguish the different algorithms. We still observe that our methods provide substantial variance reduction (see top row).

### B.3 NORM-COUPLED RLFs IN PERFORMERS

In Sec. 3.3, we used random Laplace features for attention approximation in Performers (Choromanski et al., 2020), a type of efficient transformer that estimates softmax using a low rank decomposition. Here, we discuss the results in greater detail.

**Derivation of Eq. 8.** Let $X_{ij}$ denote the random variable $\widehat{k}(\boldsymbol{x}_i, \boldsymbol{x}_j)$, which is an unbiased estimate of $\exp(\boldsymbol{x}_i^\top \boldsymbol{x}_j)$ constructed using features $\{\exp(\|\boldsymbol{x}_i\|^2/2)\phi_{\mathrm{RLF}}(\boldsymbol{x}_i)\}_{i=1}^N$. Unbiasedness follows trivially from the fact that RLFs give an unbiased estimate of the Gaussian kernel. Normalising by the sum of attention scores, let us define $\widehat{a}_{ij} := X_{ij}/\sum_j X_{ij}$. Let $X_{ij} = \mu_{ij} + \delta_{ij}$ with $\mu_{ij} := \mathbb{E}(X_{ij})$ and $\mathbb{E}(\delta_{ij}) = 0$. Then

$$
\begin{aligned}
\widehat{a}_{ij}^2 &= (\mu_{ij}^2 + 2\mu_{ij}\delta_{ij} + \delta_{ij}^2)(\sum_k \mu_{ik} + \sum_k \delta_{ik})^{-2} \\
&= \frac{\mu_{ij}^2 + 2\mu_{ij}\delta_{ij} + \delta_{ij}^2}{N^2 \bar{\mu}_i^2} \left(1 - 2\frac{\sum_k \delta_{ik}}{N\bar{\mu}_i} + 3\frac{\sum_{kl} \delta_{ik}\delta_{il}}{N^2 \bar{\mu}_i^2} + \mathcal{O}(\frac{1}{N^3})\right),
\end{aligned}
\tag{44}
$$

where we defined $\bar{\mu}_i := \frac{1}{N}\sum_j \mu_{ij}$, the average groundtruth attention score across tokens. Since we are targeting $\frac{\mu_{ij}}{N\bar{\mu}_i}$,

$$
\mathrm{MSE}(\widehat{a}_{ij}) = \frac{1}{N^2 \bar{\mu}_i{}^2} \left(\delta_{ij}^2 - \frac{4\mu_{ij}\delta_{ij}}{N\bar{\mu}_i}\sum_k \delta_{ik} + \frac{3\mu_{ij}^2}{N^2 \bar{\mu}_i^2}\sum_{kl} \delta_{ik}\delta_{il}\right) + \mathcal{O}\left(\frac{1}{N^3}\right).
\tag{45}
$$

As in the main text, denote $\mathrm{MSE}(\widehat{a}_i) := \frac{1}{N}\sum_j \mathrm{MSE}(\widehat{a}_{ij})$, the average mean squared error over the tokens to which $i$ attends. To better see the intuitive behaviour, suppose also that $\mu_{ij} = \bar{\mu}_i \,\forall\, j$. Then

$$
\begin{aligned}
\mathrm{MSE}(\widehat{a}_i) &= \frac{1}{N^2 \bar{\mu}_i{}^2} \left(\frac{1}{N}\sum_j \delta_{ij}^2 - \frac{1}{N^2}\sum_{jk} \delta_{ij}\delta_{ik}\right) + \mathcal{O}\left(\frac{1}{N^3}\right) \\
&= \frac{1}{N^2 \bar{\mu}_i^2} \left(\frac{1}{N}\sum_j \mathrm{Var}(\widehat{k}(\boldsymbol{x}_i, \boldsymbol{x}_j)) - \frac{1}{N^2}\sum_{j,k} \mathrm{Cov}(\widehat{k}(\boldsymbol{x}_i, \boldsymbol{x}_j), \widehat{k}(\boldsymbol{x}_i, \boldsymbol{x}_k))\right) + \mathcal{O}\left(\frac{1}{N^3}\right),
\end{aligned}
\tag{46}
$$

as reported in Eq. 8.

**Positive monotone coupling.** In contrast to Eq. 7, we say that random variables $(\omega_1, \omega_2)$ are *postive monotone-coupled* if $\omega_1 = \omega_2$ almost surely. This is a valid transport plan, the identity. Clearly, all $m$ frequencies in an ensemble can be simultaneously positive monotone-coupled by making them equal. It is intuitive that this should maximise the kernel estimator variance since we sample only one frequency lengthscale. The proof is a trivial extension of the arguments made for Thm. 3.2 in App. A.2, simply taking $c_{\mathrm{RLF}} \to -c_{\mathrm{RLF}}$ so that the support of the OT plan swaps. We omit it for brevity. It is also intuitive that this will increase the covariance between kernel estimates.

**Variance, covariance, and attention approximation error.** Modifying the norm coupling between frequencies changes both $\mathrm{Var}(\widehat{k}(\boldsymbol{x}_i, \boldsymbol{x}_j))$ and $\mathrm{Cov}(\widehat{k}(\boldsymbol{x}_i, \boldsymbol{x}_j))$, so $\mathrm{MSE}(\widehat{a}_i)$ can change unpredictably. Fig. 7 shows the results for randomly $N = 16$ synthetic, normally distributed 16-dimensional keys, $\boldsymbol{x} \sim \mathcal{N}(0, \frac{1}{\sqrt{d}}\mathbf{I}_d)$. Strikingly, maximising the kernel estimator variance with a positive monotone (PM) coupling ends up *improving* the quality of attention estimation since it also also increases the covariance between the unnormalised scores. This counterintuitive finding shows highlights the limitations of variance reduction as a paradigm.

**Performer experimental details.** To obtain the results in Table 2, we train a Performer-ViT (Dosovitskiy et al., 2020; Choromanski et al., 2020) on the ImageNet (1M) dataset (Deng et al., 2009). We estimate attention using $m = 128$ random features that are orthogonal with norms that are: (i) i.i.d. (ii) PNC (Def. 3.3), or (iii) positive monotone-coupled (i.e. equal among blocks of size $d$). We use a transformer with 12 layers and heads, with hidden size 768 and MLP dimension 3072. We take $16 \times 16$ patches, and train with the Adam optimiser for 90 epochs with a compound learning rate ($10^4$ steps linear warmup, constant, then cosine decay, with base LR $3 \times 10^{-3}$ and final LR $1 \times 10^{-5}$). The batch size is 4096. To get the standard errors, we average between 34 and 48 seeds per coupling. We see that the lower attention MSE unlocked by PM coupling improves predictive performance over the baseline.

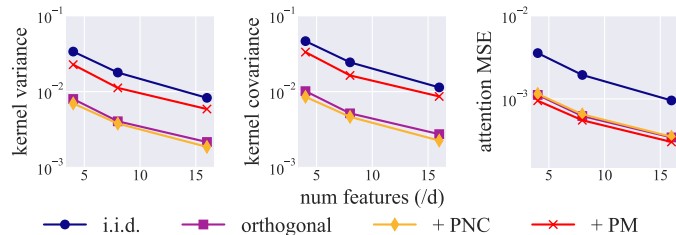

Figure 7: Kernel variance $\text{Var}(\widehat{k}(\boldsymbol{x}_i, \boldsymbol{x}_j))$, kernel covariance $\text{Cov}(\widehat{k}(\boldsymbol{x}_i, \boldsymbol{x}_j))$ and attention mean squared error $\text{MSE}(a_i)$ plotted against number of frequencies for different coupling schemes. As intended, PNC decreases the kernel estimator variance, but it also decreases the kernel estimator covariance so the attention MSE remains essentially unchanged. On the other hand, positive monotone coupling (PM) drastically increases both kernel variance (which it maximises) and kernel covariance. The latter offsets the former (Eq. 46), so that attention MSE actually drops: a remarkable, counterintuitive finding.

## C  GRAPH RANDOM FEATURES

In this appendix, we provide a self-contained introduction to graph random features (GRFs) and previously proposed techniques to reduce their variance. We also explain the motivations behind the series of approximations used in Sec. 4.1, and where possible provide empirical evidence that these approximations for tractability and efficiency do not substantially degrade the performance of the learned $\sigma$-coupling.

### C.1  CONSTRUCTING GRAPH RANDOM FEATURES

For the reader's convenience, we begin by providing a brief introduction to GRFs.

**Graph node kernels.** Recall that *graph node kernels* $k : \mathcal{N} \times \mathcal{N} \to \mathbb{R}$ are positive definite, symmetric functions defined on pairs of nodes of $\mathcal{G}$. Note that one can also define kernels that take pairs of graphs as inputs, but these are not the object of our study.

Many of the most popular graph node kernels in the literature are functions of weighted adjacency matrices (Smola and Kondor, 2003b; Chapelle et al., 2002). In particular, they are often functions of the *graph Laplacian matrix*,

$$\mathbf{L} := \mathbf{D} - \mathbf{W}, \tag{47}$$

where $\mathbf{W}$ is a (weighted) adjacency matrix and $\mathbf{D}$ is a diagonal degree matrix (satisfying $\mathbf{D}_{ii} = \sum_j \mathbf{W}_{ij}$). It is also common to consider its normalised variant,

$$\widetilde{\mathbf{L}} := \mathbf{D}^{-\frac{1}{2}} \mathbf{L} \mathbf{D}^{-\frac{1}{2}}, \tag{48}$$

whose spectrum is contained in $[0, 2]$ (Chung, 1997). We provide examples in Table 5.

| Name | Form |
|---|---|
| $d$-regularised Laplacian | $(\mathbf{I}_N + \sigma^2 \mathbf{L})^{-d}$ |
| $p$-step random walk | $(\alpha \mathbf{I}_N - \mathbf{L})^p, \alpha \geq 2$ |
| Diffusion | $\exp(-\sigma^2 \mathbf{L}/2)$ |
| Inverse Cosine | $\cos(\mathbf{L}\pi/4)$ |

Table 5: Examples of graph node kernels. The $\exp$ and $\cos$ mappings are defined via Taylor series expansions rather than element-wise, e.g. $\exp(\mathbf{M}) := \lim_{n \to \infty}(\mathbf{I}_N + \mathbf{M}/n)^n$ and $\cos(\mathbf{M}) := \text{Re}(\exp(i\mathbf{M}))$. $\sigma$ and $\alpha$ are regularisers.

**Formula for graph random features.** Computing graph node kernels exactly is computationally expensive because of the $\mathcal{O}(N^3)$ cost of e.g. matrix inversion or exponentiation. Inspired by

the success of random Fourier features in the Euclidean domain (Rahimi and Recht, 2007), the recently-introduced class of *graph random features* (GRFs) permits unbiased approximation of $k$ in subquadratic time (Choromanski, 2023; Reid et al., 2024b). Intuitively, GRFs interpret the powers of weighted adjacency matrices in the Taylor expansions of the expressions in Table 5 as weighted sums over walks on a graph. Instead of computing this infinite sum of walks exactly, we can sample $m$ finite walks (using importance sampling) to construct an unbiased estimate of $k$.

Concretely, GRFs compute a set of estimators $\{\phi_{\text{GRF}}(v_i)\}_{i \in \mathcal{N}}$ such that $k(v_i, v_j) = \mathbb{E}[\phi_{\text{GRF}}(v_i)^\top \phi_{\text{GRF}}(v_j)]$ by taking:[3]

$$\phi_{\text{GRF}}(v_i) = \frac{1}{m} \sum_{k=1}^{m} \psi(\boldsymbol{\omega}_k^{(i)}), \tag{49}$$

where $\boldsymbol{\omega}_k^{(i)}$ is the $k$th walk (out of a total of $m$) simulated out of the starting node $v_i$. We remind the reader that by a 'walk' we mean a sequence of neighbouring graph nodes. The projection function $\psi(\cdot) : \Omega \rightarrow \mathbb{R}^N$ maps from the set of graph random walks $\Omega := \left\{ (v_i)_{i=1}^{l} \mid v_i \in \mathcal{N}, (v_i, v_{i+1}) \in \mathcal{E}, l \in \mathbb{N} \right\}$ to a sparse $N$-dimensional feature vector. It is computed as follows:

$$\psi(\boldsymbol{\omega}^{(i)})_q := \sum_{\omega_{iq} \in \Omega_{iq}} \frac{\widetilde{\omega}(\omega_{iq}) f(\text{len}(\omega_{iq}))}{p(\omega_{iq})} \mathbb{I}(\omega_{iq} \in \boldsymbol{\omega}^{(i)}), \quad q = 1, ..., N. \tag{50}$$

In this expression,

- $\psi(\boldsymbol{\omega}^{(i)})_q$ is the $q$th component of the feature projection of the random walk $\boldsymbol{\omega}^{(i)}$ (itself a sequence of neighbouring graph nodes beginning with $v_i$).
- $\Omega_{iq}$ is the set of *all* graph walks between nodes $v_i$ and $v_q$, of which each $\omega_{iq}$ is a member.
- $\mathbb{I}(\cdot)$ is the indicator function that evaluates to 1 if its argument is true and 0 otherwise.
- $\omega_{iq} \in \boldsymbol{\omega}^{(i)}$ is the condition that $\omega_{iq}$, a particular walk between nodes $v_i$ and $v_q$, is a *prefix subwalk* of $\boldsymbol{\omega}^{(i)}$, the random walk we actually sample.[4]
- $f : \mathbb{N} \rightarrow \mathbb{R}$ is the *modulation function*, which controls the particular graph node kernel we choose to approximate.
- $\text{len}(\omega_{iq})$ is the length of graph walk $\omega_{iq}$.
- $p(\omega_{iq})$ is the *marginal probability* of sampling a random walk $\omega_{iq}$, which is known from the sampling strategy.
- $\widetilde{\omega}(\omega_{iq})$ is a function that returns the *products of weights* of the edges traversed by $\omega_{iq}$.

Intuitively, to construct $\psi(\boldsymbol{\omega}^{(i)})$ using a random walk $\boldsymbol{\omega}^{(i)}$, we look at the node that it visits at each of the $\text{len}(\boldsymbol{\omega}^{(i)}) + 1$ timesteps before termination. At every node, we add a contribution to the feature at the corresponding coordinate that depends on (i) the product of edge weights traversed so far, (ii) the known marginal probability of sampling the walk so far, and (iii) a 'modulation function' $f$ applied to the number of steps taken so far. We refer the reader to the original works of Choromanski (2023) and Reid et al. (2024b) for further technical details, experimental results and a proof of unbiasedness.

## C.2   ANTITHETIC TERMINATION

In this section, we briefly describe *antithetic termination* (Reid et al., 2024c): a previously-proposed variance reduction algorithm that couples the lengths of pairs of random walks.

Consider a random walker that terminates with probability $p$ at every timestep. In practice, this can be achieved by drawing a 'termination random variable' $t$ from a uniform distribution on $[0, 1]$, $t \sim \mathcal{U}([0, 1])$, and ending the walk if $t < p$. Now consider two such walkers with corresponding termination random variables $t_1, t_2$. If they are independent, $t_1$ and $t_2$ are drawn independently. Antithetic termination instead proposes to induce a nontrivial joint distribution by first drawing

---

[3]Strictly, for unbiased estimation of diagonal kernel entries $\{k(v_i, v_i)\}_{v_i \in \mathcal{N}}$ we should construct *two* independent sets of features, but this is found to make little practical difference (Reid et al., 2024b) so we omit further discussion in this manuscript.

[4]Meaning that the walk $\boldsymbol{\omega}^{(i)}$ from node $v_i$ initially follows $\omega_{iq}$, then optionally continues to visit further nodes. Note the subtle difference in usage of the symbol $\in$ compared to in the expression $\omega_{iq} \in \Omega_{iq}$, where it means 'a member of the set of walks' rather than 'a prefix subwalk'.

$t_1 \sim \mathcal{U}([0,1])$ as usual, and then setting $t_2 = \mod_1(t_1 + \frac{1}{2})$. $t_2$ still follows a uniform marginal distribution so we preserve unbiasedness, but this coupling forces walkers to terminate at different timesteps (if $p_{\text{halt}} < 0.5$) which the authors prove reduces the estimator variance under certain assumptions. This is one example of a possible hand-crafted coupling between random walk lengths that improves estimator concentration.

It is also possible to improve the accuracy of estimators using graph random walks by coupling walker *directions* (Reid et al., 2024a; Luo, 2019). This approach is complementary to our own and can be combined with it. Formulating coupled walker directions in terms of OT is an interesting and involved technical challenge; we defer its study to future work.

### C.3 Approximating the OT problem: details and discussion for Sec. 4.1

In this appendix, we discuss the steps to formulate the OT matching problem in Sec. 4.1. We will especially focus on the approximations needed to make the objective in Eq. 12 tractable, describing their motivation and where possible investigating how they modify the original objective.

To begin, we remind the reader of the OT formulation of the variance reduction problem for GRFs:

$$
\text{minimise } \mathbb{E}_{(l_1^{(i)}, l_2^{(i)}) \sim \mu, (l_1^{(j)}, l_2^{(j)}) \sim \mu} \left( \mathbb{E}_{\text{dirs}} \left[ \left( \psi(\boldsymbol{\omega}_1^{(i)}) + \psi(\boldsymbol{\omega}_2^{(i)}) \right)^\top \left( \psi(\boldsymbol{\omega}_1^{(j)}) + \psi(\boldsymbol{\omega}_2^{(j)}) \right) \right]^2 \right)
$$
$$
\text{for } \mu \in \Lambda_2(\eta_{\text{G}}), \tag{51}
$$

where $\Lambda_2(\eta_{\text{G}})$ is the set of joint distributions on $\mathbb{N}^2$ with geometrically distributed marginals, $l_1^{(i)}$ is the length of walk $\boldsymbol{\omega}_1^{(i)}$ out of node $v_i$, $\mathbb{E}_{\text{dirs}}$ averages the *directions* of walkers of a particular length, and $\psi(\boldsymbol{\omega}_k^{(i)})$ is the *projection function* that maps from random walks to feature vectors (see Eq. 9).

**Averaging walker trajectories.** As noted in the main text, the cost function in Eq. 51 is not analytically tractable. We can approximate it using MC sampling of graph random walks. To do so, it will be convenient to introduce the *direction-averaged feature vector* $\widetilde{\psi}(\cdot) : \mathbb{N} \to \mathbb{R}^N$ satisfying

$$
\widetilde{\psi}(l^{(i)}) := \mathbb{E}_{\text{dirs}}[\psi(\boldsymbol{\omega}^{(i)})]. \tag{52}
$$

This computes the *average* feature projection over all walks of a given length $l^{(i)}$ out of node $v_i$. It is straightforward to estimate by simulating a finite number of random walks. If we move the expectation over walker directions *inside* the square bracket, the (now approximate) OT problem becomes

$$
\text{minimise } \mathbb{E}_{(l_1^{(i)}, l_2^{(i)}) \sim \mu, (l_1^{(j)}, l_2^{(j)}) \sim \mu} \left[ \left( \widetilde{\psi}(l_1^{(i)}) + \widetilde{\psi}(l_2^{(i)}) \right)^\top \left( \widetilde{\psi}(l_1^{(j)}) + \widetilde{\psi}(l_2^{(j)}) \right) \right]^2 \text{ for } \mu \in \Lambda_2(\eta_{\text{G}}). \tag{53}
$$

The price we pay is that we have ignored some higher-order corrections from the variance of the directions of the walks, but as we have seen in Sec. 4.2 this does not prevent us from obtaining an effective, computationally cheap coupling.

**Using $\sigma$-couplings.** Unfortunately, Eq. 53 remains intractable. To make progress, we must restrict our search-space of joint distributions from $\Lambda_2(\eta_{\text{G}})$ – the set of *all* joint distributions with geometrically-distributed marginals – to a tractable subclass. Taking inspiration from approaches in numerical OT, we have considered the class of *permutation densities* defined by $p_\sigma(x, y) := n \mathbb{1}_{\sigma(\lceil nx \rceil) = \lceil ny \rceil}$, with $\sigma$ a permutation of order $n$ (that is, a bijection $\sigma : [\![n]\!] \to [\![n]\!]$), transformed coordinate-wise by pushing forward with the appropriate inverse CDF. This may initially seem an arbitrary choice, but it is motivated by three important observations. First, we have seen that this class of joint distributions frames the minimisation as a *matching problem* which, under further simplifying assumptions, we can efficiently solve using techniques from linear programming. Second, it is straightforward to sample from so the learned coupling will be practically useful. Third, it is well-motivated by results in discrete OT; see App. D for concrete theoretical guarantees. Once again, we have seen that in practice, even at modest permutation order $n$, this class contains couplings that can substantially outperform both i.i.d. walkers and antithetic termination (Reid et al., 2024c).

We now replace $\Lambda_2(\eta_{\text{G}})$ in Eq. 53 by the set of measures $\{\mu^{(\sigma)}\}_{\sigma \in S_n}$ obtained by pushing permutation densities through the required inverse CDF. $S_n$ is the set of permutations $[\![n]\!] \to [\![n]\!]$. Denote by $\widehat{\psi}(\cdot) : [\![n]\!] \to \mathbb{R}^N$ the function

$$
\widehat{\psi}(q^{(i)}) := \mathbb{E}_{u \sim \mathcal{U}((\frac{q-1}{n}, \frac{q}{n}])} \left( \widetilde{\psi}(F_{\eta_{\text{G}}}^{-1}(u)^{(i)}) \right). \tag{54}
$$

This takes the direction-averaged feature vector for node $v_i$ (see Eq. 52) and evaluates a *further* average over walk lengths corresponding to the $q$th quantile of the geometric distribution $\eta_G$. Again, this is straightforward to estimate by simulating walks. With a further approximation of the objective, we can consider

$$\sigma^* = \arg\min_{\sigma \in S_n} \sum_{q_1 \in [\![n]\!]} \sum_{q_2 \in [\![n]\!]} \left[ \left( \widehat{\psi}(q_1^{(i)}) + \widehat{\psi}(\sigma(q_1)^{(i)}) \right)^\top \left( \widehat{\psi}(q_2^{(j)}) + \widehat{\psi}(\sigma(q_2)^{(j)}) \right) \right]^2 \tag{55}$$

which is a matching problem.

**Solving the matching problem.** Though we can now efficiently estimate the cost function with MC, it is is *still* difficult to solve the matching problem in Eq. 55 efficiently because it is quadratic (Finke et al., 1987; Burkard et al., 1998). Minimum weights quadratic bipartite matching problems are generally NP-hard, but here the problem has extra symmetry that makes it simpler.

In App. C.3.1, we discuss possible approaches to directly solve the matching problem in Eq. 55. In particular, we prove that it can be solved with high probability at a time complexity independent of the number of nodes $N$, and give a polynomial time algorithm for the special case of diagonal kernel entries $k(v_i, v_i)$. Notwithstanding this progress, as a simpler first recourse we can just make a further approximation by restricting the sum to $q_1 = q_2$ diagonal terms. This is the approximation used to get the results in the main text. Doing so, we arrive at

$$\sigma^* = \arg\min_{\sigma \in S_n} \sum_{q \in [\![n]\!]} \left[ \left( \widehat{\psi}(q^{(i)}) + \widehat{\psi}(\sigma(q)^{(i)}) \right)^\top \left( \widehat{\psi}(q^{(j)}) + \widehat{\psi}(\sigma(q)^{(j)}) \right) \right]^2 . \tag{56}$$

This is now a minimum-weights bipartite matching problem with *linear* weights. Even though the search space of possible permutations of order $n$ is of size $n!$, it is well known that it can be solved efficiently and exactly using algorithms from linear programming (e.g. the Hungarian algorithm (Kuhn, 1955)) in time $\mathcal{O}(n^3)$. As stated in Sec. 4.1 of the main text, we can use this optimal permutation to construct $\sigma$-*coupled GRFs*.

This approximation might seem unreasonable since it involves discarding all the off-diagonal $q_1 \neq q_2$ terms from the objective, but for modest permutation order $n$ we can empirically test the effect this has on the quality of the obtained coupling. We achieve this by comparing to the *best* possible permutation obtained by exhaustively searching the $n!$ possibilities.

Fig. 8 plots the results for GRF estimates of the 2-regularised Laplacian kernel on the `eurosis` graph, comparing the approximate and exact objectives. The former can be computed efficiently but the latter is evaluated at a cost exponential in $n$, so we limit to $n = 10$ and smaller. We plot the kernel approximation error (normalised by the i.i.d. result) for different permutation orders. We also plot the *Cayley distance* between the permutations, which measures the difference between them. As $n$ grows, the permutations become more different so we are clearly not finding the same coupling. Nonetheless, we do not see a statistically significant difference in the amount of variance reduction. This suggests that our linear proxy objective is reasonable. We defer more detailed analytic study of this curious phenomenon to future work.

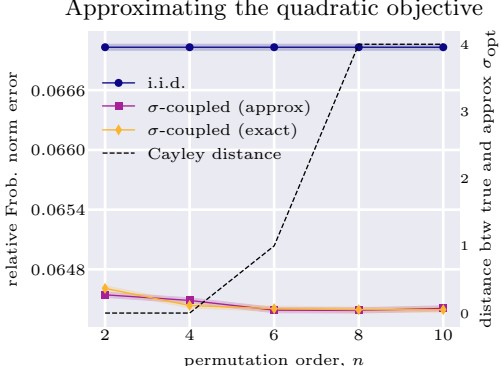

Figure 8: Relative Frobenius norm error (normalised by i.i.d. result) with permutations of order $n$ learned using exact and approximate objectives. 'Cayley distance' shows the difference between the learned permutations. Although dropping off-diagonal terms from the objective in Eq. 55 changes the $\sigma$-coupling, it does not significantly detriment variance reduction.

**How important are the approximations?** This section has provided comprehensive details about the series of approximations required to make the GRF variance reduction OT problem tractable and solve it efficiently. We close by emphasising that, even if these approximations mean that we are not *exactly* solving the original OT problem, we clearly nonetheless obtain a computationally efficient

coupling that offers much greater variance reduction compared to heuristically-motivated techniques. Analytic intractability is common in OT (Sec. 3 being an obvious exception), but fortunately this framework also equips us with a powerful arsenal of numerical tools to achieve our goal of finding effective, cheap sample couplings.

### C.3.1  PROOF OF THM. 4.1: TOWARDS AN ANALYTIC SOLUTION TO EQ. 55

To assuage any possible dissatisfaction with the $q_1 = q_2$ approximation in Eq. 56, in this section we discuss progress towards solving the optimisation problem in Eq. 55 *exactly*. In particular, we prove Thm. 4.1: that it can be solved with high probability and

1. at a time complexity independent from the dimensionality of vectors $\{\widehat{\psi}(q^{(i,j)})\}_{q=1}^n$ (at the cost of a one-time pre-processing), or
2. in polynomial time in the special case $i = j$ (under mild assumptions about the distribution of the averaged projection vectors $\{\widehat{\psi}(q^{(i,i)})\}_{q=1}^n$).

This is possible because of the extra structure in the quadratic matching problem. A general efficient solution remains (for now) out of reach, but we hope this progress will spur further research in this direction. Note especially that independence of the time complexity from the dimensionality of $\{\widehat{\psi}(q^{(i,i)})\}_{q=1}^n$ means that the difficulty of the optimisation does not depend on the number of nodes $N$: a very convenient property for large graphs.

*Proof.* First, for clarity of exposition, we will introduce more convenient notation and important definitions. For a given $\epsilon > 0$ and a vector $\mathbf{v} \in \mathbb{R}^d$, define the set $N_\epsilon(\mathbf{v}) = \{\mathbf{w} \in \mathbb{R}^d : |\alpha_{\mathbf{v},\mathbf{w}} - \frac{\pi}{2}| \leq \epsilon\}$, where $\alpha_{\mathbf{v},\mathbf{w}}$ is an angle between $\mathbf{v}$ and $\mathbf{w}$. Then define the property of $\epsilon$-*separation* as follows.

**Definition C.1.** *For a given $\epsilon > 0$, a set of vectors $\mathcal{V}$ and a vector $\mathbf{v} \in \mathcal{V}$, we say that $\mathbf{v}$ is $\epsilon$-separated in $\mathcal{V}$ if*

$$N_\epsilon(\mathbf{v}) \cap \bigcup_{\mathbf{x} \in \mathcal{V} \setminus \{\mathbf{v}\}} N_\epsilon(\mathbf{x}) = \emptyset. \tag{57}$$

Simplifying notation, we can rewrite Eq. 55 as the following optimisation problem:

$$\sigma^* = \mathrm{argmin}_\sigma \sum_{q_1, q_2 \in \{1,\dots,n\}} [(\mathbf{v}_{q_1}^{(i)} + \mathbf{v}_{\sigma(q_1)}^{(i)})^\top (\mathbf{v}_{q_2}^{(j)} + \mathbf{v}_{\sigma(q_2)}^{(j)})]^2, \tag{58}$$

where $\sigma$ is the permutation of the sequence $(1, 2, \dots, n)$ and $\mathbf{v}_q^{(i,j)} := \widehat{\psi}(q^{(i,j)})$. With every permutation $\sigma$ of the sequence $(1, \dots, n)$, we will associate the perfect matching $M(\sigma)$ in its corresponding bipartite graph with monochromatic classes of size $n$ each. Denote

$$\mathbf{f}_{M(\sigma)}^{(i)} := \sum_{e \in M(\sigma)} \mathbf{u}_e^{(i)}, \tag{59}$$

where $\mathbf{u}_e$ is given by

$$\mathbf{u}_{(k,\sigma(k))}^{(i)} = \mathrm{vec}\left[(\mathbf{v}_k^{(i)} + \mathbf{v}_{\sigma(k)}^{(i)}) \otimes (\mathbf{v}_k^{(i)} + \mathbf{v}_{\sigma(k)}^{(i)})\right]. \tag{60}$$

Here, $\otimes$ represents the outer product and vec is the 'vectorising' operation that flattens its input to a vector. Note that that $\mathbf{u}_{(k,\sigma(k))} \in \mathbb{R}^{N^2}$.

It is straightforward to see that, in this notation, the optimisation problem becomes

$$\sigma^* = \mathrm{argmin}_\sigma {\mathbf{f}_{M(\sigma)}^{(i)}}^\top \mathbf{f}_{M(\sigma)}^{(j)}. \tag{61}$$

We will now discuss progress towards solving this efficiently.

**Making the problem independent of $N$.** Let us first show how we can make the optimisation problem independent of the number of nodes $N$ with high probability, at the cost of a one-time pre-processing. Our basic approach is to conduct *dimensionality reduction* of the vectors $\mathbf{u}_e^{(i,j)}$ via the Johnson-Lindenstrauss Transform (JLT) (Dasgupta and Gupta, 2003). Specifically, we compute

$$\widehat{\mathbf{u}}_{(k,\sigma(k))}^{(i,j)} = \frac{1}{\sqrt{r}} \mathbf{G} \mathbf{u}_{(k,\sigma(k))}^{(i,j)}, \tag{62}$$

where $r \in \mathbb{N}$ denotes the number of random projections and $\mathbf{G} \in \mathbb{R}^{r \times N^2}$ is a Gaussian matrix with entries taken independently at random from $\mathcal{N}(0, 1)$. From the well-known properties of JLT (Dasgupta et al., 2010), we have that

$$\mathbb{E}[(\widehat{\mathbf{u}}_{(k,\sigma(k))}^{(i)})^\top \widehat{\mathbf{u}}_{(k,\sigma(k))}^{(j)}] = (\mathbf{u}_{(k,\sigma(k))}^{(i)})^\top \mathbf{u}_{(k,\sigma(k))}^{(j)} \tag{63}$$

since it is unbiased. We also have that

$$1 - \epsilon \leq \frac{\|\widehat{\mathbf{u}}_{(k,\sigma(k))}^{(i)} \pm \widehat{\mathbf{u}}_{(k,\sigma(k))}^{(j)}\|_2^2}{|\mathbf{u}_{(k,\sigma(k))}^{(i)} \pm \mathbf{u}_{(k,\sigma(k))}^{(j)}|_2^2} \leq 1 + \epsilon, \tag{64}$$

with $r = \mathcal{O}(\frac{\log(n)}{\epsilon^2})$ random projections, with probability $p = 1 - \text{neg}(n)$. Here, neg denotes the negligible function of $n$.

Eq. 64 describes the ability of the JLT to (approximately) preserve $L_2$-norms. The analogous property for preservation of dot-products also holds, which of interest to us given Eq. 61. It follows because for any $\mathbf{x}, \mathbf{y} \in \mathbb{R}^{N^2}$,

$$\mathbf{x}^\top \mathbf{y} = \frac{\|\mathbf{x} + \mathbf{y}\|^2 - \|\mathbf{x} - \mathbf{y}\|^2}{4}. \tag{65}$$

The properties above lead us directly to an algorithm to decouple the time complexity of the optimisation from $N$. One simply replaces every $\mathbf{u}_{(k,\sigma(k))}^{(i,j)}$ with its corresponding dimensionality-reduced $\widehat{\mathbf{u}}_{(k,\sigma(k))}^{(i,j)}$ for $r = O(\frac{\log(n)}{\epsilon^2})$ and $\epsilon > 0$ small enough. Then, by the concentration result listed above and the union bound over all pairs $(i, j)$, we conclude that the optimal $\sigma$ (potentially not unique) will remain optimal after applying the JLT (since all dot-products are approximately preserved). This completes the analysis. We stress that $r$ does *not* depend directly on the dimensionality of $\mathbf{u}_{(k,\sigma(k))}^{(i,j)}$, and therefore on $N$.

**Polynomial in $n$ algorithm for $i = j$.** We can make further progress in the special case that $i = j$, i.e. if we are minimising the variance of diagonal kernel entries $k(v_i, v_i)$. Denote by $T(n)$ time complexity of an efficient polynomial-time algorithm for solving the minimum weights matching problem on bipartite graphs with monochromatic parts of size $n$ each. The following is true.

**Lemma C.2.** *If Eq. 58 has a unique solution $\sigma^*$ and furthermore $\mathbf{f}_{M(\sigma^*)}$ is $\epsilon$-separated in $\mathcal{F} = \{\mathbf{f}_{M(\sigma)} : \sigma \in S_n\}$ for some $0 < \epsilon < \frac{\pi}{2}$ (where $S_n$ denotes the set of all permutations of the sequence $(1, ..., n)$), then, for any $k > 0$, $\sigma^*$ can be found in time $k \cdot T(n)$ with probability $1 - (1 - p_\epsilon)^k$. Here, $p_\epsilon$ is given by the following formula:*

$$p_\epsilon = \mathbb{P}\left[\left|\arccos\left(\frac{g_1}{\sqrt{g_1^2 + ... + g_{N^2}^2}}\right) - \frac{\pi}{2}\right| \leq \epsilon\right], \tag{66}$$

*with $\mathbf{g} \in \mathbb{R}^{N^2}$ a random isotropic vector.*

*Proof of Lemma C.2.* If $i = j$, the expression in Eq. 61 is simply $\|\mathbf{f}_{M(\sigma)}\|_2^2$. Thus, the goal is to find a permutation $\sigma$ that minimises the $L_2$-norm of the vector $\mathbf{f}_{M(\sigma)}$. We propose the following algorithm. At every iteration, replace each vector $\mathbf{u}_e$ with the projection $u_e = \mathbf{u}_e^\top \mathbf{g}$ for $\mathbf{g} \in \mathcal{N}(0, \mathbf{I}_{N^2})$. Note that, from $\epsilon$-separability, we have that if $\mathbf{g} \in N_\epsilon(\mathbf{v})$, then the permutation $\sigma$ corresponding to the minimum weight matching in the bipartite graph with weights given by $u_e$ is also equal to $\sigma^*$. This captures the intuition that the shortest vector will have the smallest projection on a random vector that is almost perpendicular to it, provided none of the other vectors are *also* almost perpendicular to this projection direction. From the fact that $\mathbf{g}$ is taken from the isotropic distribution, we know that the probability that $\mathbf{g} \in N_\epsilon(\mathbf{v})$ is exactly $p_\epsilon$, where $p_\epsilon$ is a geometrical constant that depends only on $N$. Our algorithm solves the projected minimum weight matching problem (in time $T(n)$) at every iteration and stores the solution $\sigma$. Vectors $\mathbf{g}$ at different iterations are chosen independently. After $k$ iterations, the algorithm computes the original objective for every previously-obtained solution and selects the one with the smallest value. From the above, this returns the optimal permutation $\sigma^*$ with probability $1 - (1 - p_\epsilon)^k$, which completes the proof. $\square$

Having considered both claims, the proof of Thm. 4.1 is complete. $\square$

# D   ASYMPTOTIC OPTIMALITY OF PERMUTATION DENSITIES FOR VARIANCE REDUCTION

In Sec. 4.1 of the main text, we alluded to the choice of permutation densities $p_\sigma(x, y) := n1_{\sigma(\lceil nx \rceil) = \lceil ny \rceil}$ depicted in Fig. 2 being not only convenient (in terms of ease of sampling and optimising), but also well-motivated by results from OT theory. Here, we make this statement concrete, showing that in the limit of infinite permutation order $n$ this class contains the solution to a broad range of OT problems.

**Theorem D.1** (Asymptotic optimality of permutation densities)**.** *Consider the class of joint measures* $\Lambda^{(\sigma)} := \{\mu^{(\sigma)}\}$ *specified by permutations $\sigma$ of order $n$, given by permutation densities $p_\sigma$ pushed forward coordinate-wise using the (left-continuous) inverse CDF $F_\eta^{-1}$. Suppose also that the marginal measure $\eta$ is absolutely continuous with respect to Lebesgue measure. Consider also a* continuous *function $f : \mathbb{R} \to \mathbb{R}$ whose expectation is to be estimated using $m = 2$ coupled samples. In the limit $n \to \infty$, the class $\Lambda^{(\sigma)}$ contains measures that converge weakly to the optimal transport plan – that is, the sample coupling which provides the smallest possible estimator variance.*

*Proof.* Consider the set $\{u_i := \frac{1}{n}\left(i - \frac{1}{2}\right)\}_{i=1}^n$ for some $n \in \mathbb{N}$. Consider also a random variable $X$ with measure $\eta \in \mathcal{P}(\mathbb{R})$ and corresponding distribution function $F$. Define its left-continuous inverse $F^{-1}(y) := \inf\{x \in \mathbb{R} : F(x) \geq y\}$. Given some permutation order $n \in \mathbb{N}$, consider the set

$$\left\{ x_i := F^{-1}(u_i) = F^{-1}\left(\frac{i - \frac{1}{2}}{n}\right) \right\}_{i=1}^n \tag{67}$$

which we will refer to as the *quantile midpoints*. Then consider the discrete measure

$$\eta_n := \frac{1}{n} \sum_{i=1}^n \delta_{x_i} \tag{68}$$

where $\delta_{x_i}(A) = 1$ if $x_i \in A$ and 0 otherwise (with $A \subset \mathbb{R}$). Denote by $F_n$ the distribution associated with $\eta_n$. Clearly, $\eta_n \to \eta$ weakly as $n \to \infty$ since $F_n(x) \to F(x)$ for any continuity point $x$ of $F$ (Billingsley, 2013). In this sense, the measure $\eta_n$ is a discrete approximation to $\eta$.

Now consider the variance reduction OT problem for estimating $\mathbb{E}_{\omega \sim \eta}[f(\omega)]$ with $m = 2$ samples,

$$\mu^* = \arg\min_{\mu \in \Lambda_2(\eta)} \left[ \mathbb{E}_{(\omega_1, \omega_2) \sim \mu} f(\omega_1) f(\omega_2) \right]. \tag{69}$$

Solving this analytically for arbitrary $f$ is not in general tractable. On the other hand, if we instead take the discretised marginals $\eta_n$, we must solve

$$\mu_n^* = \arg\min_{\mu \in \Lambda_2(\eta_n)} \left[ \mathbb{E}_{(\omega_1, \omega_2) \sim \mu} f(\omega_1) f(\omega_2) \right]. \tag{70}$$

This special case of discrete marginal measures where all points have equal mass can be solved *exactly*. Our discussion is based on the proof outlined in the introduction to Villani (2021), to which the interested reader is directed for full details.

**OT with equal-mass discrete marginal measures.** Any measure in $\Lambda_2(\eta_n)$ can be represented as a bistochastic $n \times n$ matrix $\mathbf{B} = [B_{ij}]_{i,j=1}^N \in \mathcal{B}_n$, meaning that all $B_{ij}$ are nonnegative and satisfy

$$\sum_i B_{ij} = 1 \,\forall\, j; \quad \sum_j B_{ij} = 1 \,\forall\, i. \tag{71}$$

Therefore, the Kantorovich problem reduces to

$$\inf_{B \in \mathcal{B}_n} \left[ \frac{1}{n} \sum_{ij} B_{ij} C_{ij} \right] \tag{72}$$

where we have encoded the transport costs in the cost matrix $\mathbf{C} = [f(x_i)f(x_j)]_{i,j=1}^n$. $B_{ij}$ is interpreted as the mass of the singleton $(x_i, x_j)$. This is a linear minimisation problem on a convex bounded set. It is well known that a solution always exists and corresponds to the convex hull of optimal permutation matrices. In more detail: by Choquet's theorem the problem admits solutions which are at the extremal points of $\mathcal{B}_n$ (points that cannot be written as a nontrivial linear combination

of other points in $\mathcal{B}_n$). By Birkhoff's theorem these are exactly the permutation matrices $\mathbf{B}^{(\sigma)} := [\delta_{\sigma(i)j}]_{i,j=1}^n$ with $\sigma \in S_n$. So the optimal discrete coupling is the convex hull of the discrete joint measures corresponding to the set of optimal permutations. See the work of Villani (2021) for further details. Choosing just one of these optimal permutations $\sigma$ (since any convex combination of the corresponding measures will give the same amount of variance reduction), we have that

$$\mu_n^* = \mu_n^{(\sigma)} := \sum_{i=1}^n \frac{1}{n} \delta_{(x_i, x_{\sigma(i)})}. \tag{73}$$

**Stability of OT plans.** Another important result in optimal transport is the *stability of transference plans*: namely, that if $\eta_n \to \eta$ weakly then the optimal coupling $\pi_n \in \Lambda_2(\eta_n)$ also converges to the optimal $\pi \in \Lambda_2(\eta)$ weakly provided the cost function $c(X_i, X_j)$ (in our case $f(X_i)f(X_j)$) is continuous and

$$\limsup_{n \to \infty} \int c(x_1, x_2) \mathrm{d}\pi_n(x, y) < \infty. \tag{74}$$

This important observation underpins the effectiveness of numerical approximations to optimal transport. We direct the reader to Thm. 1.7.2 by Panaretos and Zemel (2020) for a proof sketch and Thm. 5.20 by Villani et al. (2009) for more detailed exposition. It follows that $\mu_n^*$ converges weakly to the true optimal coupling $\mu^*$.

**The OT plan is (asymptotically) in our search class.** Lastly, we have that our search class of couplings $\Lambda^{(\sigma)}$ (corresponding to permutation densities pushed forward by $F_\eta^{-1}$) contains measures that converge in distribution to $\mu_n^{(\sigma)}$ when $n \to \infty$. In this mathematical sense, the asymptotic limit the class of couplings amongst which we optimise includes measures that give the greatest possible variance reduction: roughly speaking, our method is 'asymptotically optimal'. This is intuitive because as the order of the permutation grows each 'tile' of nonzero density narrows and can be increasingly well-approximated by a delta function.

To see this, consider the measure on $[0, 1]^2$ described by the permutation density $p_\sigma(x, y) = n \mathbf{1}_{\sigma(\lceil nx \rceil) = \lceil ny \rceil}$. Consider also the discrete measure on $[0, 1]^2$ given by $\frac{1}{n} \sum_{i=1}^n \delta_{u_i}$ with the set $\{u_i\}_{i=1}^n$ the quantile midpoints defined previously. These measures converge in distribution (their corresponding joint CDFs can differ by at most $\frac{1}{n}$ at any point, which goes to 0 as $n \to \infty$). They will also converge in distribution when pushed forward coordinate-wise by $F_\eta^{-1}$ by continuity of $\eta$. It follows that at asymptotic $n$ $\mu_n^{(\sigma)}$ and $\mu^{(\sigma)}$ converge in distribution, which completes the proof. $\quad\square$

We remark that not all the assumptions in Thm. D.1 hold in the GRF setting. In particular, neither the marginal measure $\eta$ nor the function $f$ is continuous. The intention of including Thm. D.1 is to build intuition for the reader and provide some motivation for our choice of permutation densities. We defer a rigorous investigation into relaxing these restrictive assumptions – a tough measure-theoretic problem – to future work.

# E  ESTIMATING PAGERANK

In this appendix, we demonstrate a further use of the variance reduction OT techniques developed for graph random walks in Sec. 4: estimating the *PageRank* vector (Page et al., 1998). This popular measure of the relative importance of the nodes $\mathcal{N}$ of a graph is expensive to compute exactly so is often estimated by sampling random walks (Fogaras et al., 2005). We will show that OT can be used to find a coupling between the walks to reduce the estimator variance, demonstrating the versatility of our approach beyond improving RFs.

## E.1  SETUP: ESTIMATING PAGERANK WITH RANDOM WALKS

The PageRank vector is the stationary distribution of Markov chain whose state space is the set of all graph nodes $\mathcal{N}$, with a transition matrix

$$\widetilde{\mathbf{P}} := (1 - p_{\text{halt}})\mathbf{P} + \frac{p_{\text{halt}}}{N}\mathbf{E}. \tag{75}$$

Here, $p_{\text{halt}} \in (0, 1)$ is a scalar, $N$ is the number of nodes and $\mathbf{E} = [1]_{i,j \in \mathcal{N}}$ is a matrix whose entries are all ones. $\mathbf{P}$ is the transiton matrix of a simple random walk,

$$P_{ij} = \begin{cases} \frac{1}{d_i} & \text{if } (i, j) \in \mathcal{E} \\ 0 & \text{otherwise} \end{cases} \tag{76}$$

with $d_i$ the degree of the $i$th node. Since $\widetilde{\mathbf{P}}$ is stochastic, aperiodic and irreducible, we define the unique PageRank vector $\boldsymbol{\rho} \in \mathbb{R}^N$:

$$\boldsymbol{\rho}^\top \widetilde{\mathbf{P}} = \boldsymbol{\rho}^\top, \quad \boldsymbol{\rho}^\top \mathbf{1} = 1, \tag{77}$$

where we normalised the sum of vector entries to $1$. Computing $\boldsymbol{\rho}$ analytically is expensive. Rearranging and Taylor expanding $(1 - (1 - p_{\text{halt}})\mathbf{P})^{-1}$, it is straightforward to see that

$$\boldsymbol{\rho}_i = \frac{p_{\text{halt}}}{N} \sum_{j \in \mathcal{N}} \sum_{k=0}^{\infty} (1 - p_{\text{halt}})^k \mathbf{P}_{ji}^k. \tag{78}$$

This is simply a sum over all walks from each of the graph nodes $v_j$ to node $v_i$, weighted by their respective probabilities – that is, the expected number of random walkers ending at node $v_i$ if they terminate with probability $p_{\text{halt}}$ at every timestep – which invites the estimator proposed by Fogaras et al. (2005),

$$\widehat{\boldsymbol{\rho}}_i = \frac{1}{Nm} \sum_{v_j \in \mathcal{N}} \sum_{n=1}^{m} \mathbb{I}[n\text{th walk from node } v_j \text{ terminates at node } v_i]. \tag{79}$$

An interesting and practically important question is whether the variance of the estimator $\widehat{\boldsymbol{\rho}}$ can be improved by coupling random walks. As with GRFs, this can be achieved using antithetic termination (Reid et al., 2024c) (see Sec. C.2). However, we will see that our OT length coupling approach does equally well or better.

### E.2 OT FORMULATION OF PAGERANK VARIANCE REDUCTION

Given $m = 2$ walks, the variance of the PageRank estimator $\text{Var}(\widehat{\boldsymbol{\rho}}_i)$ depends on the quantity

$$\mathbb{E}(\widehat{\boldsymbol{\rho}}_i^2) = \mathbb{E}\left[ \frac{1}{N^2 m^2} \sum_{v_j, v_k \in \mathcal{N}} \sum_{n_1, n_2 = 1}^{2} \mathbb{I}[n_1\text{th walk from node } v_j \text{ terminates at node } v_i] \right. \tag{80}$$
$$\left. \cdot \mathbb{I}[n_2\text{th walk from node } v_k \text{ terminates at node } v_i] \right].$$

After a few lines of algebra, the OT problem to minimise this quantity is

$$\text{minimise } \mathbb{E}_{(l_1, l_2) \sim \mu} \sum_{v_j \in \mathcal{N}} \mathbb{P}[\text{RW of length } l_1 \text{ from node } v_j \text{ terminates at node } v_i] \tag{81}$$
$$\cdot \mathbb{P}[\text{RW of length } l_2 \text{ from node } v_j \text{ terminates at node } v_i] \text{ for } \mu \in \Lambda_2(\eta_{\text{G}})$$

without making any approximations. $\mathbb{P}(\cdot)$ denotes the probability of the event in brackets. As with GRFs, for tractability we restrict our search space of joint distributions to permutation densities pushed forward coordinate-wise with the geometric distribution inverse CDF, and this becomes

$$\sigma^* = \arg\min_{\sigma} \sum_{v_j \in \mathcal{N}} \sum_{q \in [\![n]\!]} \mathbb{P}[\text{RW with length in } q\text{th quadrant from node } v_j \text{ terminates at node } v_i] \tag{82}$$
$$\cdot \mathbb{P}[\text{RW with length in } \sigma(q)\text{th quadrant from node } v_j \text{ terminates at node } v_i].$$

The probabilities can be efficiently estimated by simulating random walks on the graph and recording where they terminate. This leaves us with a minimum-weights bipartite matching problem which can as usual be solved efficiently with the Hungarian algorithm (Kuhn, 1955).

Note that we made fewer simplifications than for GRFs (Sec. C.1). They only requirements for tractability are restricting the class of considered joints to $\sigma$-couplings and computing MC estimates of the terms in the OT cost matrix. We do not need to e.g. move $\mathbb{E}_{\text{dirs}}$ inside a square or approximate a quadratic matching problem by a linear-weights counterpart.

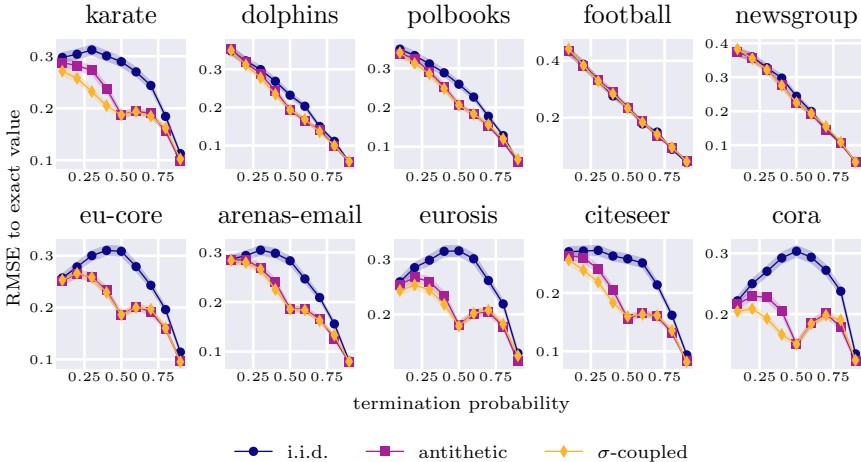

Figure 9: PageRank estimator errors $\|\boldsymbol{\rho} - \widehat{\boldsymbol{\rho}}\|_2$ for a range of real-world graphs when 2 random walkers are taken to be i) i.i.d., ii) antithetic or iii) $\sigma$-coupled. Lower is better. The error is always at least as small and often substantially better when the walkers are coupled using our OT coupling. One standard error is shaded.

### E.3 EMPIRICAL RESULTS FOR PAGERANK

We now provide experiments using our OT coupling to improve the convergence of PageRank estimates. Fig. 9 shows the results. For termination probabilities $p_{\text{halt}} \in \{0.1, 0.2, 0.3, 0.4, 0.5, 0.6, 0.7, 0.8, 0.9\}$, we compute the optimal permutation $\sigma$ by solving the matching problem in Eq. 82 on an Erdős-Rènyi graph with $N = 100$ nodes, taking permutation order $n = 10$ as a hyperparameter. We then test these couplings on a range of real-world graphs (Ivashkin, 2023) of different sizes, plotting the estimator error $\|\boldsymbol{\rho} - \widehat{\boldsymbol{\rho}}\|_2$ (where $\boldsymbol{\rho}$ is the true PageRank vector and $\widehat{\boldsymbol{\rho}}$ is its MC estimate) against the termination probability $p_{\text{halt}}$. We include walkers that are i.i.d., antithetic (Reid et al., 2024c) and $\sigma$-coupled. At every value of $p_{\text{halt}}$, our method performs *at least as well* as the baselines, and in some cases (e.g. on the very large cora graph) it gives much greater variance reduction.

Note that we only consider connected subgraphs and take 1000 repeats for standard errors. Reading the plots from left to right for the top row and then the bottom, the number of nodes are: $\{34, 62, 105, 115, 398, 986, 1133, 1272, 2120, 2708\}$. The shape of the curve and size of the gain from coupling depends on the particular graph being considered. For some graphs it is not possible to obtain a big reduction in PageRank estimator variance by coupling walkers (e.g. football and newsgroup), so neither antithetic termination nor our $\sigma$-coupling can provide a big gain. A detailed investigation of how these observations relate to the mathematical graph structure is deferred to future work.

## F GRFs FOR GPs: ADDITIONAL DETAILS AND EXPERIMENTAL RESULTS

In this appendix, we supplement Sec. 4.2 by providing further experimental results for GRFs, including Gram matrix estimation on a wider range of graphs and a scalable graph-based GP on a different real-world dataset. We also provide technical background and tedious details considered too long for inclusion in the main text.

### F.1 MORE RESULTS FOR GRAM MATRIX ESTIMATION

First, we run the experiment in the first part of Sec. 4.2 for more real-world graphs. We approximate the 2-regularised Laplacian kernel $(\mathbf{I} - \sigma^2 \widetilde{\mathbf{L}})^{-2}$ with a regularisation parameter $\sigma = 1$ and $m = 2$ walkers. Fig. 10 shows the results, with the kernel approximation error normalised by the i.i.d. result (unlike in Fig. 3) for clarity of comparison. The $\sigma$-coupling consistently reduces the estimator

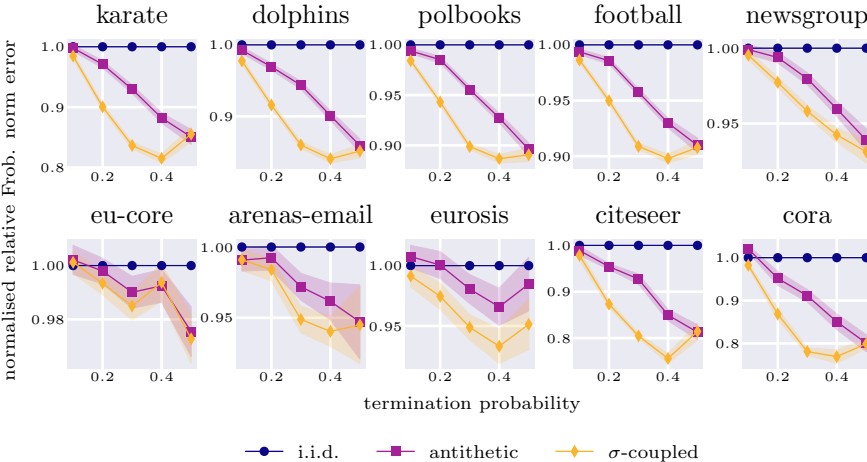

Figure 10: Relative Frobenius norm error $\|\mathbf{K} - \widehat{\mathbf{K}}\|_{\mathrm{F}}/\|\mathbf{K}\|_{\mathrm{F}}$ between true and approximated Gram matrices for different termination probabilities when walkers are i) i.i.d., ii) antithetic (Reid et al., 2024c) or iii) $\sigma$-coupled using our OT approach. All results are normalised by the i.i.d. variance for easy comparison. Lower is better. The $\sigma$-coupling is consistently as good or better across values of $p_{\mathrm{halt}}$ and graph topologies and sizes. One standard error is shaded.

variance, providing equally good or better results compared to the hand-crafted antithetic termination algorithm. The size of the gain depends on the graph structure.

### F.2 SCALABLE GPs ON GRAPHS

**A very short introduction to graph-based GPs.** In certain applications of Gaussian processes, kernels based on Euclidean distance are unsuitable – e.g. when predicting traffic congestion since locations that are spatially close may not be connected by roads. Kernels defined on the nodes of a graph $\mathcal{G}$ may be more appropriate, giving structured covariance matrices that encode the dependence between pairs of vertices. We can then perform inference and make predictions using *GPs on graphs* (Borovitskiy et al., 2021; Zhi et al., 2023).

Like their Euclidean counterparts, exact GPs on graphs suffer $\mathcal{O}(N^3)$ time complexity, making them impractical for very large structures. Techniques to improve their scalability include graph Fourier feature approximations, which approximate the kernel matrix using a truncated eigenvalue expansion computed using the Lanczos algorithm. The price of increased speed is that our kernel estimate is biased. Another approach is to restrict to kernels whose inverses are sparse, e.g. a Matérn kernel with a small integer $\nu$-parameter, at the expense of lost flexibility (Borovitskiy et al., 2021). In this paper, we have proposed to instead use GRFs, which give an unbiased estimate to the kernel matrix in subquadratic time. GRFs are sparse: intuitively, $\phi_{\mathrm{GRF}}(v_i)$ is constructed by simulating $m$ random walks out of $v_i$ and adding weighted contributions only at the coordinates corresponding to visited nodes. For walks with geometrically-distributed lengths this is typically a small subset of $\mathcal{N}$. Therefore, the matrix $\widehat{\mathbf{K}} \coloneqq [\phi(v_i)^\top \phi(v_j)]_{i,j=1}^N$ is a sparse, unbiased estimate to *any* kernel, not just specific families.[5] We can use established numerical routines for sparse linear algebra to speed up training, providing computational savings on big graphs. This is analogous to the sparse spectrum GPs discussed in App. B (Lázaro-Gredilla et al., 2010); the interested reader might benefit from reviewing this section first.

**Our contribution to scalable graph-based GPs.** Using GRFs for scalable approximate inference on graphs is itself a novel contribution that deserves detailed exploration in future work. Since our central goal is to use perspectives from OT to improve the convergence of GRFs, we confine our focus to the question: *can coupled random walks give more accurate estimates of graph GP posteriors?* We defer a detailed comparison of our method to other scalable graph-based GP techniques to a future self-contained paper.

---

[5] Strictly: any graph node kernel that is expressible as a function of a weighted adjacency matrix.

**Training GRF-GPs: technical details.** Here, provide details to supplement Sec. 4.2. We use mesh graphs made available by Dawson-Haggerty (2023), taking the *faces* as our graph nodes. Reading left to right, the number of nodes are: $894, 999, 1280, 1572, 3838, 8700$. For probabilistic mesh interpolation, we use the diffusion kernel $\mathbf{K} = \kappa \exp(-\gamma^2 \tilde{\mathbf{L}})$. Here, $\tilde{\mathbf{L}}$ is the normalised graph Laplacian (Eq. 48) and $\kappa, \gamma$ are learnable parameters corresponding to the signal variance and kernel lengthscale respectively. The observation noise $\sigma$ is also learnable. For every graph, we construct a kernel estimate by sampling $m = 8$ i.i.d. random walks out of every node. We then train the corresponding vanilla GP by optimising the log marginal likelihood on observed nodes, using the Adam optimiser (Kingma and Ba, 2014) for $1000$ epochs (after which both the objective value and the hyperparameters are empirically found to have converged). We freeze $\kappa, \gamma, \sigma$ and compute the corresponding exact kernel. We approximate this exact kernel using $\{16, 32, 64\}$ random walks that are either i.i.d., antithetic or $\sigma$-coupled. We perform approximate Bayesian inference in each case, computing the accuracy of the kernel estimate, the average test RMSE and the KL divergence to the true posterior distribution (unnormalised by the number of test points). We use permutations of order $n = 50$ and a termination probability $p_{\text{halt}} = 0.4$. As reported in the main body, *coupling the lengths of graph random walkers permits better approximate Bayesian inference with GPs on graphs.*

**Intuition for the effectiveness of coupling lengths for GRFs.** When we couple the lengths of random walks, we 'diversify' the ensemble, sampling walks with a different number of hops. This is a very direct way to ensure we explore the graph more effectively, avoiding (by chance) simulating pairs of very similar walks. It is intuitive that improving 'coverage' of the graph means we estimate the kernel more effectively, letting us model more interactions between different graph nodes. This may explain why our method is so effective in this setting. However, we stress that coupling schemes for estimators defined on discrete spaces like graphs were only introduced very recently (Reid et al., 2024c), so further work is needed for a rigorous understanding of this phenomenon.

### F.3 PROBABILISTIC TRAFFIC INTERPOLATION

In this section, we provide further details about the final experiment of Sec. 4.2, which uses a scalable graph-based GP to predict traffic flow speeds on the highways of San Jose, California.

**Dataset.** Readers are directed to the original paper (Borovitskiy et al., 2021) for exhaustive details about the dataset curation and graph computation. Here, we simply remark that this graph is sufficiently large that exact GP methods become slow ($N = 1016$), and observations are only made at a small subset of the nodes ($325$) so good uncertainty quantification is essential.

**Results for GRFs.** We randomly divide the nodes into training and test datasets of sizes $N_{\text{train}} = 250$ and $N_{\text{test}} = 75$ respectively. As described in App. F.2, we train a scalable GRF-GP with $m = 8$ i.i.d. walkers set up to approximate the diffusion kernel, optimising the training data negative log marginal likelihood. Next, we freeze the kernel and noise parameters, and compare the performance of GRFs with $m \in \{4, 8, 16\}$ i.i.d., antithetic (Reid et al., 2024c) and $\sigma$-coupled walkers. As before, we consider the quality of Gram matrix approximation, the accuracy of predictions (test RMSE) and the quality of the predictive uncertainties (KL divergence to true posterior). Fig. 11 shows the results. $\sigma$-GRFs consistently do best on all metrics, even with a severely restricted sampling budget.

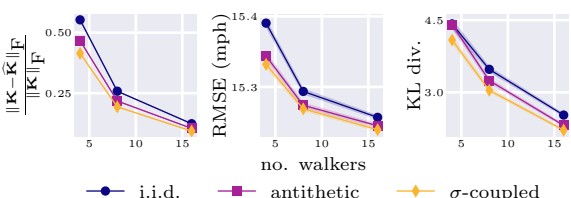

Figure 11: Results for probabilistic traffic interpolation experiment. Plots show the kernel approximation accuracy, test RMSE and KL divergence to the true posterior for graph kernels estimated using i.i.d., antithetic and $\sigma$-coupled GRFs. Lower is better. $\sigma$-coupling gives the best results for approximate inference. One standard error over random draws of walkers is shaded.

## G  ADDENDUM: FURTHER EVIDENCE OF GENERALISABILITY

Here, we give further evidence of the generalisability of our scheme. We show that our PNC algorithm (Def. 3.3) provides strong empirical improvements for the convergence of estimates of other radial basis function (RBF) kernels, beyond the Gaussian example considered in Sec. 3. In particular, we consider approximating the popular *Matérn class* of covariance functions (Williams and Rasmussen, 2006), defined by

$$k_{\text{Matérn}}(\boldsymbol{x}_i, \boldsymbol{x}_j) := \frac{2^{1-\nu}}{\Gamma(\nu)} \left( \frac{\sqrt{2\nu}\|\boldsymbol{x}_i - \boldsymbol{x}_j\|_2}{l} \right) K_\nu \left( \frac{\sqrt{2\nu}\|\boldsymbol{x}_i - \boldsymbol{x}_j\|_2}{l} \right) \tag{83}$$

with positive parameters $\nu$ and $l$, where $K_\nu$ is a modified Bessel function. It is well known that this covariance function has spectral density

$$S(\boldsymbol{\omega}) = \frac{2^d \pi^{d/2} \Gamma(\nu + \frac{d}{2})(2\nu)^\nu}{\Gamma(\nu) l^{2\nu}} \left( \frac{2\nu}{l^2} + 4\pi^2 \boldsymbol{\omega}^\top \boldsymbol{\omega} \right)^{-(\nu + d/2)} \tag{84}$$

in $d$ dimensions. We can sample frequency vectors $\{\boldsymbol{\omega}_i\}_{i=1}^m$ from this density and use them to construct RFFs (Eq. 3), thereby unbiasedly approximating the Matérn kernel.

Clearly, $S(\boldsymbol{\omega})$ is isotropic, so it is possible condition that frequency vectors are mutually orthogonal. Moreover, we are free to couple their norms using our PNC scheme, though now each obeys a different marginal distribution compared to the Gaussian case.

**Experimental results.** We construct RFFs to approximate the Matérn kernel with smoothness parameters $\nu \in \{\frac{1}{2}, \frac{3}{2}, \frac{5}{2}\}$, taking random frequencies marginally distributed as in Eq. 84. As before, we consider sampling frequencies that are (i) i.i.d., (ii) orthogonal with independent norms, and (iii) orthogonal with PNC norms. For the dataset, we randomly generate $N = 64$ vectors $\boldsymbol{x}_i \sim \mathcal{N}(0, \mathbf{I}_d/d)$ of dimensionality $d = 8$. For simplicity, we set the kernel lengthscale $l = 1$. Fig. 12 shows the error on the Gram matrix approximation as the number of features grows. Even for this different kernel and across different choices for the smoothness parameter $\nu$, PNC continues to substantially suppress kernel estimator variance. This provides strong evidence that our OT-driven norm coupling scheme is generalisable across RBF kernels, and is not specific to the Gaussian example.

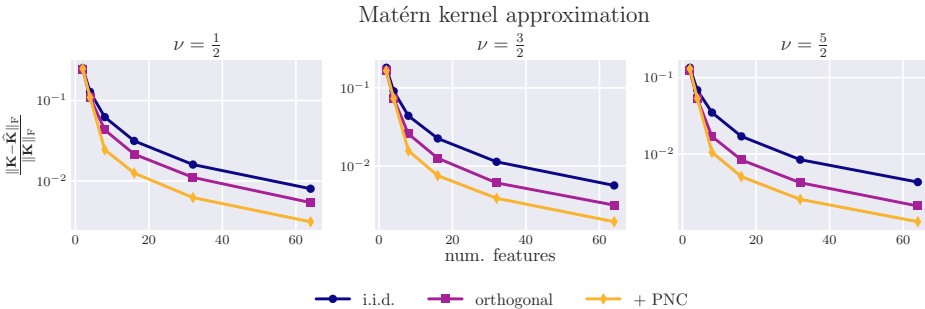

Figure 12: Kernel estimation RMSE of the Matérn kernel with different smoothness parameters $\nu$, taking different coupling schemes for the frequency vectors. Once again, we see that PNC provides substantial variance reduction, outperforming all the baselines.

