# OpenReview forum: "Variance-Reducing Couplings for Random Features"
_ICLR.cc/2025/Conference — ICLR 2025 Poster_

### Official Review · Reviewer_L1dC · 2024-10-29

**Soundness:** 3
**Presentation:** 3
**Contribution:** 3
**Rating:** 6
**Confidence:** 3

**Summary:**

This paper proposes an efficient construction of random features for kernel methods. Motivated by the optimal transport, it finds couplings to reduce the variance of random features defined on both Euclidean and discrete input spaces. Numerical experiments are conducted and theoretical properties are analyzed.

**Strengths:**

1. The connection between variance reduction of RFs and OT is clearly presented.
2. RFs for different settings including RFFs, RLFs and GRFs are deeply studied.
3. A through simulation study has been conducted and some interesting phenomenons show up in the numerical experiments.

**Weaknesses:**

Please see following questions.

**Questions:**

1. In Lemma 3.1, the forms of c_{RFF} and c_{RLF} are very complicated and the solution \mu^* seem to depend on x and y. In Theorem 3.2 with m=2, the solution seems to be independent of x and y. Is it the case for m=2? Or the solution will always be irrelevant to x and y no matter how large m is? In addition, it will be better to give more explanation about the statement ``provided z is sufficiently small". For example, how small z is w.r.t. the order? How to check if z is small enough in practice?

2. In Section 3.3, what is the definition of RMSE? Is it the point-wise RMSE or the Frobenius norm error for the gram matrix approximation?  What is the choice of d and how RMSEs change with d empirically?

3. Though the paper provides that Theorem that the pairwise norm-coupled RFs is guaranteed to be lower than orthogonal RFs with independent norms, it will be better to provide an explicit error upper bound for the variance, which will be easier to see the dependence of the error on different parameters.

4. It is interesting to see that the variance reduction alone cannot guarantee an improvement in the downstream tasks. I am wondering is it the case for most datasets considered in Table 1? If so, the application of this RFs may be somehow limited.

5. In page 8, the paper proposes to ``move Edirs inside the square" to make the computation feasible. It will be better to discuss the effect of such adjustment.  By moving the expectation, the objective function indeed changes and solving the adjusted objective function does not necessarily reduce the variance?

6. In the numerical experiments w.r.t. RGFs, the proposed method performs the best also in the downstream tasks for different datasets, which shows a different performance from RFFs and RLFs.  Is there any intuition behind this?

---

> ### Author Response · Authors · 2024-11-15
> **Thanks for the review**
>
> We thank the reviewer for their comments on the manuscript. We are pleased that they find the paper clear and thorough, and that they find the numerical experiments interesting. We address all their questions in detail below.
>
> 1. *Data dependence of the optimal coupling*. Thanks for the great question. In general, the cost function will depend on $(x,y)$ so the optimal coupling may indeed depend on the data. Remarkably, as the reviewer correctly identifies, for RFFs and RLFs with $m=2$ the optimal coupling turns out to be *independent* of $(x,y)$, and therefore independent of the data distribution. For general $m$, the exact solution to the (generically intractable) multi-marginal OT problem may indeed depend on the data. However, **our proposed algorithm (Def 3.3) is data-independent, guaranteed to be better than i.i.d., and empirically still close to optimal** (see Sec 3.2 for numerical evidence).
> 2. *Meaning of ‘provided $z$ is sufficiently small*. Thanks for the question. **We precisely characterise what we mean by ‘provided $z$ is sufficiently small’ in Lemma A.6** (line 990). The discussion is mathematically involved, so we omit it from the main text to preserve its flow and readability. Eq 32 and the following discussion explains how to choose $z$ to guarantee that it is small enough. We invite the reviewer to read this section for full details. Note that in our experiments we find that this condition tends not to matter in practice: for a variety of real-world datasets, our algorithm provides substantial variance reduction for RFFs (see Table 1). We do not expect practitioners to actually need to check Eq. 32 when using the algorithm; it is a technical detail.
> 3. *RMSE and $d$*. By RMSE, we mean the Frobenius norm error between the true and approximate Gram matrices. We will make this clearer – thanks. $d$ is not chosen by the practitioner; it is the dimensionality of the data. The standard UCI datasets we use have a range of dimensionalities. We do not see a consistent trend for how this modifies the effectiveness of our coupling, which depends more subtly on the data distribution.
> 4. *Bound for the difference in variance*. Thanks for the great suggestion. Using Lemma A.6, it is actually very straightforward to extend Eq. 32 to show that the reduction in variance is lower bounded by  $z^4 c / 32 \Gamma(2+d/2) - 2g(z)$, where $c$ is defined in Eq. 30 and $g(z)$ is defined on line 1019. We have added this comment to the paper. Thanks again for prompting it.
> 5. *Downstream tasks for Table 1*. We respectfully invite the reviewer to inspect App. B.2, which gives a detailed account of the experiment they suggest here. They are right to note that the fact that variance reduction may not always improve downstream performance, and that OT couplings should be optimised for a different objective, is an **important conclusion of our work**. Indeed, we already show how a different OT coupling can improve the accuracy of Performer ViTs by **+0.8%** (see Table 2). We hope our paper will inspire future research in this crucial new direction.
> 6. *Moving $\mathbb{E}_\textrm{dirs}$ inside the square*. Thanks for the great question. We agree that moving the expectation inside the square modifies the objective. This is one of several simplifications needed to make the problem tractable, **all discussed in detail in App. C.3**. This particular step is essential for scalability – without it, we cannot write the objective in a way that can be efficiently estimated using random walks, so the problem becomes intractable. However, we emphasise that, even with these approximations, our method still achieves **SOTA variance reduction for length-coupled GRFs compared to previously-proposed alternatives** [3].
> 7. *Intuition for GRFs*. Thanks for the great question. When we couple the lengths of random walks, we ‘diversify’ the ensemble, sampling walks with a different number of hops. This is a very direct way to ensure we explore the graph more effectively, avoiding (by chance) simulating pairs of very similar walks. It is intuitive that improving ‘coverage’ of the graph means we estimate the kernel more effectively, letting us model more interactions between different graph nodes. The benefits of coupling the norms or orthogonal frequencies in RFFs and RLFs – effectively modelling the lengthscales of perpendicular components in the data – are less clear. We hope this helps build intuition for the difference between these cases, and have added this discussion to the paper.
>
> We again thank the reviewer. Having answered all their questions (including adding a bound for the amount of variance reduction), we very much hope they will consider raising their score. We warmly invite them to respond if anything remains unclear.
>
> CONTINUED BELOW.

---

> > ### Author Response · Authors · 2024-11-15
> > **Thanks for the review (part 2)**
> >
> > _______________
> > [1] Simplex Random Features, Reid et al., ICML 2023,  https://doi.org/10.48550/arXiv.2301.13856
> > [2] Orthogonal Random Features, Yu et al., NeurIPS 2016,  https://doi.org/10.48550/arXiv.1610.09072.
> > [3] Quasi Monte Carlo Graph Random Features, Reid et al., NeurIPS 2023,  https://doi.org/10.48550/arXiv.2305.12470

---

> > > ### Author Response · Authors · 2024-11-18
> > > **Manuscript updated**
> > >
> > > This is a brief note to let the reviewer know that we have uploaded a new version of the manuscript. Changes are marked in red. They may be pleased to see that we have clarified the meaning of RMSE (line 266) and added the lower bound on the variance reduction due to coupling (Eq. 33). Following their comments, we also added a new section providing intuition for the good performance of coupled GRFs, beginning on line 2013. Thanks for the suggestions. Having made these improvements and clarified all other questions and concerns raised above, we again respectfully request that they consider raising their score.

---

> > > > ### Comment · Reviewer_L1dC · 2024-11-20
> > > > **Thanks for the response**
> > > >
> > > > Thank the authors for the effort in addressing my comments. I will keep my positive rate for the work.

---

> > > > > ### Author Response · Authors · 2024-11-20
> > > > > **Anything else?**
> > > > >
> > > > > We thank the reviewer for the response. We are are very happy that they consider all their comments addressed.
> > > > >
> > > > > **Is there anything else that we may clarify for them, or any other additions they would like to see in the manuscript?** It would be fantastic if they would consider raising their score. We will be happy to discuss any further improvements they deem necessary.

---

> > > > > > ### Author Response · Authors · 2024-11-24
> > > > > > **Discussion period drawing to a close: any further questions?**
> > > > > >
> > > > > > As the discussion period draws to a close, this is one final polite reminder that we will be happy to answer any further questions the reviewer may have. We believe that we have resolved all their concerns, having added an extra section providing intuition for the good performance of coupled GRFs and a lower bound on the amount of variance reduction. **If they consider all their comments addressed, we respectfully ask them to consider raising their score to reflect this**. We thank the reviewer once more for their feedback.

---

### Official Review · Reviewer_huMM · 2024-10-30

**Soundness:** 3
**Presentation:** 2
**Contribution:** 2
**Rating:** 6
**Confidence:** 3

**Summary:**

The paper considers the problem of finding a suitable joint measure for random features in a kernel evaluation by applying a variance reduction principle. The application of such a principle in general results in a coupling of the features, expressed, for example, by a constraint on their modulus in the case of features in the Euclidean case. The principle is given in terms of an optimal transport formulation and its application is exemplified by a number of experiments on both Euclidean kernels and graph kernels. The authors show that if on the one hand, a smaller kernel variance is obtained as expected, other quantities might benefit from different strategies, encouraging a rethinking of the relevance of kernel variance reduction.

**Strengths:**

The paper aims to solve the problem of minimizing the variance of kernel estimators by random features by mapping the problem in an optimal transportation (OT) problem. By using OT concepts, (sub-optimal but interesting) coupling strategies are identified in both the Euclidean case and in a discrete case, namely graph kernels. The paper links therefore two ideas (kernel random features and optimal transport) that are very popular in the machine learning community, with an interesting digression on graph random features and graph kernels. Moreover, I appreciate the rich numerical exploration of the results, that highlighted the benefits and limitations of the method.

**Weaknesses:**

My main concern is that the proposed OT formulation does not obviously lead to a simple, applicable rule, as shown by the authors themselves. Even in the random Fourier features case, under the simplifying pairwise coupling assumption, guarantees can be proven only for close enough arguments of the kernel function to estimate. The examples discussed and the appendices show that the application of the OT ideas might be cumbersome, not straightforward, and require a certain ingenuity to perform a number of simplifications and approximations (whose effectiveness is only shown *a posteriori*): this comes with no surprise as solving an OT problem is generically hard. More importantly, if the quantities of interest are nonlinear functions of the kernel, counterintuitive results might be found, e.g., *maximizing* the variance might be beneficial. It appears then unclear how to decide *a priori* what the best coupling strategy is. Even if, in principle, one might think to generalize the minimization problem in Eq 2 (so that the functional $\mathcal I(\mu)$ depends on the variance of other target quantities), as the authors themselves explain, the possible complication of such generalization, and its feasibility, seem hard to anticipate.

**Questions:**

Here below some comments and observations:

- After Eq 1, it is not clear if *each* $\phi$ has to be intended depending on $m$ random frequencies (also, $\\{\boldsymbol\omega\\}$ should be replaced with $\\{\boldsymbol\omega_i\\}$: it would be sufficient, for example, to write down $\phi(\boldsymbol x;\boldsymbol\omega_{1:m})$ in place of $\phi(\boldsymbol x)$. At line 048 I am not sure if Eq 1 is a "MonteCarlo estimate" of $k$, as in Eq 1 there is no empirical sampling.
- I might being missing something trivial, but why $\mathcal I(\mu)=\mathbb E_\mu[(\phi^\top\phi)^2]$ and not $\mathcal I(\mu)=\mathbb E_\mu[(\phi^\top\phi)^2]-\mathbb E_\mu[\phi^\top\phi]^2$? I suppose that, given the constraint on the marginals, the second term is irrelevant if $\phi(\boldsymbol x;\boldsymbol\omega_{1:m})^\top\phi(\boldsymbol y;\boldsymbol\omega_{1:m})=\sum_{i=1}^m\psi_i(\boldsymbol x;\boldsymbol\omega_i)^\top\psi_i(\boldsymbol y;\boldsymbol\omega_i)$ for some $\psi_i$, but this might be not true in general. Unless I am wrong, would the more general case lead to a much more complicated treatment? The treatment of RFF and RLF seems to be strongly case-specific, although relevant.
- The discussion of the application to GRFs leads to encouraging results. Do the authors have an intuition on why this is the case? Are such positive performances retained independently from the level of the sparsity of the graph? In general, I was wondering how much the "sparsity" of the underlying space where the kernel lives affects the effectiveness of the method.

---

> ### Author Response · Authors · 2024-11-15
> **Thanks for the review (part 1)**
>
> We thank the reviewer for their comments on the paper. We are pleased that they find the connection between Monte Carlo and optimal transport interesting, and that they appreciate our detailed numerical results. We address all their concerns and questions in detail below.
>
> *'OT does not obviously lead to a simple rule'*. We agree that solving optimal transport problems can in general be very difficult, and that the multi-marginal (MMOT) case is even harder. However, we respectfully bring the reviewer’s attention to the following observations.
>
> 1. *For RFFs and RLFs, there \***is**\* a simple rule*. For $m=2$, we are able to solve the OT problem *exactly* for RFFs and RLFs (asymptotically for the former). Even better, the optimal joint measure is trivial to sample from, and is independent of the data. This is a great example of a lightweight, simple coupling scheme derived using OT, which guarantees substantial (in fact, optimal) variance reduction. Although the MMOT problem is harder, Sec 3.2 provides strong numerical evidence that, in spite of its simplicity, the $m=2$ scheme is still almost optimal in this more general setting. This shows that OT sometimes gives couplings that are simple and work well.
> 2. *Coupled GRFs are cheap and give SOTA variance reduction*. Whilst we agree that the series of approximations needed in Sec. 4.1 may not be obvious at a first glance, we emphasise that the final coupling algorithm is very efficient – one simply simulates random walks on a test graph, uses them to set up a linear weights bipartite matching, and efficiently solves it with the Hungarian algorithm. Once again, the (approximately) optimal measure is very cheap to sample from. Importantly, it also achieves **SOTA variance reduction for GRFs**. Whilst the reviewer says the approximations might take a bit of `ingenuity’, the resulting GRFs algorithm is both practical and effective, across graph topologies and tasks.
> 3. *The counterintuitive results are important*. We agree that the fact that variance reduction may not improve downstream importance is counterintuitive. We believe it to be underappreciated in the literature: an enormous number of papers have been published at top ML conferences seeking to reduce the variance of kernel estimates  [1,2,3,4,5,6,7,8,9,10], but to our knowledge none have yet sought to optimise a *different* property of the coupling like the MSE of the row-normalised attention scores. We agree that more work is needed to fully understand how practitioners should best achieve this and we agree that certain OT objectives will turn out to be intractable, but we have already seen in Table 2 that other simple couplings exist that perform better than previous approaches. In particular, we have shown that the *positive* monotone coupling is very effective  for attention approximation in Transformers, giving accuracy gains of **+0.8%**. More work in this direction is certainly needed; we hope our broader perspective beyond variance reduction will help guide it. As such, we believe that the counterintuitive nature of the findings is not a weakness of our paper, but rather an important strength.
>
> For the reasons above, whilst we agree that some of our mathematical arguments are technically involved, we respectfully suggest that our algorithms (Def 3.3 an Def 4.2) are actually **both simple to implement and empirically very effective**.
>
> As a minor point, we are unsure what the reviewer means by ‘guarantees can be proven only for close enough arguments of the kernel function to estimate’. For RFFs, our theoretical guarantees for improving estimates of $k(\boldsymbol{x},\boldsymbol{y})$ hold provided $\|\boldsymbol{x} - \boldsymbol{y}\|_2$ is sufficiently small. This is a very standard asymptotic condition for proofs with RFFs in the literature [1,3], and in our experiments we do not find it to be restrictive. If we misunderstand this comment, please may they clarify?
>
> CONTINUED BELOW.

---

> ### Author Response · Authors · 2024-11-15
> **Thanks for the review (part 2)**
>
> **Other questions and minor points**
> 1. *Notation and 'Monte Carlo sampling’*. In Eq. 1, each feature indeed depends on the $m$ random frequencies, so we agree that $\phi(\boldsymbol{x}; \\{ \boldsymbol{\omega}_i \\} )$ might be clearer. Eq. 1 is not yet a Monte Carlo estimator; it becomes one when we replace the expectation by the average of empirical samples drawn from $p(\\{\boldsymbol{  \omega  }\\})$. We have now phrased this more carefully. Thanks for the suggestions.
> 2. *No second term in $\mathcal{I}(\mu)$*. We can omit the $\mathbb{E}(\phi^\top \phi)^2$ term from our objective because, for *any* coupling with the correct fixed marginals, the estimator is unbiased. Hence, $\mathbb{E}(\phi^\top \phi)=k$, the true kernel evaluation. Since this term in the objective will be the same for any valid coupling, we omit it. This is not specific to the RFF and RLF case. It will be true for variance reduction with any MC random feature.
> 3. *Intuition for GRFs*. Thanks for the interesting question. When we couple the lengths of random walks, we 'diversify’ the ensemble, sampling walks with a different number of hops. This is a very direct way to ensure we explore the graph more effectively, avoiding (by chance) simulating pairs of very similar walks. It is intuitive that improving `coverage’ of the graph means we estimate the kernel more effectively, letting us model more interactions between different graph nodes. This may explain why our method is so effective in this setting. However, we stress that coupling schemes for estimators defined on discrete spaces like graphs were only introduced very recently [11], so further work is needed for a rigorous understanding of this phenomenon.
>
> We again thank the reviewer for reading the manuscript. We hope that, having clarified the practicality of our method and addressed minor concerns, they will consider raising their score. We warmly invite them to respond with any further questions.
>
> ____________
> [1] Orthogonal Random Features, Yu et al., NeurIPS 2016,  https://doi.org/10.48550/arXiv.1610.09072.
> [2] Geometrically Coupled Monte Carlo Sampling, Rowland et al., NeurIPS 2018,  https://mlg.eng.cam.ac.uk/adrian/NeurIPS18-gcmc.pdf.
> [3] Simplex Random Features, Reid et al., ICML 2023,  https://doi.org/10.48550/arXiv.2301.13856
> [4] Chefs’ Random Tables, Likhosherstov 2023, NeurIPS 2022,  https://doi.org/10.48550/arXiv.2205.15317
> [5] Quasi Monte Carlo Maps for Shift Invariant Kernels, Yang et al., ICML 2014,  http://proceedings.mlr.press/v32/yangb14.html
> [6] FastFood, Le et al., ICML 2013, https://doi.org/10.48550/arXiv.1408.3060
> [7] Structured adaptive and random spinners for fast machine learning computations, Bojarski et al, AISTATS 2017, https://doi.org/10.48550/arXiv.1610.06209
> [8] The unreasonable effectiveness of structured random orthogonal embeddings, Choromaksi et al., NeurIPS 2017, https://doi.org/10.48550/arXiv.1703.00864
> [9] Spherical structured feature maps for kernel approximation, Lyu et al, ICML 2017, http://proceedings.mlr.press/v70/lyu17a/lyu17a.pdf
> [10] Random features for shift-invariant kernels with moment matching, Shen et al, AAAI 2017, https://doi.org/10.1609/aaai.v31i1.10825
> [11] Quasi Monte Carlo Graph Random Features, Reid et al., NeurIPS 2023,  https://doi.org/10.48550/arXiv.2305.12470

---

> > ### Comment · Reviewer_huMM · 2024-11-16
> >
> > I thank the authors for their careful reply to my comments. I take the opportunity to clarify some points I raised and their motivation and possibly clarify some doubts.
> >
> > My observation about the possible lack of simplicity was due to the following: I have the impression that once a given (but general) RF measure is assumed, the subsequent optimal OT strategy has to be in general designed and certified "case by case". I agree that, for RFFs and RLFs, this task has been precisely performed by the authors in the present submission, leading to a simple algorithm. It is not obvious to me if it is obvious that other kinds of random features might lead to simple and efficient enough OT reformulation, but please correct me if I am wrong.
> >
> > Moreover, I agree with the authors that counterintuitive results are important (and I very appreciated that part of the paper) but, unless I am misinterpreting, this aspect of the submission can be interpreted in two ways. It can be seen as a weakness if the submission is intended as focused on efficient variance reduction strategies, as it raises the question of whether variance reduction is the correct strategy to pursue after all. If this is not the case, one might wonder why putting an effort into implementing an OT strategy is relevant given that, a priori, there is no guarantee that optimal (in some sense depending on the task) results will be affected by such optimization and, in general, if any extremization procedure in the OT problem leads to optimal results. On the other hand, another way of reading the contribution might be the following: optimal variance reduction strategies might not lead to optimal results. As the authors say, such evidence of the limitations of this general framework is a strength. One might argue that this reading key of the manuscript might actually transform the aforementioned ''weaknesses'' into strengths. I would appreciate a comment from the authors in this sense.
> >
> > *Minor points*
> > I would like to thank the authors for clarifying essentially all my doubts (e.g., about the notation and the reported expression for $\mathcal I(\mu)$) and to have pointed out the possible reason for the effectiveness of the strategy for GRFs.
> >
> > About the comment on the RFFs guarantees, it was due to the possible disrupting effects of pairs $(\boldsymbol x,\boldsymbol y)$ not satisfying the hypothesis. This was anyway a minor point: my observation therein was not specifically on the fact that the result was to be intended as ''weak'' but instead on the fact that a hypothetical general case might be tricky to analyze as even for the RFF case (which allows a quite complete analysis) an additional, although result-wise not impactful, assumption on the kernel arguments had to be made.

---

> ### Author Response · Authors · 2024-11-16
> **Thanks for the response**
>
> We sincerely thank the reviewer for their swift and detailed reply. We are grateful for their interesting perspectives. We are happy to share our thoughts below.
>
> *Generality of the framework and lack of simplicity*. The reviewer is correct to note that, since we take the RF estimator variance as an OT cost function, the OT map will in general differ depending on the specific RF construction. Therefore, the ‘difficulty’ of the problem is upper bounded by the difficulty of OT, and by constructing an ‘adversarial’ RF one could in principle make the variance reduction problem very difficult or intractable. However, we respectfully raise the following points.
> 1. We already consider two Euclidean RF constructions (RFFs and RLFs) and two different kernels (Gaussian and softmax). Up to an asymptotic assumption for RFFs, the OT formulation yields the same, simple coupling. It is not *a priori* obvious that this should be the case, since the two RFs considered use totally different basis functions (exponentials vs sine and cosine). In this sense, the PNC coupling is already actually quite general.
> 2. Moreover, in the literature, it is often found that coupling strategies that are effective for the Gaussian kernel – like conditioning orthogonality [1] – are also effective for a range of other radial basis function kernels such as the Matérn kernel [2]. Whilst we defer rigorous study to future work, we are optimistic that this norm coupling strategy may also reduce the variance of other kernels. It seems intuitively reasonable that trying to sample the norms of frequency vectors more diversely might improve estimator convergence for a wider variety of kernels.
> 3. Lastly, since the Gaussian and softmax kernels are so ubiquitous in machine learning (both for attention in Transformers and in Bayesian ML), we suggest that variance-reducing couplings *specifically* for these cases may still be of general interest. All the papers cited in our previous rebuttal focus on these examples; it is common for researchers to tackle this special case.
> For these reasons, we suggest that our method will be of general interest.
>
> *Counterintuitive results for variance reduction*. Thanks for the balanced perspective. We agree that there are different possible interpretations of our findings, although (predictably!) we tend to prefer the latter suggested by the reviewer. This is for the following reasons.
> 1. *Evidence that variance reduction is not the correct strategy is very important for the field*. As cited above, very many papers and hours of research have been dedicated to variance reduction. Our smoking gun evidence that variance is *not* the last word for applications (as the reviewer says, we actually **’solve’** the variance reduction problem for RFFs/RLFs in particular special cases) gives compelling motivation to redirect and broaden these efforts, thinking about ‘optimal couplings’ in a more general sense than just optimising the estimator variance. We believe this to be practically important for the field’s future direction, even if in some sense it could be considered a null result.
> 2. *OT is not specific to variance reduction*. Moreover, whilst we mostly focus on coupling for variance reduction (apart from the brief section about Transformers where we *maximise* variance), we stress that the OT framework is not specific to this case. The reviewer also pointed out this important observation earlier. In principle, the OT framework can be used to optimise *any* property of the estimator. Whilst we agree that the OT problem may in general be difficult depending on the choice objective, we believe that our perspective and research toolbox – both theoretical (Thm 3.2, Corollary 3.4, Thm 3.5, Thm 4.1) and numerical (Sec. 3.2, Sec. 4.1) – will be helpful.
>
> As such, we respectfully suggest that the counterintuitive findings are not a problem with the OT reformulation per se (read: the optimisation method), but rather with variance reduction as a goal (read: the objective function). Our results will be important in the quest for better couplings.
>
> Once again, we sincerely thank the reviewer for the response. Their thought-provoking questions have improved the manuscript. If satisfied with the above, we again politely ask that they recommend acceptance.
>
> ________________
> [1] The Geometry of Random Features, Choromanski et al., AISTATS 2018, https://proceedings.mlr.press/v84/choromanski18a.html
> [2] Orthogonal Random Features, Yu et al., NeurIPS 2016,  https://doi.org/10.48550/arXiv.1610.09072.

---

> > ### Author Response · Authors · 2024-11-18
> > **Manuscript updated**
> >
> > This a brief note to let the reviewer know that we have uploaded a new version of the manuscript. Changes are marked in red. They may be pleased to see that we have updated the notation in Eq. 1 and corrected our error on line 46. We also added a new section providing intuition for the good performance of coupled GRFs, beginning on line 2013. Thanks for these great suggestions! Having made these changes and (we hope) allayed all other concerns, we respectfully ask that the reviewer consider raising their score and recommending acceptance.

---

> > > ### Comment · Reviewer_huMM · 2024-11-20
> > >
> > > I would like to thank the authors for the reply and the update.
> > >
> > > I am still concerned about the generalizability of the approach proposed in the manuscript (beyond the general idea of merging two fashionable topics such as OT and RFs). I agree with the fact that OT is a very general framework and likely many optimization problems involving measures can be cooked as an OT problem. Surely OT might play a role in defining a good optimization rule aiming at some performance goal, but this is intrinsic to the nature of the tool, and the challenge is to identify *where and how* to apply it in a *manageable and efficient* way. On top of this, as I commented in my previous reply, one possible take of the submission is that variance reduction is *not* in general the correct strategy to follow, an observation that might lead to the question of why an OT formulation of a variance reduction strategy is relevant after all. Finding the correct optimization problem to solve possibly with OT is what we are left with (one might say we are left with the hammer, that we already had, without knowing what nail to use it on).
> > >
> > > Nevertheless, RFFs and RLFs are indeed important RF construction (although I would say that exponentials and sine/cosine basis are not *totally different basis functions*...). Moreover, the presented analysis might be of interest from the theoretical point of view to show how variance reduction can ultimately fail in a performance boost in prototypical cases. It seems from their reply that the authors see the limitation analysis as one of the main delivery of the paper: I recommend stressing this point even further. I am raising my score just above the acceptance threshold due to this fact, i.e., the possible contribution of the paper in clarifying the role and importance of variance reduction strategies beyond the specific OT formulation.

---

> ### Author Response · Authors · 2024-11-21
> **Thanks for the response**
>
> We thank the reviewer for continuing to engage so wholeheartedly in the reviewing process.
>
> *Generalisability*. Following the reviewer’s comments, to further demonstrate the generalisability of our approach, **we have now added an extra experiment using OT-coupled RFs to approximate the Matérn kernel**. Across a range of smoothness parameters $\nu$ describing very different function behaviours, we again find that our algorithm provides excellent variance reduction, outperforming all baselines. This demonstrates that our approach can be used to improve estimates of a totally different, very popular class of covariance functions, beyond the Gaussian example. Please take a look at Sec G.1 (page 39), especially Fig. 12.
>
> To summarise, this means that our framework has now been successfully applied:
> 1. To RFFs and RLFs;
> 2. To estimate the Gaussian kernel and the Matérn kernel (the latter for smoothness parameters $\nu \in \\{1/2,3/2,5/2 \\})$;
> 3. For GRFs to estimate kernels on graph nodes;
> 4. For both real and synthetic graphs (where variance reduction gives big downstream gains);
> 5. And for estimating the PageRank vector (a totally different application, which isn’t even about RFs! See App. E for full details).
>
> We firmly believe that this shows that our approach generalises well, across RF basis functions, target kernels, and even input domains. It is very typical for papers to focus on just one of these instances listed above, e.g. Yu et al. [1] just consider approximating the Gaussian kernel with RFFs, and Reid et al. [2] just consider improving approximating of the regularised Laplacian graph kernel with GRFs. On the other hand, we provide substantial gains for all of them.
>
> *Variance reduction as a goal*. We remind the reviewer that, for GRFs, **variance reduction gives big, SOTA gains in downstream inference tasks** (see e.g. Fig. 4 and Fig. 11). Clearly, variance reduction *can* bring benefits and should still be considered an important goal for Monte Carlo research -- hence its continued popularity. However, for RFFs and RLFs, we do indeed find that the benefits of variance reduction saturate. Alternative properties for the coupling are found to be important e.g. in efficient Transformers, for which we suggest an interpretation in terms of OT. To be clear, we still believe variance reduction to be an important research enterprise. However, **pushing it to its limits using OT has revealed that we may *also* need to consider optimising other properties of the coupling**. We will make this nuanced point clearer in the text. Thanks for the suggestion.
>
> *Summary of novel contributions*. To avoid getting bogged down in the specific part of the paper about the limitations of variance reduction for RFFs/RLFs, we thought it might be helpful to recapitulate the **paper’s main novel contributions**.
> 1. First application of OT to improving the convergence of RFs;
> 2. First analytic solution to OT problem for orthogonal RFFs/RLFs (may be of independent interest to OT researchers, Thm 3.2);
> 3. Novel numerical methods for when the OT problem is analytically intractable:
> - Numerical copula approach for optimising couplings in Euclidean space (Sec. 3.2);
> - Permutation densities and Hungarian algorithm for optimising couplings in discrete space (Sec. 4.1).
> 4. First data-based optimisation of a coupling between random walks, and first application of GRFs to Gaussian processes on graphs (where our variance reduction techniques unlock better approximate inference and beat all baselines -- see Sec. 4.2).
> 5. (*Already discussed in detail* -- demonstration that benefits from variance reduction can saturate with RFFs/RLFs, and proposal of other properties of the coupling to optimise (Sec. 3.3)).
>
> To run with the hammer and nail analogy suggested by the reviewer: we have substantially upgraded the hammer (points 1-3), introduced some new nails (point 4), but found that at some point existing nails may stop benefiting from a better hammer (point 5). We believe all these contributions to be integral to our work.
>
> Once again, we sincerely thank the reviewer for their engagement and for encouraging us to spell out these points more clearly. With the additional Matérn kernel experiments and discussion in mind, we politely ask whether they might consider a further score increase.
>
> ____________
> [1] Orthogonal Random Features, Yu et al., NeurIPS 2016,  https://doi.org/10.48550/arXiv.1610.09072.
> [2] Quasi Monte Carlo Graph Random Features, Reid et al., NeurIPS 2023,  https://doi.org/10.48550/arXiv.2305.12470

---

> > ### Author Response · Authors · 2024-11-26
> >
> > As the phase of the discussion period during which we are able to update the PDF draws to a close, this is a final polite reminder that we will be happy to discuss any remaining questions.
> >
> > We believe that the comments made above, including the **new Matérn kernel experiments added to Sec G.1** (page 40), have provided very strong evidence of our framework's generalisability. Thanks for prompting these additions and clarifications. When you have time, please may you take a look? We also hope that our concise summary the paper's main novel contributions, including *beyond* variance reduction for RFFs and RLFs, proves helpful.
> >
> > May we clarify anything else, or do you consider your concerns resolved? If so, we respectfully ask that you consider a further score increase. Thanks once more for your efforts.

---

> > > ### Comment · Reviewer_huMM · 2024-11-27
> > >
> > > I would like to acknowledge the answers of the authors and thank them for their effort in adding to the manuscript supporting evidence of the effectiveness of PNC beyond my original concerns. I think that some points raised by the manuscript, although 'counterintuitive', are interesting and I appreciated the authors' effort to make more clear how their analysis helps to put the importance of variance reduction in context. I confirm my previously increased rating of the manuscript above the acceptance threshold.

---

### Official Review · Reviewer_93BC · 2024-10-31

**Soundness:** 3
**Presentation:** 3
**Contribution:** 3
**Rating:** 6
**Confidence:** 2

**Summary:**

The paper explores variance reduction techniques for random features (RFs) in kernel methods to improve efficiency in large-scale machine learning tasks. By framing variance reduction as an optimal transport (OT) problem, the authors propose couplings that enhance the convergence of RFs on both Euclidean and discrete input spaces. They introduce novel coupling techniques for various RF classes, including Random Fourier Features (RFFs), Random Laplace Features (RLFs), and Graph Random Features (GRFs). The paper highlights the effectiveness of the proposed methods in reducing variance and improving downstream performance in certain cases, although it also discusses limitations.

**Strengths:**

1. **Innovative Use of Optimal Transport**: The paper employs OT as a novel framework to address the variance reduction problem in RFs, which is a creative approach that ties together theoretical insights with practical application.
2. **Comprehensive Coverage**: It addresses both Euclidean and discrete input spaces, providing a unified strategy applicable across different domains.
3. **Theoretical Guarantees and Empirical Validation**: This paper offers theoretical guarantees and empirical validations, enhancing the understanding of variance reduction in RF performance.

**Weaknesses:**

1. The definition and relationship between variance reduction in kernel estimation and optimal transport should be more detailed and introduced for non-specialists to understand directly.
2. The computation and theorem (Theorem 3.2) is only for $m=2$, and the authors apply the copula tool as numerical OT solvers for multi-marginal OT.
3. Despite variance reduction, the downstream impact on model performance can be inconsistent, especially for transformers.
4. More comparisons between this paper and [1] and [2] should be clarified, since some of the idea and technical proof skill is from these paper.


[1]. *General Graph Random Features*. Nips 2023.
[2]. *Simplex Random Features*. ICML 2023.

**Questions:**

1. What are the computational implications of implementing these optimal transport techniques at scale?
2. Can the theoretical findings on variance reduction be generalized to other kernel methods or models?

**Details Of Ethics Concerns:**

None details of your concerns beyond what is described above.

---

> ### Author Response · Authors · 2024-11-15
> **Thanks for the review**
>
> We thank the reviewer for their comments on the manuscript. We are pleased that they find our approach innovative and practical, and that they appreciate the benefits of a unifying perspective applicable to both Euclidean and discrete spaces. We address all their questions and concerns below.
>
> 1. *Relationship between OT and variance reduction*. The Kantorovich formulation of OT seeks to find a joint measure that minimises the expectation of some cost function, conditioned on fixed marginal measures. Meanwhile, a MC estimator will be unbiased if each random variable’s marginal measure is equal to the target measure under which we would like to estimate the expectation. Therefore, simply letting the OT cost function be the variance of the MC estimator, it follows that variance reduction can be considered an OT problem. We will be sure to make this clear in the text.
> 2. *Theorem 3.2*. Thm 3.2 exactly solves the OT problem for RFFs and RLFs with $m=2$. Corollary 3.4 extends this to propose a coupling scheme that is strictly better than i.i.d. for *any* $m$, not just $m=2$. Sec 3.2 then runs numerical experiments to show that this coupling is actually close to optimal – i.e. the (generically intractable) solution to the *multi-marginal* optimal transport problem – for any $m$. Respectfully, we do not believe any of the above to be a weakness of the paper. We will be happy to respond if the reviewer can further clarify.
> 3. *Variance reduction does not always improve downstream performance*. The reviewer is correct to identify that a surprising finding of the paper is that optimising a coupling for variance reduction may not always improve downstream performance. In fact, in Table 2 we give evidence that optimising a *different* property of the coupling – i.e. solving a different OT problem – can actually improve downstream performance by more. This leads to a **+0.8%** accuracy gain with a Performer ViT on ImageNet [1]. We believe these observations to be underappreciated in the literature, which almost exclusively focuses on variance reduction [2,3,4,5,6,7,8,9,10,11]. Highlighting these problems and presenting effective alternatives is a major contribution of this work. Respectfully, we do not consider this important part of our work to be a weakness.
> 4. *Relationship to other papers*. Thanks for the question. ‘General Graph Random Features’ [12] first proposed the GRFs algorithm for estimating graph node kernels. However, the walkers were taken to be i.i.d. – here, we improve on it by coupling walkers’ lengths. Observe that we comprehensively beat this baseline in Figs 3 and 4. Meanwhile, ‘Simplex Random Features’ [4] improved RLFs by coupling the directions of the frequency vectors. Importantly, it did not couple their norms. The reviewer is correct that we build on the theoretical analysis in this paper, but substantially extend it by introducing OT and for the first time coupling vector *norms*.
> 5. *Computational implications of implementing these techniques at scale*. PNC (Def 3.3) is trivial to implement at scale, and is no more expensive than i.i.d. sampling of norms even though it guarantees lower kernel estimator variance. Meanwhile, the GRF coupling is learned very efficiently using the Hungarian algorithm (line 430), and in our experiments we scale it to massive graphs with as many as $8700$ nodes (see Fig. 4). Indeed, our algorithms are *designed to scale well to massive datasets* – this is one of their key strengths. We will be sure to make this clear.
> 6. *Can the results be generalised to other kernel methods and models?* We already consider kernels defined on both Euclidean and discrete spaces, and GRFs can approximate any graph node kernel that is written as a function of a weighted adjacency matrix. **This is already a very broad class**. Whether the negative monotone coupling (Eq 7) is optimal for any other Euclidean kernels beyond Gaussian and softmax is an excellent open question, which we defer to future work. Thanks for raising it.
>
> Once again, we thank the reviewer. Having provided detailed clarifications to all their questions, we hope they will consider raising their score. We warmly invite them to respond.
>
> CONTINUED BELOW.

---

> > ### Author Response · Authors · 2024-11-15
> > **Thanks for the review (part 2)**
> >
> > ______________
> >
> >
> > [1] Rethinking Attention with Performers, Choromanski et al., ICLR 2021, https://arxiv.org/abs/2009.14794
> > [2] Orthogonal Random Features, Yu et al., NeurIPS 2016,  https://doi.org/10.48550/arXiv.1610.09072.
> > [3] Geometrically Coupled Monte Carlo Sampling, Rowland et al., NeurIPS 2018,  https://mlg.eng.cam.ac.uk/adrian/NeurIPS18-gcmc.pdf.
> > [4] Simplex Random Features, Reid et al., ICML 2023,  https://doi.org/10.48550/arXiv.2301.13856
> > [5] Chefs’ Random Tables, Likhosherstov 2023, NeurIPS 2022,  https://doi.org/10.48550/arXiv.2205.15317
> > [6] Quasi Monte Carlo Maps for Shift Invariant Kernels, Yang et al., ICML 2014,  http://proceedings.mlr.press/v32/yangb14.html
> > [7] FastFood, Le et al., ICML 2013, https://doi.org/10.48550/arXiv.1408.3060
> > [8] Structured adaptive and random spinners for fast machine learning computations, Bojarski et al, AISTATS 2017, https://doi.org/10.48550/arXiv.1610.06209
> > [9] The unreasonable effectiveness of structured random orthogonal embeddings, Choromaksi et al., NeurIPS 2017, https://doi.org/10.48550/arXiv.1703.00864
> > [10] Spherical structured feature maps for kernel approximation, Lyu et al, ICML 2017, http://proceedings.mlr.press/v70/lyu17a/lyu17a.pdf
> > [11] Random features for shift-invariant kernels with moment matching, Shen et al, AAAI 2017, https://doi.org/10.1609/aaai.v31i1.10825
> > [12] General Graph Random Features, Reid et al., ICLR 2024, https://arxiv.org/abs/2310.04859

---

> > > ### Comment · Reviewer_93BC · 2024-11-20
> > > **Thank you for the response**
> > >
> > > Thank the authors for their effort and very detailed response to my comments. After going through the other reviewers' comments and the authors' responses, I choose to keep my positive rates.

---

> > > > ### Author Response · Authors · 2024-11-20
> > > > **Any other questions?**
> > > >
> > > > We thank the reviewer for their response. **Is there anything else that we may clarify for them, or do they consider all their questions and concerns resolved?** If so, we respectfully ask that they consider raising their score.

---

> > > > > ### Author Response · Authors · 2024-11-24
> > > > > **Discussion period drawing to close: any further questions?**
> > > > >
> > > > > As the discussion period draws to a close, this is a polite reminder that we will be happy to answer any further questions the reviewer may have. We believe that we have resolved all their concerns, having explained the relationship to existing works they identified in the literature and shown how these techniques can be applied at scale. **If the reviewer is happy with the above, we respectfully ask that they consider raising their score before the deadline**. We thank them again for their time and efforts.

---

### Official Review · Reviewer_GwbX · 2024-11-01

**Soundness:** 3
**Presentation:** 4
**Contribution:** 3
**Rating:** 6
**Confidence:** 4

**Summary:**

The paper presents a novel approach for improving kernel estimation methods via random features. By utilising optimal transport, they tackle the specific cases of random fourier features, random laplace features, and graph random features. They demonstrate this technique on a range of real world datasets. And while some practical advantages are found, it is also noted that reducing the kernel variance does not necessarily improve the quality of the forecasts.

**Strengths:**

The manuscript is well presented, structured, and clearly written.

There is a high degree of novelty in the application of optimal transport in determining the feature couplings.

The proposed method is tested on a good selection of real world datasets.

**Weaknesses:**

I have some concerns regarding some of the experimental results presented in the manuscript:

Regarding Table 1 - Running only d features for RFF strikes me as a rather low number to choose, especially for these modestly sized datasets. It would be helpful to clarify how the results impacted when using a larger numbers of features. It would be great to see something analogous to the Figure 1 of the Yu et al 'Orthogonal Random Features' paper which depicts 1 < d/D < 10.

While I can appreciate that normalisation of RMSE's to 1.0 makes for a clean reference point, this also obfuscates the absolute performance from the reader (which is significant for the purposes of reproducibility, and comparisons with other works).

Regarding Table 3 - In addition to, or instead of, providing the KL results, I think the predictive performance (NLL or NLPD) would be preferable. I believe this would provide a better proxy for the model evidence. My concern with the KL divergence is that it does not heavily penalise models which are overconfident.

**Questions:**

Regarding the choice of metric, what are the benefits of using KL over predictive NLL (or NLPD)? I noticed that on page 22, it is mentioned that the predictive NLL is computed, so perhaps I missed where these results are presented.

Are there applications where the kernel reconstruction is the primary focus, rather than the derived confidence intervals? If so it might be beneficial to emphasise these to the reader.

One further small suggestion - at present the authors describe as "surprising" the result that kernel accuracy improvement often does not lead to downstream test improvements. On the contrary, I think there have been several reports of this phenomenon in the literature. For example, from Liu et al 2004.11154 "We find that ORF and SSF yield the best approximation quality. Despite that most algorithms achieve different approximation errors, there is no significant difference in the test accuracy."

---

> ### Author Response · Authors · 2024-11-15
> **Thanks for the review**
>
> We sincerely thank the reviewer for their thoughtful comments on the manuscript. We are pleased that they believe it to be well written and highly novel, and that they agree we test on a good selection of real world datasets. We address all their concerns and questions below.
>
> *Number of features plot*. Thanks for the suggestion. **We actually already include this in Fig 6**: a plot showing the kernel estimator RMSE, prior and posterior KL divergence, and predictive RMSE against numbers of features $m/d \in \\{1,2,4\\}$. This is exactly analogous to Fig. 1 in 'Orthogonal Random Features’ by Yu et al. [1]. Note also that, for all methods considered, the kernel estimator RMSE goes down as $\sim (m/d)^{-1/2}$ as we introduce extra `blocks’ of size $d$. This is because the couplings are within each block rather than between different blocks. Therefore, the *ratio* of respective variances between the different methods does not change as $m/d$ grows, since they all just scale down together. In this sense, just considering $m=d$ to show variance reduction is sufficient (though of course other downstream quantities will saturate as the estimators converge to the true kernel).
>
> *Normalisation of RMSEs*. Thanks for the suggestion. As the reviewer says, we normalised the predictive RMSEs for simplicity of comparison. However, we agree that including the overall scales might be useful for reproducibility, **so have added these to the manuscript**. (With that said, we do emphasise that the purpose of this experiment is to demonstrate greater variance reduction compared to previous variance reduction approaches, rather than necessarily achieving SOTA performance on the UCI regression tasks).
>
> *KL and NLL*. Thanks also for this suggestion. Our intention with the experiments in Sec. B.2 is to compare the quality of approximate inference when the kernel is estimated with different methods. To achieve this, we report the KL divergence between the true and approximate GP predictive posteriors (Table 3), which captures how close the approximate posterior is to the result with the full rank kernel. We also report the predictive RMSEs on held out test sets (Table 4), which straightforwardly measures the quality of the predictions. *Note that we are more interested in quantifying the accuracy of our approximate predictor compared to the groundtruth with an exact kernel, rather than making a statement about the predictive performance of the exact kernel in absolute terms*. That said, other choices like the predictive NLL are also possible. When we compute the NLL In experiments, we find that it shows similar behaviour as our existing metrics: variance reduction by coupling does not always improve approximate inference. We initially omitted these NLL results for brevity – the paper is already 38 pages! However, we agree that they might be a nice addition, **so we have now added them to the paper**. Thanks again for the comment.
>
> *Are there applications where kernel reconstruction is the primary focus?* We certainly find that the importance of kernel approximation quality depends on the particular setting and downstream task – e.g. for RFFs and RLFs variance reduction by coupling norms appears less effective, whereas for GRFs we see very big gains. Understanding this complicated behaviour is challenging. We hope our results will prompt future work in this direction.
>
> *On being 'surprised’*. Thanks for the extra pointer to Liu et al. [2004], which we have incorporated into the manuscript. However, we still respectfully suggest that the complicated relationship between variance reduction and downstream performance is not always fully appreciated in the literature. An extraordinary number of papers have been published on variance reduction, including in top ML conferences [1,2,3,4,5,6,7,8,9,10]. But here we show that a modification as simple as normalising attention scores by their row-wise sum can mean that the *maximum* variance coupling actually works better (from line 287). More detailed, thorough work is needed to optimise the `right’ properties of the MC coupling. We hope that our work will prompt further investigation, and that it will help provide mathematical and algorithmic tools by making the connection to optimal transport.
>
> Once again, we thank the reviewer for their helpful comments and suggestions. We believe that we have addressed all their concerns, and respectfully ask that they consider raising their score. We warmly invite them to respond with any further questions.
>
> CONTINUED BELOW.

---

> > ### Author Response · Authors · 2024-11-15
> > **Thanks for the review (continued from above)**
> >
> > _________________
> > [1] Orthogonal Random Features, Yu et al., NeurIPS 2016,  https://doi.org/10.48550/arXiv.1610.09072.
> > [2] Geometrically Coupled Monte Carlo Sampling, Rowland et al., NeurIPS 2018,  https://mlg.eng.cam.ac.uk/adrian/NeurIPS18-gcmc.pdf.
> > [3] Simplex Random Features, Reid et al., ICML 2023,  https://doi.org/10.48550/arXiv.2301.13856
> > [4] Chefs’ Random Tables, Likhosherstov 2023, NeurIPS 2022,  https://doi.org/10.48550/arXiv.2205.15317
> > [5] Quasi Monte Carlo Maps for Shift Invariant Kernels, Yang et al., ICML 2014,  http://proceedings.mlr.press/v32/yangb14.html
> > [6] FastFood, Le et al., ICML 2013, https://doi.org/10.48550/arXiv.1408.3060
> > [7] Structured adaptive and random spinners for fast machine learning computations, Bojarski et al, AISTATS 2017, https://doi.org/10.48550/arXiv.1610.06209
> > [8] The unreasonable effectiveness of structured random orthogonal embeddings, Choromaksi et al., NeurIPS 2017, https://doi.org/10.48550/arXiv.1703.00864
> > [9] Spherical structured feature maps for kernel approximation, Lyu et al, ICML 2017, http://proceedings.mlr.press/v70/lyu17a/lyu17a.pdf
> > [10] Random features for shift-invariant kernels with moment matching, Shen et al, AAAI 2017, https://doi.org/10.1609/aaai.v31i1.10825

---

> > > ### Author Response · Authors · 2024-11-18
> > > **Manuscript updated**
> > >
> > > This a brief note to let the reviewer know that we have uploaded a new version of the manuscript. We hope they will be pleased to see Table 4 (page 25), which now includes negative log likelihoods and unnormalised predictive RMSEs as requested. We also added their helpful reference to Liu et al 2004.11154 on line 531. Thanks for both these suggestions! We hope that, with these clarifications and changes, they will consider raising their score and recommending acceptance.

---

> > > > ### Comment · Reviewer_GwbX · 2024-11-21
> > > >
> > > > I appreciate the authors efforts in updating the manuscript, and providing a thoughtful response.
> > > >
> > > > > Note that we are more interested in quantifying the accuracy of our approximate predictor compared to the groundtruth with an exact kernel, rather than making a statement about the predictive performance of the exact kernel in absolute terms.
> > > >
> > > > I fully agree with the authors' observation that one doesn’t need to present SOTA results on these datasets. However, I do feel that it's important to show empirical performances that are representative. And my main concern here is that some of the central results (Table 1) lie in a fairly impractical regime, owing to the choice of a very small number of features. And to illustrate this point, I'm not sure we are outperforming linear regression in many of these cases. I could be wrong but I believe linear regression scores about 9.5 for a test RMSE on Concrete which is lower than any of the methods presented here. I therefore remain concerned whether this fiducial setup represents a useful or fair demonstration of these different methods.
> > > >
> > > > The listed papers the authors provided in the above comment [1-10] provide a helpful collection of references focusing on variance reduction. But almost all are also accompanied with some meaningful benefit in downstream metrics (I think only 2 and 9 don't do so explicitly), and they typically achieve this with a significantly greater number of features than presented here.
> > > >
> > > > > We actually already include this in Fig 6: a plot showing the kernel estimator RMSE
> > > >
> > > > Ah indeed, but I would stress that Fig 6 is in the supplementary material. While I appreciate it's challenging to decide what material to fit within the confines of the main manuscript, I believe some discussion of the impact of the number of features in the main text would be greatly beneficial.
> > > >
> > > > Again I appreciate the thoughtful feedback, and will be happy to reconsider my score at the end of the discussion period.

---

> ### Author Response · Authors · 2024-11-23
>
> We appreciate the reviewer’s response and their thoughts about the best setting to test the downstream effects of variance reduction.
>
> *$m=d$ choice*. We initially chose $m=d$ features because at this point the **absolute** difference in variance between the algorithms is at its greatest. (As mentioned in our previous response, the **relative** difference does not change as $m/d$ grows in integer multiples because the correlations are within each ‘block’ of size $d$). Hence, we expected that any difference in downstream performance to be most obvious in this regime. This is indeed reflected in Fig. 6, where the improvements in predictive RMSE and KL divergence are found to be greatest to the left of the plots when $m=d$. This behaviour is also well-established in the literature. To give a few examples, see:
>
> 1. Table 2 by Yu et al. [1], where orthogonal random features only consistently appear to outperform i.i.d. features for SVM classification accuracy when $m \leq 4d$.
> 2. Fig 8 by Reid et al. [3], when the gap between i.i.d., orthogonal and simplex random features rapidly drops with $m$ in the KRR experiment when $1 \leq m/d \leq 10$.
> 3. Table 1 by Yang et al. [5], where the gap between the different methods on the regression task is found to drop as the number of features grows.
>
> The reason is that, with more features, all the estimates approach the true kernel evaluation. Therefore, the performance of each method saturates at the performance of the exact kernel. They soon become indistinguishable. It is not the case that one should only expect gains from variance reduction at large $m$.
>
> That said, we do agree that it may be helpful to *also* compare the algorithms in the setting with more features, where all the kernel approximations are better so the overall performance on downstream tasks is more competitive/realistic. **Therefore, we have repeated the experiments with $m=8d$ features and added the results to App. B.2**. Please see Table 4, page 26. The trend is the same: despite providing variance reduction compared to the baselines, the *downstream* benefits from PNC (and in this case actually variance reduction more generally) are less clear. Note that the results are consistent with Fig. 6 for a split of the power dataset, where the difference between all methods becomes small as they approach the groundtruth kernel. The reviewer will also note that with more features we outperform linear regression, which gives predictive RMSE $10.43 \pm 0.56$ on 10-fold cross validation on the concrete dataset (script to get this result temporarily uploaded to supplementary material). Thanks for prompting this addition to the paper.
>
> (Note that these preliminary results are averaged over fewer random seeds than Table 3 because of time constraints; in the final draft the standard errors will be smaller and minor
> discrepancies will be ironed out).
>
> *Fig 6*. Thanks for the comments about Fig. 6. We agree that it’s a nice contribution! We found it difficult to squeeze it into the main body with space restrictions: we would like to leave enough space for our important GRF results, where we *do* see very substantial benefits from variance reduction, and our extensive theoretical contributions. **However, we have updated the text to signpost interested readers to this figure more explicitly**.
>
> Thanks again for the feedback, and for confirming that you will be happy to reconsider your score at the end of the discussion period. We look forward to hearing your response, and warmly welcome any further questions.
>
> ________________
> [1] Orthogonal Random Features, Yu et al., NeurIPS 2016, https://doi.org/10.48550/arXiv.1610.09072
> [2] Geometrically Coupled Monte Carlo Sampling, Rowland et al., NeurIPS 2018, https://mlg.eng.cam.ac.uk/adrian/NeurIPS18-gcmc.pdf
> [3] Simplex Random Features, Reid et al., ICML 2023, https://doi.org/10.48550/arXiv.2301.13856
> [4] Chefs’ Random Tables, Likhosherstov 2023, NeurIPS 2022, https://doi.org/10.48550/arXiv.2205.15317
> [5] Quasi Monte Carlo Maps for Shift Invariant Kernels, Yang et al., ICML 2014, http://proceedings.mlr.press/v32/yangb14.html.
> [6] FastFood, Le et al., ICML 2013, https://doi.org/10.48550/arXiv.1408.3060
> [7] Structured adaptive and random spinners for fast machine learning computations, Bojarski et al, AISTATS 2017, https://doi.org/10.48550/arXiv.1610.06209
> [8] The unreasonable effectiveness of structured random orthogonal embeddings, Choromaksi et al., NeurIPS 2017, https://doi.org/10.48550/arXiv.1703.00864
> [9] Spherical structured feature maps for kernel approximation, Lyu et al, ICML 2017, http://proceedings.mlr.press/v70/lyu17a/lyu17a.pdf
> [10] Random features for shift-invariant kernels with moment matching, Shen et al, AAAI 2017, https://doi.org/10.1609/aaai.v31i1.10825

---

> > ### Comment · Reviewer_GwbX · 2024-11-24
> >
> > Thank you for your response, and the additional experiments. My main concern remains that choosing m/d=1 maximizes the difference in performance, but does not represent a practical regime for RFF. In practice, such a low ratio often leads to poor absolute performance, where as I mentioned, linear regression might even do better. This makes m/d=1 an unrepresentative choice for practitioners.
> >
> > Regarding the three references you mentioned:
> >
> > Table 2 in Yu et al. presents results for m/d=2, 4, 6, 8, 10, but not m/d=1.
> > Reid et al. and Yang et al. also both show performance extending to higher multiples of m/d within the main text.
> >
> > I would therefore recommend either incorporating a range of m/d values into the main text, as the above references have done, or choosing a more representative value of m/d that mitigates the risk of misleading the reader on realistic performance gains. While I appreciate the inclusion of experiments with more features in Appendix B.2, their placement there doesn't resolve the key issues within the main text.

---

> ### Author Response · Authors · 2024-11-24
>
> Thanks for the response. Let us be clearer on these points.
>
> Previously, the only experimental results in the main text for coupled RFFs and RLFs were kernel estimator RMSE (Table 1). This shows the precise amount of variance reduction – after all, our chief goal – so it is probably the most important result to include. Since we normalise each kernel estimator RMSE by the value for i.i.d. features, the results are the same for any $m=kd$ (or $m=2kd$ for RLFs), where $k \in \mathbb{N}$ is a positive integer. Table 1 is *not* specific to $m=d$. For example, on the concrete dataset, using orthogonal + PNC RFFs instead of i.i.d. RFFs will always reduce the variance by the same ratio, whether you take $m=d$ or $m=1000d$. We have made this clearer in the text.
>
> We agree that considering downstream performance (*predictive* RMSE, KL divergence between true and approximate posterior, NLLs, etc.) on a range of values of $m/d$ may be interesting, including when $m$ is big so performance is better/more realistic. This is why, following the reviewer's comments, we now include full results not only for 1) $m=d$ (where absolute differences in kernel estimator variance, and hence performance, are biggest), but also 2) $m=8d$ (where differences between methods become very small because all approximations are close to the exact kernel). The latter is of course substantially better than linear regression for all datasets. It would also be very simple to include more values of $m$ between these regimes (or yet bigger $m$) if the reviewer thinks this would be a nice addition – this is just a matter of space.
>
> Therefore, **our paper already includes all the results the reviewer is asking for**. It is simply a case of where they are located in the text – the main body or the appendix. We agree that this is an important decision. On reflection, we agree that the reader might benefit from seeing downstream results in the main text, so **we have moved the ablation over $m$ (now Fig. 1) into the main text**. To make space, we were forced to move the copula schematic into the Appendix – a tricky choice. Thanks for this suggestion, which has probably made our message clearer overall.
>
> Could the reviewer please confirm that this latest change – including downstream performance for multiple $m/d$ in the main text, rather than in the appendix – has resolved their remaining concerns?

---

> > ### Author Response · Authors · 2024-11-26
> >
> > A brief update: to communicate the message above even more directly to the reader, we have updated Fig. 1 in the main text to include the full range $m/d \in \\{1,2,4,8,12\\}$, extending beyond that considered by Yu et al.. We have also added test predictive RMSEs with 1) the exact kernel (to which we converge at large $m$) and 2) linear regression (which is naturally worse than kernel-based methods) as benchmarks. When they have time, would the reviewer please take a look?
> >
> > We trust that this latest reorganisation of material makes the text totally unambiguous on the points about which they raised earlier concerns. Thanks again.

---

> > > ### Comment · Reviewer_GwbX · 2024-11-26
> > >
> > > Thank you for the thoughtful updates, which have significantly improved the paper. The new Figure 1 efficiently conveys a great deal of information to the reader.  It might be worth justifying in the text why Fig 1 shows the result of a single split, as opposed to averaged results. I presume it was with the intention to illustrate the small scatter within a given split, although these are too small to see.
> > >
> > > >  It is simply a case of where they are located in the text – the main body or the appendix
> > >
> > > From a reviewing perspective, this is a major distinction: the main text needs to be self-contained, without relying upon additional information from the appendix or supplementary materials. (Indeed, the ICLR call states that reviewers need not read the Appendix at all!)  I agree that it can be challenging to decide which figures to include in the main text. If the authors are keen to reintroduce the copula schematic, I would suggest there might be scope to trim the introduction and/or background material, which at three pages runs a little on the long side at present.

---

> > > > ### Author Response · Authors · 2024-11-27
> > > >
> > > > Thanks for raising your score and the helpful feedback. We are pleased that you agree our updates have improved the paper. Wishing you all the best.

---

### Official Review · Reviewer_kXYZ · 2024-11-04

**Soundness:** 3
**Presentation:** 3
**Contribution:** 3
**Rating:** 8
**Confidence:** 3

**Summary:**

This paper addresses a well-known Monte Carlo method for estimating the value of kernel functions using random feature embeddings. In particular, it proposes a method to reduce the variance of the estimate by coupling individual random features. This task is framed as a general optimal transport problem and is further discussed in two different scenarios: first random Fourier and Laplace features, and second features obtained by a random walk on graphs, corresponding to graph kernels. In the first case, exact results for two features are obtained and some extensions to more features is discussed. Moreover a search algorithm over a parameterized family of couplings is proposed. In the second case, a simplified coupling of the length of the random walks is discussed.

Numerical experiments are presented for each case that show the effectiveness of these algorithms.

**Strengths:**

I find the general idea of the paper fascinating, namely formulating the variance reduction task as optimal transport. This approach can also be applied to similar Monte Carlo methods in future studies. It also allows for the application of various techniques in OT for studying and improving kernel estimation methods.

**Weaknesses:**

Although the paper is well-written and easy to understand, the presentation of the paper can be improved. Much of the technical details that are summarized in the main body cannot be easily understood without referring to the appendix. There are confusing notations and definitions, especially for the case of graph kernels. I will mention some of them in the next part.

I also have a more specific questions/concerns about the justification of this framework which is explained in the next part.

I am willing to raise my score if my concerns are addressed.

**Questions:**

My main concern is about the justification and also computational aspects of the proposed framework. As the authors explain, the main limitation of this work is that variance is reduced at a specific pair of points, which also leads to some limitations in the performance of downstream tasks. The condition of fixed marginals in the OT framework actually guarantees an unbiased kernel as a process, i.e. for every pair of points. Is not it more reasonable to minimize the variation of the entire process rather than a specific point?

The pointwise view also leads to a potential computational limitation. If I understood correctly, in practice one needs to generate an independent set of coupled features for each pair of points, which requires to run the variance reduction method once for each pair. This leads to a huge computational burden in a large-scale problem.

More specific questions regarding the presentation:

1- I am not familiar to the notion of negative monotone coupling and for me (7) does not make sense. Can you provide more explanation?
2- If I am correct, theorem 3.2. only works for sufficiently small z, this should be explained in the introduction before this equation to avoid confusion.
3-In line 290, what is the definition of \hat{a}_i?
4-In line 387, does "walker's direction" mean the random walk conditioned on its length?
5-In Theorem 4.1, I do not understand the definition of q^{i,j}. Also q^{i} is not exactly defined.

---

> ### Author Response · Authors · 2024-11-15
> **Thanks for the review**
>
> We thank the reviewer for their comments on the manuscript. We are pleased that they enjoyed reading it, and that they found it well written and easy to understand. We clarify all points of confusion and address their questions below.
>
> *Computational aspects and pairs of points*. Thanks for the question. As written, Eq. 2 indeed considers variance reduction for a single pair of points, and the optimal transport map might in general depend on this specific pair. However, in Sec. 3.1 we find that, after working through the maths, the optimal RLF and RFF coupling is actually *independent* of the data (Eq. 7). Hence, the negative monotone coupling in Eq 7 is optimal *for any data distribution*. Meanwhile, for GRFs, we average the cost function over different pairs of nodes of the graph to learn a coupling that works well for the whole kernel matrix (line 463). This is the same as the reviewer’s suggestion to ‘minimize the variation over the process’. We also find that the optimised coupling generalises very well to new graph topologies (line 466). To be clear, **one does not need optimise the coupling separately for every pair of datapoints – the coupling we use is either data-independent or the cost function is averaged over a training distribution**. Therefore, our methods continue to scale well to very large datasets, including massive graphs with 8700 nodes (see e.g. the experiments in Fig. 4).
>
> *Questions regarding the presentation*. Thanks for the questions. We are happy to clarify them all below, and will make sure the manuscript is as clear as possible on these points.
> 1. *Negative monotone coupling*. Draw a random variable $u_1$ from the uniform distribution on $[0,1]$. Push it forward with the inverse CDF of the Chi distribution, defining a new variable $\omega_1 = {F^{-1}}(u_1)$. The variable $\omega_1$ will follow the Chi distribution. Define $u_2 = 1-u_1$, and let  $\omega_2 = {F^{-1}}(u_2)$. The marginal distribution of $u_2$ is still uniform on $[0,1]$ so the marginal distribution of $\omega_2$ is still the Chi distribution. However, $u_1$ and $u_2$ are now negatively correlated, so $\omega_1$ and $\omega_2$ are also negatively correlated. This is the negative monotone coupling, shown in Eq. 7.
> 2. *Theorem 3.2 and asymptotic $z$*. For random Laplace features, the negative monotone coupling is *always* optimal. For random Fourier features, it is optimal provided $z$ is sufficiently small. Formulating this asymptotic statement precisely is technically involved so we put the details and proof in the Appendix (Lemma A.6). We also emphasise that, in experiments on real data, the negative monotone coupling performs extremely well even when $z$ is not small (see Table 1).
> 3. *Definition of $\textrm{MSE}(\widehat{a}_i)$*. We defined attention $a_{ij} = k(x_i,x_j)/ \sum_l k(x_i, x_l)$, the row-normalised kernel evaluations, on line 287.
> The estimator $\hat{a}_{ij}$ is obtained by replacing the exact kernel evaluations $k(x_i, x_j)$ with their stochastic Monte Carlo estimates $\hat{k}(x_i, x_j)$.
> Then $\textrm{MSE}(\hat{a}_i)$ averages the MSE of this estimator over the keys indexed by $j$, as shown in line 290. We have made this clearer in the text – thanks.
>
> 4. *’Expectation over walkers’ directions’*. Yes, $\mathbb{E}_\textrm{dirs}$ averages the walkers’ directions – i.e. which of their neighbouring nodes they choose at every timestep. It does not consider their lengths, which are instead shown in the first expectation of Eq. 12.
> 5. *Notation in Thm. 4.1*. The vector $\widehat{\psi}(q^{(i)})$ is defined on line 414. In Thm. 4.1, it would be clearer if we had written
> $\\{\widehat{\psi}(q^{(i)}), \widehat{\psi}(q^{(j)})\\}^n_{q=1}$ so we have updated the notation. Thanks for the suggestion.
>
> We again thank the reviewer for their time. We have clarified the efficiency of our scheme (especially, that we do *not* need to solve the OT problem separately for every possible pair of datapoints). We have also improved the presentation of the text where there were points of minor misunderstanding. We hope that the reviewer will consider raising their score, and invite them to respond with any further questions.

---

> ### Author Response · Authors · 2024-11-18
> **Manuscript updated: please take a look**
>
> This is a brief note to let the reviewer know that we have uploaded a new version of the manuscript. Changes are shown in red. They may be especially interested to see the updated notation on lines 288 (shows $\widehat{a}_{ij}$) and 419 (shows $\widehat{\psi}(q^{(i)})$) following their helpful comments. We look forward to hearing their thoughts, and hope that they will consider raising their score.

---

> > ### Author Response · Authors · 2024-11-24
> > **Any further questions?**
> >
> > As the discussion period draws to a close, this is a polite reminder that we will be happy to answer any further questions the reviewer may have. We believe that we have resolved all their concerns, having clarified that variance reduction does *not* have to be applied separately to every pair of points in the dataset and updated some small points of notational ambiguity. **If the reviewer is happy, we respectfully request that they consider raising their score before the deadline**. We thank them again for their time and efforts.

---

> > > ### Comment · Reviewer_kXYZ · 2024-11-26
> > > **Response to Reviewer's comments**
> > >
> > > Many thanks for clarification and apologies for my late reply. The authors responded my main concerns regarding computation and  pointwise optimization. My only remaining concern is that I still do not see how (7) reflects the above description. If I am correct, one of the two F functions is really the conditional CDF of a variable given the other one. So, I encourage the authors to clarify. At any rate, I have already raised my score.

---

> > > > ### Author Response · Authors · 2024-11-26
> > > >
> > > > Thanks for the response and for raising your score!
> > > >
> > > > Suppose we draw a random variable $u_1$ from the uniform distribution on $[0,1]$ and push it forward with the inverse CDF of the Chi distribution, defining a new variable $\omega_1 = {F^{-1}}(u_1)$. Define $u_2 = 1-u_1$, and let  $\omega_2 = {F^{-1}}(u_2)$.
> > > > Then $\omega_1$ and $\omega_2$ are both marginally distributed as a Chi distribution, but are negatively correlated because when one is small (i.e. since $u_1=0.1$) the other is conditioned to be big (i.e. then $u_2=0.9$).
> > > >
> > > > To relate back to Eq. 7, if $\omega_1$ and $\omega_2$ are generated in this way, then clearly $F(\omega_1) + F(\omega_2) = u_1 + u_2 = u_1 + (1 - u_1) = 1$ for any given draw of $\omega_1$ and $\omega_2$. Knowing one of the pair deterministically fixes the other. We hope this makes sense.
> > > >
> > > > Once again, we thank the reviewer for their efforts.

---

### Author Response · Authors · 2024-11-27
**Overall response: thanks for the reviews**

We thank the reviewers for their efforts. **We are pleased that, following clarification of minor points of misunderstanding and manuscript updates, essentially all concerns appear resolved and all reviewers recommend acceptance**. To summarise our improvements (shown in red):

1. Notational tweaks, correction of typos, and rephrasing of confusing passages
2. Reorganisation of some material between main body and appendices (line 216), inclusion of extra negative log likelihood results for the RFF/RLF experiments, and increase in the number of MC samples used in downstream applications (page 26)
3. Additional intuition for the effectiveness of coupling lengths for GRFs (line 2068)
4. Novel use of our algorithm to approximate the Matérn kernel, providing further evidence of its generalisability (page 40).

Once more, we thank the reviewers and the AC. Of course, we will be very happy to answer any new questions during the remainder of the discussion period.

---

### Meta-Review · Area_Chair_QJJF · 2024-12-20

**Metareview:**

This paper provides a novel approach to reducing the variance of Randon Fourier feature maps via a connection to optimal transport. All reviewers felt that the connection was interesting and new, and will likely be of interest to many in the ICLR community. There were some concerns regarding ultimate value to downstream applications, as reducing embedding variance did not always lead to better downstream performance. I think the authors make a convincing argument that highlighting this point can be viewed a feature instead of a bug, as many prior works focus on variance reduction. Overall, a nice that that we would be happy to see at ICLR.

**Additional Comments On Reviewer Discussion:**

Reviewers felt positively about the paper from the start. The authors were unnecessarily verbose in their responses during the rebuttal period, with multiple summaries of other discussions, request for followup, etc., but this seems to be the new norm, no matter how good the initial reviews. They did help clear up some concerns about value to downstream applications. See main response for further discussion about this.

---

### Decision · Program_Chairs · 2025-01-22

Accept (Poster)